# LlmSynthor: Macro-Aligned Micro-Records Synthesis with Large Language Models

## Abstract

Macro-aligned micro-records are essential for simulations in social science and urban studies. For instance, epidemic models of urban disease spread are only credible when micro-level records reproduce realistic individual mobility and contact patterns, while macro-level aggregates match real-world statistics such as case counts or travel flows. Still, large-scale collection of such fine-grained data is impractical, leaving researchers with only macro-statistics (e.g., travel surveys or case counts). Large Language Models (LLMs), leveraging rich real-world priors learned from vast corpora, excel at generating realistic micro-records, but standard record-by-record sampling is inefficient and fails to enforce alignment with target macro-statistics. Given this, we propose **LlmSynthor**, a framework capable of synthesizing realistic micro-records that are statistically aligned with target macro-statistics. LlmSynthor transforms a pre-trained LLM into a macro-aware simulator that incrementally builds a synthetic dataset through an iterative process. At each iteration, a batch of micro-records is generated to reduce the discrepancy between synthetic and target macro-statistics. By treating the LLM as a nonparametric copula for inferring joint dependencies over variable combinations, the iterative process ensures the synthetic data are macro-statistically aligned with the target marginals and joints. To address sampling inefficiency, we introduce LLM Proposal Sampling, where the LLM, guided by discrepancies, generates a plan of proposals, each defining specific values or ranges for all variables and specifying the number of records to generate. This enables the framework to minimize discrepancies efficiently while preserving the realism grounded in the LLM's priors. Evaluations on synthetic and real-world datasets (mobility, e-commerce, population) encompassing diverse formats and settings show that LlmSynthor achieves high record realism, statistical fidelity, and practical utility, positioning it broadly applicable across economics, social science, urban studies, and beyond. Source codes are in https://anonymous.4open.science/r/LLMSynthor.

## 1 Introduction

High-stakes decisions in domains like public health and urban planning are increasingly supported by agent-based simulations of complex human behavior (Von Hoene et al., 2025). At the micro-level, individual records capture behavioral detail such as mobility and contact patterns, which drive realistic dynamics. At the macro-level, aggregated statistics ensure consistency with population-level trends. Only when micro-records are collectively aligned with real-world macro-statistics can such simulations yield valid insights (Jiang et al., 2024), yet these data are *unattainable* because large-scale collection is infeasible due to both prohibitive costs and stringent privacy constraints (Bellovin et al., 2019). Consequently, researchers and policymakers must rely on macro-statistics, such as census reports or case counts, leaving a micro-macro gap (Zhou et al., 2022). The core challenge is therefore to synthesize realistic micro-records that are statistically faithful to these known macro-statistics.

This micro-macro synthesis task, however, is beyond the capabilities of existing generative paradigms. Current methods, from classical statistical models to modern deep generative networks (Goodfellow et al., 2014; Kingma et al., 2013; Ho et al., 2020), are ill-equipped, as they all require access to a large volume of micro-records that are unavailable for model fitting. Furthermore, their reliance on rigid parametric assumptions or implicit model biases often leads to the generation of unrealistic records, such as a six-year-old with a doctorate, necessitating inefficient post-hoc fixes like rejection sampling.

Figure 1: Comparison of generative approaches and LLMSYNTHOR. Left: Existing methods either rely on micro-records (Generative Models) or cannot enforce macro-level consistency (LLMs), leading to unrealistic or inefficient generation. Right: LLMSYNTHOR synthesizes realistic micro-records directly from macro-statistics, enabling efficient, format-agnostic, and macro-aligned generation.

Most of these approaches also lack the generality to handle the heterogeneous or unstructured data common in social sciences and urban studies (Ma et al., 2020), or require extensive manual engineering (Sun et al., 2018). This highlights the urgent need for a new framework that can synthesize realistic and complex micro-records that are statistically grounded, guided solely by macro-statistics.

The advent of Large Language Models (LLMs) offers a compelling yet insufficient solution. Harnessing rich real-world priors, they excel at generating realistic, complex, and even unstructured micro-records without requiring any fine-tuning, making them powerful universal generative priors (Achiam et al., 2023; Kojima et al., 2022; Kwon et al., 2024). However, existing strategies remain limited. Fine-tuning LLMs requires massive micro-records and significant computational resources, which are often unavailable in practice. In-context or few-shot prompting can condition local generations, but it still operates record by record, making the process both *inefficient and incapable of enforcing macro-statistical alignment* (Wang et al., 2024a). This inefficiency becomes critical when dealing with large-scale datasets, where generating sufficient high-quality data quickly is essential for population-scale simulations. As a result, synthetic datasets produced by standard LLM generation may be extremely time-consuming and appear realistic at the micro-level but remain statistically unfaithful to the target macro-statistics. The central challenge is therefore to transform the LLM's generation process into one that is both efficient and macro-statistically controlled.

To address this challenge, we present LLMSYNTHOR, a framework that turns the LLM into a macro-aware simulator of micro-records. LLMSYNTHOR generates synthetic micro-records and incrementally builds a synthetic dataset through an iterative feedback loop, guided only by the target macro-statistics. In each iteration, LLMSYNTHOR measures the discrepancy between its synthetic and target macro-statistics, then prompts the LLM to generate a corrective batch of micro-records that minimizes this gap. This process is enabled by two core technical innovations. *First*, we treat the LLM as a powerful nonparametric copula, allowing it to infer the complex, nonlinear joint dependencies between variables without rigid statistical assumptions. This allows LLMSYNTHOR to align the synthetic data with the target by matching all *available* marginal and joint macro-statistics. *Second*, to overcome generation efficiency bottlenecks, we introduce LLM Proposal Sampling, where the LLM creates a generation plan of micro-record proposals, each defining a localized joint distribution over all variables with an associated generation count. This approach uniquely combines the LLM's rich prior knowledge for micro-level realism with rigorous, macro-level statistical control. Figure 1 compares our framework with existing paradigms. In summary, our main contributions are:

- We introduce LLMSYNTHOR, which transforms a pre-trained LLM into a macro-aware simulator for synthesizing micro-records that preserve realism while aligning with target macro-statistics.

- We propose (i) a nonparametric copula interpretation of the LLM to capture joint dependencies over variable combinations and (ii) LLM Proposal Sampling to ensure efficient and realistic generation.

- We design a discrepancy-guided synthesis process that iteratively extends and aligns the synthetic and the target macro-statistics, providing rigorous, dataset-level statistical control.

- Evaluations on synthetic and real-world datasets (mobility, e-commerce, population) across diverse formats and settings show that LLMSYNTHOR consistently outperforms expert baselines in record realism, statistical fidelity, and practical utility, while maintaining broad versatility.

## 2 RELATED WORK

**Data Synthesis** Early data synthesis methods focused on explicit statistical control, using techniques like iterative proportional fitting (IPF) (Beckman et al., 1996; Mueller & Axhausen, 2011), Bayesian networks (Sun & Erath, 2015; Sun et al., 2018; Zhang et al., 2019), and copula models (Nelsen, 2006; Avramidis et al., 2009; Okhrin et al., 2017) to match marginals and preserve dependencies. While interpretable, these methods often rely on strong assumptions and face challenges with scalability and heterogeneity. Deep generative models, including VAEs (Apellániz et al., 2024; Tazwar et al., 2024), GANs (Xu et al., 2019; Baowaly et al., 2019; Esteban et al., 2017; Lopes et al., 2023), and recent diffusion- or flow-based models (Kotelnikov et al., 2023; Kamthe et al., 2021; Kim et al., 2022; Zhang et al., 2023a), have improved realism and high-dimensional modeling. However, they tend to entangle marginals with dependencies and often require expensive retraining for new domains. LLM-based methods, such as GReaT (Borisov et al., 2022) and HARMONIC (Wang et al., 2024b), treat structured data as natural language, enabling zero-shot transfer and broad domain coverage through autoregressive decoding. However, these methods lack direct control over marginal and joint distributions, are sample inefficient, and struggle to scale with large or heterogeneous datasets.

**LLMs as Data Generators** LLMs have shown exceptional versatility as data generators across various domains. They have been used to augment data (Ding et al., 2024), create instruction-finetuning datasets (Wang et al., 2022; Li et al., 2024), generate tabular data (Wang et al., 2024b; Borisov et al., 2022), synthesize executable code (Nijkamp et al., 2022; Mankowitz et al., 2023), and produce personal mobility data aligned with user preferences (Tang et al., 2024) and routines (Jiawei et al., 2024). LLMs also generate question-answering pairs to enhance model robustness (Chowdhury & Chadha, 2023) and privacy-preserving text via topic modeling (Tan et al., 2025). While existing methods excel at ensuring the semantic or functional quality of individual records, they often fail to control global statistical properties of the dataset. Some works introduce limited control, such as ensuring topic completeness (Zhang et al., 2023b) or data coverage (Chen et al., 2025). In parallel, recent work studies LLMs as attributed training-data generators and behavioral simulators, focusing on diversity, bias, and persona alignment rather than explicit aggregate control (Yu et al., 2023; Choi & Li, 2024; Bui et al., 2025; Holt et al., 2025). However, these approaches lack explicit macroscopic statistical control. Most methods generate data record by record or in small, independent batches, without enforcing global statistical properties, which highlights the gap between generating realistic individual records and ensuring statistical fidelity across the entire dataset. This challenge is directly addressed by LLMSYNTHOR.

## 3 METHODOLOGY

**Definition 1** (Micro-record and Dataset). *Let $\mathcal{V}$ be a predefined set of variables. A **micro-record** $x$ is defined as a set of variable-value pairs: $x = \{(v_i, a_i)\}_{i=1}^{d_x}$, where $v_i \in \mathcal{V}$ and $a_i$ is its corresponding value. A **dataset** is defined as a collection of such records, denoted by $\mathcal{D} = \{x_j\}_{j=1}^{|\mathcal{D}|}$.*

**Definition 2** (Macro-statistics). *Let $\mathcal{D}$ be a dataset of micro-records. We define an operator $\Phi$ that maps the dataset into a set of macro-statistics $\mathcal{S}$, denoted as: $\Phi : \mathcal{D} \mapsto \mathcal{S}$, where $\mathcal{S} = \{\phi_i\}_{i=1}^{|\mathcal{S}|}$. Here, each element $\phi_i$ represents a specific aggregated statistic derived from $\mathcal{D}$.*

**Problem 1** (Macro-aligned Micro-records Synthesis). *Given a set of target macro-statistics $\mathcal{S}_{\text{target}}$, the objective is to construct a dataset of $n$ realistic micro-records, $\mathcal{D}_{\text{synth}} = \{\hat{x}_j\}_{j=1}^{n}$, such that the macro-statistics induced from $\mathcal{D}_{\text{synth}}$ closely align with $\mathcal{S}_{\text{target}}$. Formally, we seek $\min_{\mathcal{D}_{\text{synth}}} Q(\Phi(\mathcal{D}_{\text{synth}}), \mathcal{S}_{\text{target}})$, where $\Phi$ is the aggregation operator defined in Definition 2, and $Q$ is a suitable discrepancy measure in the macro-statistics space.*

In this work, a micro-record represents a data record for an individual entity. Note that the number of variables $d_x$ is allowed to vary across records to accommodate the flexibility of unstructured data. Macro-statistics capture the distributional patterns of a dataset and serve as the target that synthetic micro-records should align with. They are often the only information available from external sources. Specifically, the statistics in $\mathcal{S}$ can be classified as: (1) Marginal statistics: describe the distribution of a single variable (e.g., a frequency vector of education levels); (2) Joint statistics:

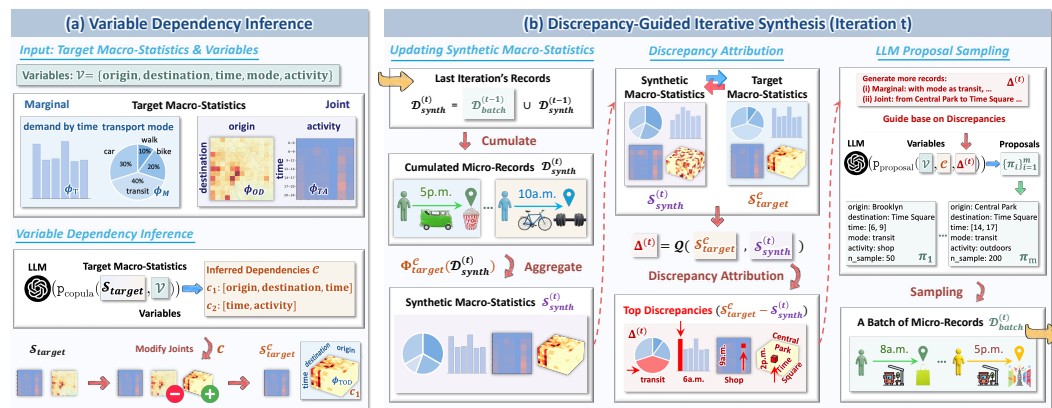

Figure 2: Overview of LLMSYNTHOR. The system (a) first uses an LLM to infer key variable dependencies, and then modifies the target macro-statistics to align with these inferred dependencies; (b) it then performs discrepancy-guided iterative synthesis. In each iteration, the newly generated micro-records are aggregated to form updated synthetic macro-statistics, which are compared with the target. The differences are attributed back to specific variable groups, and these discrepancies guide the LLM to propose new micro-records that correct discrepancies. This feedback loop gradually aligns the synthetic data with the target macro-statistics.

describe dependencies among combinations of variables (e.g., a contingency table of education by employment status). While $\Phi$ can theoretically compute arbitrary statistics, in practice its instantiation is application-specific, determined by the available target macro-statistics in a given context.

Take a mobility dataset as an example. A micro-record can represent a single trip, such as $x = \{(\texttt{origin}, \text{'Times Square'}), (\texttt{destination}, \text{'Central Park'}), (\texttt{mode}, \text{'Bike'}), (\texttt{time}, 17)\}$. Here, the variable set is $\mathcal{V} = \{\texttt{origin}, \texttt{destination}, \texttt{mode}, \texttt{time}\}$, and each micro-record assigns specific values to these variables. Alternatively, an *unstructured* micro-record could represent an entire daily travel diary, in which the number of trips varies across individuals.

In practice, only macro-statistics are available from external sources, denoted $\mathcal{S}_{\text{target}}^{\text{mob}}$. For example, this set may include *marginal statistics*, such as the frequency vector of transport modes chosen by travelers, and *joint statistics*, such as the contingency table of origin and destination zones. The aggregation operator $\Phi_{\text{target}}^{\text{mob}}$, defined in Definition 2, is then instantiated for the mobility context so that $\Phi_{\text{target}}^{\text{mob}}(\mathcal{D}_{\text{synth}}^{\text{mob}})$ produces those statistics that are directly comparable to $\mathcal{S}_{\text{target}}^{\text{mob}}$. The task is therefore to construct a dataset of micro-records $\mathcal{D}_{\text{synth}}^{\text{mob}}$ such that $\Phi(\mathcal{D}_{\text{synth}}^{\text{mob}})$ closely matches $\mathcal{S}_{\text{target}}^{\text{mob}}$. For example, if the target specifies 30% of trips by bike (marginal) and 10% from Times Square to Central Park (joint), the synthetic dataset must match these proportions when aggregated by $\Phi_{\text{target}}^{\text{mob}}$.

## 3.1 OVERVIEW

LLMSYNTHOR consists of two key components: (i) *Variable Dependency Inference*, where the LLM acts as a copula-like mechanism to capture dependencies among variables and refine the set of target joint macro-statistics, and (ii) *Discrepancy-Guided Iterative Synthesis*, where at each iteration $t$, the LLM generates a batch of micro-records to reduce the discrepancy $Q$. An overview of the framework is in Figure 2, and a simplified example that illustrates the process is in Figure 20 in Appendix F.

## 3.2 VARIABLE DEPENDENCY INFERENCE

LLMSYNTHOR operates solely on macro-statistics, which provide complementary information, but some joint statistics can be redundant, while others may be missing. However, marginal statistics are essential as they describe the basic distributions of variables. Therefore, dependency inference focuses on identifying which *variable combinations should be treated as joint dependencies*, while retaining all available marginals. This ensures the synthesis is based on the most informative dependencies.

We draw inspiration from copula theory (Sklar, 1959), which represents a joint distribution as marginals plus a dependency structure. We use a pre-trained LLM as a nonparametric copula to infer

variable subsets $\mathcal{C} = \{c_k\}_{k=1}^{|\mathcal{C}|}$, $c_k \subseteq \mathcal{V}$, where each subset $c_k$ is expected to exhibit strong statistical dependence. The LLM infers these dependencies by leveraging (i) the semantic meaning of variable names (e.g., "education" and "income" are related), and (ii) the available macro-statistics $\mathcal{S}_{\text{target}}$:

$$\mathcal{C} \sim \texttt{LLM}\big(\texttt{p}_{\text{copula}}(\mathcal{S}_{\text{target}}, \mathcal{V})\big), \tag{1}$$

where the prompt $\texttt{p}_{\text{copula}}$ asks for informative variable combinations (as provided in Appendix E.2). For example, for a variable set $\mathcal{V} = \{\texttt{origin}, \texttt{destination}, \texttt{mode}, \texttt{time}\}$, the LLM might infer that the variable subsets $\mathcal{C}$ exhibit strong dependencies: $\{c_1 = [\texttt{origin}, \texttt{destination}, \texttt{time}], c_2 = [\texttt{time}, \texttt{activity}]\}$.

Next, we retain all marginals and expand or filter the joint statistics in $\mathcal{S}_{\text{target}}$ based on $\mathcal{C}$, yielding $\mathcal{S}_{\text{target}}^{\mathcal{C}}$. If an inferred dependency lacks corresponding data, practitioners are encouraged to collect additional statistics, if unavailable, the LLM can approximate the joint distribution using existing macro-statistics. The aggregation operator is then updated as $\Phi_{\text{target}} \mapsto \Phi_{\text{target}}^{\mathcal{C}}$, ensuring that the induced statistics from any dataset are directly comparable to $\mathcal{S}_{\text{target}}^{\mathcal{C}}$.

For example, in Figure 2(a), the LLM identifies the dependency $c_1$ between the variables $[\texttt{origin}, \texttt{destination}, \texttt{time}]$, replacing the original joint macro-statistic $\phi_{\text{OD}}$ with a new one, $\phi_{\text{TOD}}$, which captures the time-dependent OD demand. In contrast, the dependency $c_2$ between $[\texttt{time}, \texttt{activity}]$ remains unchanged, and the corresponding macro-statistic $\phi_{\text{TA}}$ is preserved.

## 3.3 DISCREPANCY-GUIDED ITERATIVE SYNTHESIS

With the updated target macro-statistics $\mathcal{S}_{target}^{\mathcal{C}}$ and operator $\Phi_{\text{target}}^{\mathcal{C}}$ in place, LLMSYNTHOR then enters an iterative synthesis loop that progressively builds the synthetic dataset $\mathcal{D}_{\text{synth}}$. At iteration $t$, three steps are performed: *(i)* updates the macro-statistics for the *cumulative* dataset via $\mathcal{S}_{\text{synth}}^{(t)} = \Phi_{\text{target}}^{\mathcal{C}}(\mathcal{D}_{\text{synth}}^{(t)})$; *(ii)* compute discrepancy signals $\Delta^{(t)} = Q\big(\mathcal{S}_{\text{target}}^{\mathcal{C}}, \mathcal{S}_{\text{synth}}^{(t)}\big)$; *(iii)* under the guidance of discrepancy signals, generate a *batch* of new records $\mathcal{D}_{\text{batch}}^{(t)}$ to reduce $\Delta^{(t)}$, and update the cumulative set $\mathcal{D}_{\text{synth}}^{(t+1)} = \mathcal{D}_{\text{synth}}^{(t)} \cup \mathcal{D}_{\text{batch}}^{(t)}$. We now elaborate on each step.

**1. Updating Synthetic Macro-statistics.** At the beginning of iteration $t$, the synthetic dataset $\mathcal{D}_{\text{synth}}^{(t)}$ is aggregated into macro-statistics aligned with the target $\mathcal{S}_{\text{synth}}^{(t)} = \Phi_{\text{target}}^{\mathcal{C}}\left(\mathcal{D}_{\text{synth}}^{(t)}\right)$. To ensure comparability across heterogeneous variables, each record $x \in \mathcal{D}_{\text{synth}}^{(t)}$ is mapped into a unified discretized space. Discrete variables are represented by frequency vectors, while continuous variables are discretized into bins (e.g., quantile-based or fixed intervals) that follow the same scheme as available in the target macro-statistics. When multiple variables are involved, the corresponding bins define a contingency table that is directly comparable to the observed target.

**2. Discrepancy Attribution.** Given the updated synthetic macro-statistics $\mathcal{S}_{\text{synth}}^{(t)}$ and the target macro-statistics $\mathcal{S}_{\text{target}}^{\mathcal{C}}$, the next step is to measure their discrepancy. Formally:

$$\Delta^{(t)} = \left\{\delta_1^{(t)}, \delta_2^{(t)}, \dots, \delta_{|\mathcal{S}_{target}^{\mathcal{C}}|}^{(t)}\right\} = Q\big(\mathcal{S}_{\text{target}}^{\mathcal{C}}, \mathcal{S}_{\text{synth}}^{(t)}\big), \tag{2}$$

where $Q(\cdot, \cdot)$ is a discrepancy measure. At each iteration, we generate a batch of micro-records to expand the synthetic dataset. To guide this process, we implement $Q$ as the directed difference $\mathcal{S}_{\text{target}}^{\mathcal{C}} - \mathcal{S}_{\text{synth}}^{(t)}$, which captures the frequency distribution differences of each macro-statistic. The positive and negative values indicate underrepresented and overrepresented portions in the synthetic data compared to the target, respectively. By focusing on large positive discrepancies, we generate additional data to address these gaps, gradually reducing the discrepancy over time. Since $\Phi(\cdot, \cdot)$ operates on a discretized representation of $\mathcal{D}_{\text{synth}}^{(t)}$, the resulting macro-statistics are tied to specific bins or discrete values (e.g., "time = 5-6 p.m. and mode = bike"), making discrepancies interpretable and *attributable* to particular subsets of $\mathcal{V}$. This enables the formulation of actionable LLM prompts.

For example, consider the directed frequency difference $(\mathcal{S}_{\text{target}}^{\mathcal{C}} - \mathcal{S}_{\text{synth}}^{(t)})$ for transport modes: $\delta_M^{(t)} = \{\text{bike} : +30\%, \text{car} : -20\%\}$. This difference indicates that the target distribution has a 30% higher proportion of bike trips compared to the synthetic distribution, so we can prompt the LLM

with <generate more bike trips>, gradually reducing the gap and aligning the synthetic distribution closer to the target. An illustrative example of this process is shown in Figure 21. Theorem 1 further proves that this approach converges over time. Additional details on the discrepancy measure can be found in Appendix B.1.

**3. Discrepancy-Guided LLM Proposal Sampling.** The discrepancy signals $\Delta^{(t)}$ are then used to guide the generation of new records. Rather than producing one micro-record per LLM request, which is inefficient and hard to align with the target macro-statistics, the LLM is shifted from a direct generator to a *planner*. We introduce LLM Proposal Sampling, where the LLM outputs $m$ proposals:

$$\{\pi_1, \ldots, \pi_m\} \ \sim \ \texttt{LLM}\big(\texttt{p}_{\text{proposal}}(\mathcal{V}, \mathcal{C}, \Delta^{(t)})\big), \tag{3}$$

given a prompt $\texttt{p}_{\text{proposal}}$ that instructs it to design proposals aimed at reducing the discrepancies in $\Delta^{(t)}$ while respecting the dependency structure $\mathcal{C}$. The implemented prompt is provided in Appendix E. Each proposal $\pi_i$ defines specific values or ranges for all variables and specifying the number of records to generate. For discrete attributes, this corresponds to fixing categories (e.g., $\texttt{mode} = \texttt{bike}$); for continuous ones such as $\texttt{time}$, it specifies ranges (e.g., $\texttt{time} \in [17, 20]$). In addition, the LLM also plans the number of records to be drawn from each proposal, so that generation is not only localized but also quantitatively guided. Thus, every $\pi_i$ is interpretable, directly sampleable, and aligned with the discretization scheme used in $\Phi^{\mathcal{C}}_{\text{target}}$, while also grounded in the LLM's priors to ensure that the proposed joint configurations remain realistic. An example is shown in the right panel of Figure 2(b), where the LLM generates multiple proposals for record creation. The first proposal $\pi_1$ specifies 50 records with origin "Brooklyn", destination "Time Square", mode "transit", activity "Shop", and time range [6, 9]. This proposal defines a targeted joint distribution over variables, addressing identified discrepancies and guiding the efficient generation.

From these proposals, a batch of synthetic records $\mathcal{D}^{(t)}_{\text{batch}}$ is sampled and merged into the cumulative dataset: $\mathcal{D}^{(t+1)}_{\text{synth}} = \mathcal{D}^{(t)}_{\text{synth}} \cup \mathcal{D}^{(t)}_{\text{batch}}$. By operating at the level of proposals rather than individual records, LLMSYNTHOR functions as an efficient high-level distributional controller: it bridges macro-level discrepancy signals and micro-level record generation, efficiently reducing targeted mismatches while retaining the realism induced by the LLM's priors.

**Iterative Synthesis.** We summarize the iterative synthesis process in Algorithm 1. At each iteration $t$, three steps are performed: *First*, the cumulative synthetic dataset $\mathcal{D}^{(t)}_{\text{synth}}$ is aggregated into macro-statistics $\mathcal{S}^{(t)}_{\text{synth}}$ via the updated operator $\Phi^{\mathcal{C}}_{\text{target}}$, and compared with the target $\mathcal{S}^{\mathcal{C}}_{\text{target}}$ using the discrepancy measure $Q$, yielding discrepancy signals $\Delta^{(t)}$. *Second*, these signals are provided to the LLM, which generates proposal distributions $\{\pi^{(t)}_i\}$ through LLM Proposal Sampling, each specifying a localized distribution over joint variable configurations aimed at reducing the dominant mismatches. *Third*, records sampled from these proposals form a batch $\mathcal{D}^{(t)}_{\text{batch}}$, which is merged into the cumulative dataset to obtain $\mathcal{D}^{(t)}_{\text{synth}}$. This loop progressively reduces the discrepancies, refining $\mathcal{D}_{\text{synth}}$ until it closely aligns the target macro-statistics.

---

**Algorithm 1** Iterative Synthesis

**Require:** Target $\mathcal{S}_{\text{target}}$, operator $\Phi_{\text{target}}$, discrepancy $Q$, iterations $T$, $\mathcal{D}^{(0)}_{\text{synth}} \leftarrow \emptyset$
1: $\mathcal{C} \leftarrow \texttt{LLM}\big(\texttt{p}_{\text{copula}}(\mathcal{S}_{\text{target}}, \mathcal{V})\big)$
2: $\mathcal{S}^{\mathcal{C}}_{\text{target}} \xleftarrow{\mathcal{C}} \mathcal{S}_{\text{target}}$; $\Phi^{\mathcal{C}}_{\text{target}} \xleftarrow{\mathcal{C}} \Phi_{\text{target}}$
3: **for** $t = 1$ **to** $T$ **do**
4: $\quad \mathcal{S}^{(t)}_{\text{synth}} \leftarrow \Phi^{\mathcal{C}}_{\text{target}}(\mathcal{D}^{(t)}_{\text{synth}})$
5: $\quad \Delta^{(t)} \leftarrow Q\big(\mathcal{S}^{\mathcal{C}}_{\text{target}}, \mathcal{S}^{(t)}_{\text{synth}}\big)$
6: $\quad \{\pi^{(t)}_i\} \leftarrow \texttt{LLM}\big(\texttt{p}_{\text{proposal}}(\mathcal{V}, \mathcal{C}, \Delta^{(t)})\big)$
7: $\quad \mathcal{D}^{(t)}_{\text{batch}} \leftarrow \bigcup_i \{\hat{x}^{(t)}_i\}, \ \hat{x}^{(t)}_i \sim \pi^{(t)}_i$
8: $\quad \mathcal{D}^{(t+1)}_{\text{synth}} \leftarrow \mathcal{D}^{(t)}_{\text{synth}} \cup \mathcal{D}^{(t)}_{\text{batch}}$
9: **end for**
10: **return** $\mathcal{D}^{(T)}_{\text{synth}}$

---

## 4 EXPERIMENTS

To comprehensively evaluate LLMSYNTHOR, our experiments are structured to answer three research questions (RQs). *RQ1*: How effectively can LLMSYNTHOR synthesize realistic and usable micro-records from limited aggregate-level macro-statistics? *RQ2*: How does LLMSYNTHOR's statistical fidelity compare to state-of-the-art models trained on full micro-record datasets? *RQ3*: How *realistic* can LLMSYNTHOR effectively synthesize *unstructured* micro-records, without task-specific manual engineering? We address these questions using three practical tasks detailed below. Unless otherwise stated, we perform experiments using the Chat Completion mode of GPT-4.1-nano (Openai, 2025).

## 4.1 MOBILITY SYNTHESIS

Mobility synthesis aims to generate a complete dataset of realistic micro-records, each detailing an individual's time-stamped origin-destination trip, activity, and transport mode. This is an essential capability for urban applications like transport planning and event simulation, as comprehensive, individual-level mobility data for an entire population is impractical to collect.

**Task Setup.** To answer *RQ1*, we design a mobility synthesis task that mirrors a common real-world constraint: fusing aggregate information from *multiple* complementary data sources. From OpenPFLOW (Kashiyama et al., 2017), we extract trips (origin, destination, timestamp) and assign transport modes. Since OpenPFLOW lacks activity labels, we incorporate time-activity patterns from LLMob (Jiawei et al., 2024). This task tests the ability to align spatiotemporal and behavioral data by generating 30,000 trips in a day in Tokyo to match both macro-statistics. As existing methods cannot handle such *Mixed-Source* synthesis without manual adaptations, we focus on a qualitative assessment of LLMSYNTHOR's unique capabilities. Further details are provided in Appendix A.3.

**Results.** The results provide strong evidence for the first component of *RQ1*: LLMSYNTHOR's ability to generate realistic micro-records. Figure 3 compares the synthetic data against the target macro-statistics. The time-activity heatmaps on the left show close alignment, accurately capturing commuting peaks and midday activity rises. The OD flow heatmaps during the morning peak confirm that the synthetic trips reproduce key spatial patterns, matching high-density areas. These findings demonstrate that the synthesized population, in aggregate, successfully reproduces the guiding macro-level patterns. Furthermore, as shown in the Appendix C.3 (Figures 18 and 19), the generated micro-records also exhibit realistic internal structures, such as realistic correlations between travel mode and distance, reflecting their micro-level realism.

**Controllable Mobility Synthesis for Events Simulation** To address the second component of *RQ1* concerning the utility of the generated data, we demonstrate a key advantage of our framework: the ability to effortlessly incorporate arbitrary context into the synthesis process. We test LLMSYN-THOR's controllability in a "what-if" scenario. We simulate a concert at Tokyo Dome (20-24h) by simply adding the prompt < There will be a concert from 20-24 at Tokyo Dome > during proposal generation. As shown in Figure 3, this simple intervention causes LLMSYNTHOR to generate a surge of trips to the event location while preserving realistic background flows. This demonstrates LLMSYN-THOR 's potential as a powerful tool for scenario planning, allowing policymakers to simulate the effects of large events using detailed synthetic micro-records.

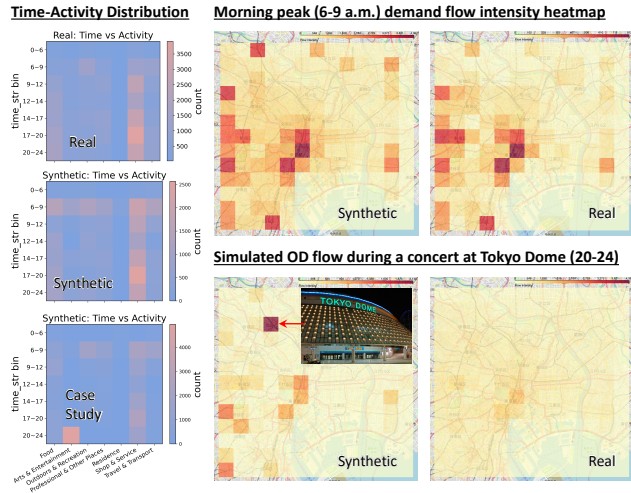

Figure 3: Real vs. synthetic mobility patterns.

## 4.2 E-COMMERCE TRANSACTION SYNTHESIS

Having demonstrated LLMSYNTHOR's unique capabilities, we now rigorously assess its statistical fidelity against strong, established baselines to answer *RQ2*. We use a controlled e-commerce synthesis task, providing a comprehensive comparison with state-of-the-art *Tabular* synthesis models. This experiment critically tests whether LLMSYNTHOR, guided only by macro-statistics, can match or exceed the performance of models trained on the full set of micro-records.

**Task Setup.** We simulate a controlled environment where each transaction is sampled from a closed-form Bayesian network over six variables: $\{v_A, v_G, v_L, v_C, v_X, v_M\}$, representing `user_age`, `gender`, `location_tier`, `product_category`, `price`, and `payment_method`. The generative process follows a structured probabilistic graphical model with a known joint distribution: $p(v_A, v_G, v_L, v_C, v_X, v_M) = p(v_A)\,p(v_G)\,p(v_L)\,p(v_C|v_A, v_G)\,p(v_X|v_C)\,p(v_M|v_L)$. This con-

| Methods | Rej.% ↓ | Was ↓ | Gap ↓ | Tvd ↓ | Gap ↓ | Tvd ↓ | Gap ↓ | Tvd ↓ | Gap ↓ | Was ↓ | Gap ↓ | Tvd ↓ | Gap ↓ |
|---|---|---|---|---|---|---|---|---|---|---|---|---|---|
| | | age | | gender | | location | | category | | price | | payment | |
| TVAE | 1.7 ± 0.2 | 2.06 | 0.032 | 0.008 | 0.01 | 0.056 | 0.043 | 0.054 | 0.02 | 113.194 | 0.085 | 0.017 | 0.013 |
| CTGAN | 5.1 ± 1.1 | 4.429 | 0.057 | 0.117 | 0.065 | 0.162 | 0.076 | 0.080 | 0.028 | 138.998 | 0.059 | 0.088 | 0.022 |
| CopulaGAN | 4.8 ± 1.0 | 4.82 | 0.027 | 0.052 | 0.016 | 0.045 | 0.031 | 0.057 | 0.024 | 151.239 | 0.047 | 0.045 | 0.014 |
| GReaT | 0.8 ± 0.1 | 2.862 | 0.052 | 0.016 | 0.009 | 0.039 | 0.020 | 0.045 | 0.027 | 169.866 | 0.104 | 0.009 | 0.012 |
| TabSyn | 1.9 ± 0.3 | 1.196 | **0.012** | 0.012 | 0.022 | 0.007 | 0.022 | 0.045 | **0.01** | 114.12 | 0.067 | 0.028 | 0.005 |
| Spada | 4.1 ± 0.9 | 1.42 | 0.025 | 0.013 | 0.019 | **0.003** | 0.022 | 0.016 | 0.024 | 54.603 | 0.023 | 0.025 | 0.009 |
| IPF | 3.9 ± 0.6 | 5.27 | 0.087 | 0.018 | **0.008** | 0.026 | 0.016 | 0.012 | 0.024 | 139.957 | 0.078 | 0.035 | 0.013 |
| LLM-ICL | **0.1 ± 0.0** | 20.61 | 0.194 | 0.026 | 0.026 | 0.024 | 0.014 | 0.441 | 0.215 | 279.743 | 0.205 | 0.163 | 0.081 |
| Ours | 0.3 ± 0.1 | **1.13** | 0.023 | **0.002** | 0.008 | **0.002** | 0.012 | **0.010** | 0.022 | **12.762** | **0.011** | **0.003** | **0.004** |

Table 1: Marginal alignment evaluation (↓ is better). Methods differ by supervision: IPF, LLM-ICL, and Ours (LLMSYNTHOR) are purely *macro-supervised*; Spada is *macro+micro supervised*; while others are standard *micro-supervised* models. Metrics include Wasserstein distance (Was), Total Variation Distance (Tvd), and the classifier two-sample test gap (Gap, defined as $|Acc - 0.5|$).

| Methods | Jsd ↓ | Gap ↓ | Jsd ↓ | Gap ↓ | Jsd ↓ | Gap ↓ |
|---|---|---|---|---|---|---|
| | $[v_A, v_G, v_C]$ | | $[v_C, v_X]$ | | $[v_L, v_M]$ | |
| TVAE | 0.23 | 0.074 | 0.245 | 0.106 | 0.185 | 0.051 |
| CTGAN | 0.133 | 0.055 | 0.298 | 0.098 | 0.145 | 0.076 |
| CopulaGAN | 0.133 | 0.057 | 0.280 | 0.102 | 0.069 | 0.018 |
| GReaT | 0.087 | 0.058 | 0.382 | 0.177 | 0.038 | 0.020 |
| TabSyn | 0.083 | **0.022** | 0.237 | 0.082 | 0.027 | 0.015 |
| Spada | 0.085 | 0.024 | 0.225 | 0.033 | 0.103 | 0.03 |
| IPF | 0.117 | 0.049 | 0.504 | 0.154 | 0.113 | 0.025 |
| LLM-ICL | 0.437 | 0.216 | 0.591 | 0.242 | 0.476 | 0.081 |
| Ours | **0.071** | **0.022** | **0.134** | **0.020** | **0.007** | **0.007** |

Table 2: Joint evaluations.

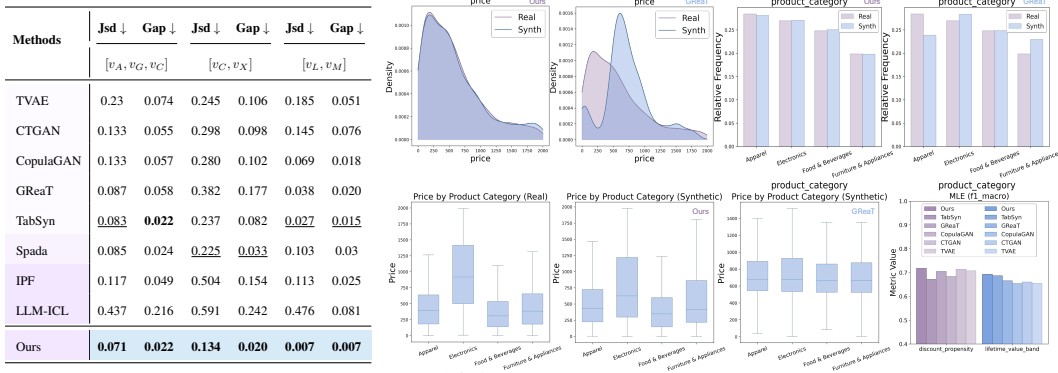

Figure 4: Qualitative Distributions and Comparisons.

trolled setting enables a precise and rigorous evaluation. We generate a 2,000-record reference dataset to serve as the ground truth. To ensure a fair and direct comparison against baselines, we compute a set of target macro-statistics directly from this reference dataset, following the detailed procedure described in Appendix A.1. These macro-statistics serve as the sole input for LLMSYNTHOR. In contrast, all baseline models are trained on the full 2,000 individual records. This setup allows us to compare the statistical fidelity of different data synthesis methods and reflects how well the joint dependencies inferred by LLMSYNTHOR's internal logic match the real underlying structure.

**Baselines.** Given the structured tabular nature of this task, we compare LLMSYNTHOR to representative baselines across major generative paradigms that are trained on the full record-level data: (1) TVAE and CTGAN (VAE- and GAN-based); (2) CopulaGAN (GAN with copula modeling); (3) GReaT (autoregressive transformer); (4) TabSyn (diffusion-based); (5) IPF (Iterative Proportional Fitting); (6) Spada (LLM-induced dependency graph with normalizing flows); and (7) LLM-ICL (In-context Learning). For a fair comparison of output quality, we apply rejection sampling to ensure record realism. Detailed baseline information is provided in Appendix B.2.1.

**Results.** The results clearly answer *RQ2* in the affirmative. Despite being restricted to macro-statistics, LLMSYNTHOR consistently outperforms baselines that were trained on the full micro-record dataset across measures of statistical fidelity and downstream utility. We evaluate the synthetic data from two perspectives. First, to assess statistical fidelity, Tables 1 and 2 report key metrics for both marginal and joint distributions. Across all measures, including Wasserstein distance (W), Total Variation Distance (TVD), and the Classifier Two-Sample Test (C2ST) Gap ($|acc - 0.5|$), Jensen-Shannon Divergence (JSD), LLMSYNTHOR achieves the lowest divergence scores. The visualizations in Figure 4 further confirm that LLMSYNTHOR's generated distributions most closely match the ground truth, demonstrating its superior ability to preserve the specified dependency structures. Second, we evaluate the practical downstream utility of the synthetic data. We introduce two derived variables grounded in economic theory: discount_propensity and lifetime_value_band (see Appendix A.1 for definitions). We then train standard classifiers (logistic regression, decision trees, and random forests) on the data from each synthesizer.

As shown in Figure 4, models trained on LLM-SYNTHOR's data generalize best to real data, proving that its high statistical fidelity translates directly to superior performance on practical tasks. We further validate the semantic realism of the generated records. As detailed in Appendix C.2.3, LLMSYNTHOR achieves a near-zero rejection rate (Table 10), demonstrating that LLMSYNTHOR, by leveraging the inherent world knowledge of LLMs, effectively avoids implausible attribute combinations often observed in baseline methods. While LLM-ICL yields a marginally lower rejection rate, confirming the inherent semantic strength of LLM backbones, its poor distributional fit highlights the critical value of our discrepancy-guided synthesis in bridging semantic realism with statistical fidelity. Finally, as shown in Figure 5, LLM-SYNTHOR maintains high diversity while progressively mitigating bias. Efficiency analysis further demonstrates the framework's flexibility: with only 10 iterations, LLMSYNTHOR outperforms GReaT in both speed and quality. Increas-

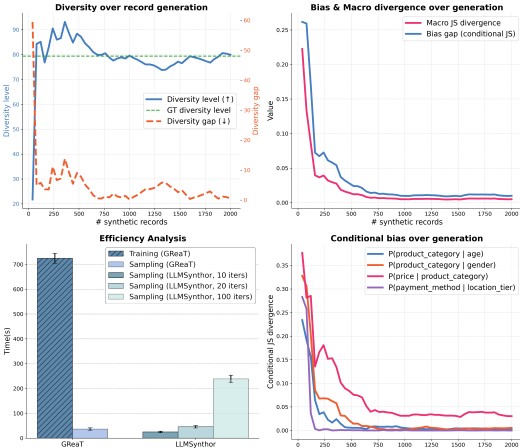

Figure 5: Diversity, macro-level alignment, and conditional bias all rapidly converge toward ground truth as more synthetic records are generated, while LLMSynthor also shows flexible efficiency, outperforming GReaT at low iteration counts and achieving highest fidelity with more iterations.

ing iterations further enhances fidelity, offering a controllable trade-off between computational cost and performance. Detailed results are provided in Appendix C.1. More analyses, including tracking the *convergence* of distributional divergence over iterations, *ablation* studies confirming the effectiveness of *Variable Dependency Inference*, and *robustness* checks demonstrating stable performance when using smaller, open-source models (Qwen2.5-7B and 3B), are in Appendix C.1.

### 4.3 POPULATION SYNTHESIS

After establishing LLMSYNTHOR's quantitative superiority on structured data, we address *RQ3* by evaluating its ability to generalize to complex, unstructured data formats. We tackle a real-world population synthesis task, a domain where data is inherently *Unstructured* due to varying household size. This experiment tests whether LLMSYNTHOR's general-purpose framework can outperform specialized models in a setting that typically requires significant, task-specific manual engineering.

**Task Setup.** We use population microdata from the American Community Survey (ACS) for households in South Carolina. This dataset is a prime example of the challenge in *RQ3*, as it includes both household- and person-level attributes, resulting in unstructured records due to varying household sizes. After preprocessing, we obtain a dataset of about 15,000 households. The task is to generate a synthetic population that preserves the complex joint distributions across all demographic and household features. To assess real-world utility, we define 16 policy-relevant queries (e.g., proportion of multigenerational households) across six categories, which serve as a practical proxy for statistical fidelity. Full data and query details are provided in Appendix A.2.

**Baselines.** We compare LLMSYNTHOR against a range of strong baselines specifically designed for population synthesis. These include (1) CP (a tensor factorization method), (2) HMM (a hierarchical probabilistic model), and (3) NVI (a deep variational framework). These baselines represent diverse and powerful specialized approaches, providing a rigorous benchmark for evaluating the performance of our general-purpose framework on macro-aware synthesis.

**Results.** The results provide a resounding affirmative answer to *RQ3*. As shown in Table 3, LLMSYNTHOR achieves the lowest mean relative error (MRE) on policy-relevant queries in every single category, often by a large margin. For instance, on equity-related queries, the error is reduced from 4.23 (HMM) to just 0.25. Similar significant gains are observed for demographics, employment, mobility, and vulnerability metrics. Although LLMSYNTHOR does not win on every individual query, its consistent superiority at the category level confirms that it more accurately captures the complex

joint dependencies inherent in the population data, leading to synthetic populations with far greater practical utility. Beyond statistical utility, Table 9 reveals a critical advantage in semantic realism. Unlike transaction data where outliers are merely improbable, population data is governed by strict logical constraints (e.g., a child cannot be a householder). Traditional probabilistic baselines fail to capture these rigid structural dependencies, resulting in rejection rates as high as 96.8% (NVI) and 73.9% (CP). In contrast, LLMSYNTHOR drastically reduces the overall rejection rate to 13.3%. By leveraging the semantic reasoning capabilities of LLMs, our method intuitively respects complex intra-household rules, such as ensuring valid householder assignments and age-appropriate education levels, without requiring explicit constraint programming. Details of these validity checks and breakdowns are provided in Appendix C.2.

| Methods | Rej. Rate% | Demog. | Employment | Equity | Household | Mobility | Vuln. |
|---|---|---|---|---|---|---|---|
| CP | 73.9 | 0.54 | 1.02 | 5.79 | 2.34 | 1.47 | 0.86 |
| HMM | 57.8 | 0.56 | 0.32 | 4.23 | 2.01 | 0.48 | 0.91 |
| NVI | 96.8 | 0.53 | 0.27 | 5.49 | 2.06 | **0.24** | 1.06 |
| Ours | **13.3** | **0.21** | **0.2** | **0.25** | **0.13** | 0.35 | **0.37** |

Table 3: Rejection rate and category-wise MRE across queries.

| Check | NVI | CP | HMM | Real | Ours |
|---|---|---|---|---|---|
| *Overall rejection rates* ↓ | | | | | |
| Household rejection rate | $96.8 \pm 0.3$ | $73.9 \pm 0.6$ | $57.8 \pm 0.5$ | $0.0 \pm 0.0$ | $\mathbf{13.3 \pm 0.4}$ |
| Person rejection rate | $45.9 \pm 0.4$ | $34.1 \pm 0.7$ | $27.8 \pm 0.6$ | $0.0 \pm 0.0$ | $\mathbf{6.3 \pm 0.2}$ |
| *Household-level checks* ↓ | | | | | |
| householder_type_adult_consistency | $4.5 \pm 0.2$ | $4.4 \pm 0.3$ | $15.2 \pm 0.5$ | $0.0 \pm 0.0$ | $\mathbf{3.8 \pm 0.2}$ |
| persons_all_valid | $96.7 \pm 0.4$ | $73.5 \pm 0.6$ | $56.1 \pm 0.7$ | $0.0 \pm 0.0$ | $\mathbf{10.7 \pm 0.3}$ |
| *Person-level checks* ↓ | | | | | |
| age_range | $0.1 \pm 0.0$ | $0.5 \pm 0.1$ | $0.3 \pm 0.1$ | $0.0 \pm 0.0$ | $\mathbf{0.0 \pm 0.0}$ |
| race_valid | $0.0 \pm 0.0$ | $0.8 \pm 0.1$ | $0.0 \pm 0.0$ | $0.0 \pm 0.0$ | $\mathbf{0.0 \pm 0.0}$ |
| employment_age_consistency | $41.6 \pm 0.7$ | $30.9 \pm 0.6$ | $24.5 \pm 0.5$ | $0.0 \pm 0.0$ | $\mathbf{4.4 \pm 0.2}$ |
| education_age_consistency | $42.7 \pm 0.8$ | $8.1 \pm 0.4$ | $10.7 \pm 0.5$ | $0.0 \pm 0.0$ | $\mathbf{2.2 \pm 0.1}$ |

Table 4: Rejection rates (%) by rule across methods. Checks with zero rejections in all methods are omitted for brevity.

### 4.4 DISCUSSION

The experimental results answer the research questions positively: LLMSYNTHOR can synthesize useful micro-records from multi-source macro-statistics (*RQ1*), outperform baselines in quantitative fidelity with full micro-record access (*RQ2*), and generate realistic micro-records for complex, unstructured data without task-specific engineering (*RQ3*).

Despite these strong results, the framework has some limitations. *First*, LLMs carry inherent priors that can occasionally introduce biases, misaligning with the target macro-statistics. This can be mitigated by using more constrained prompts or removing semantic cues during generation. *Second*, the framework's scalability is limited by the context window and reasoning capabilities of the backbone LLM, especially for high-dimensional datasets with hundreds of variables. However, this limitation is expected to improve with future LLM advancements. *Third*, while LLMSYNTHOR performs well with mixed-type *i.i.d.* data, it is not designed for perceptual or tightly sequential data such as images or raw time series. Nevertheless, it could be used as a high-level controller to guide domain-specific generators for these modalities. Finally, although LLMSYNTHOR does not provide rigorous privacy guarantees, its synthesis process minimizes the risk of direct data re-identification by focusing on aligning with aggregate statistics rather than memorizing individual records.

## 5 CONCLUSION

In this work, we introduced LLMSYNTHOR, a novel framework designed to bridge the critical micro-macro gap in data synthesis. We address the challenge of generating realistic, individual-level micro-records when only aggregate macro-statistics are available. Our central contribution is a paradigm shift: we repurpose a pre-trained LLM from an uncontrolled, record-by-record generator into a macro-aware simulator. This simulator operates within an iterative feedback loop, where it is continuously guided by discrepancies to ensure dataset-level statistical alignment.

Our experiments demonstrated that LLMSYNTHOR successfully synthesizes realistic micro-records with utility from multi-source statistics (*RQ1*), quantitatively outperforms state-of-the-art models with full data access (*RQ2*), and generalizes to realistic, unstructured data where other methods require manual engineering (*RQ3*). By providing a robust and general-purpose solution for generating statistically grounded micro-records, LLMSYNTHOR opens new possibilities for data-driven research, agent-based simulation, and evidence-based policymaking. As language models continue to advance, the principles of statistically-guided synthesis presented here offer a scalable path toward creating reliable, high-fidelity synthetic worlds for a broad range of scientific and societal applications.

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

APPENDIX

The appendix is organized into several sections, each focusing on different aspects of the work. Section A details the datasets used, Section B describes the experimental setups and baselines, and Section C presents supplementary results. Section D provides the theoretical analysis, while Section E outlines the prompts used in our framework. Finally, Section F offers running examples to illustrate the methodology.

**Table of Contents**

**The Use of Large Language Models (LLMs):** The authors used LLMs to enhance clarity and readability, and LLMs are also an integral part of our methodology. We take full responsibility for the content after review.

**Ethics Statement:** This work adheres to the ICLR Code of Ethics. No human subjects were involved, and all datasets used are publicly available with proper consent. We ensure fairness, transparency, and privacy, and have disclosed any conflicts of interest or sponsorships.

**Reproducibility Statement:** We provide a link to an anonymous repository containing the source code, along with supplementary materials. All experimental setups, baseline configurations, data preprocessing steps, and full results are included in the appendix to ensure reproducibility of our results.

# A  DATA

## A.1  E-COMMERCE TRANSACTIONS

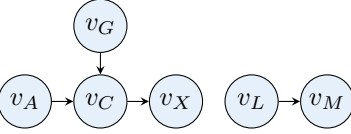

Figure 6: Bayesian network representing the generative process of e-commerce transactions.

**Data Generating Process.**  Let $v_A$, $v_G$, $v_L$, $v_C$, $v_X$, and $v_M$ denote the random variables for user age, gender, location tier, product category, price, and payment method, respectively. We assume the following generative mechanisms:

1. $v_A \sim \left( \sum_{i=1}^{3} \pi_i \mathcal{N}(\mu_i, \phi_i^2) \right) \mathbf{1}_{[18,90]}(v_A)$,  where $\sum_{i=1}^{3} \pi_i = 1, \ \pi_i > 0$,

2. $v_G \sim \text{discrete}\big(p_G^{(\text{male})}, p_G^{(\text{female})}\big)$,  $p_G^{(\text{male})} + p_G^{(\text{female})} = 1$,

3. $v_L \sim \text{discrete}\big(p_L^{(1)}, p_L^{(2)}, p_L^{(3)}\big)$,  $\sum_{k=1}^{3} p_L^{(k)} = 1$,

4. $v_C \mid (v_A, v_G) \sim \text{discrete}\big(p_C^{(g(v_A),\, v_G)}\big)$,  $g(v_A) = \begin{cases} \text{young}, & v_A < 35, \\ \text{middle}, & 35 \le v_A < 55, \\ \text{old}, & v_A \ge 55, \end{cases}$

5. $v_X \mid v_C = c \sim \mathcal{N}(\mu_c, \phi_c^2)\, \mathbf{1}_{[\ell,u]}(v_X)$,  $[\ell, u]$ denotes the valid price range,

6. $v_M \mid v_L = k \sim \text{discrete}\big(p_M^{(k)}\big)$,  $\sum_{j} p_M^{(k)}(j) = 1 \quad \forall k$.

Under this model, the joint distribution factorizes as

$$p(v_A, v_G, v_L, v_C, v_X, v_M) = p(v_A)\, p(v_G)\, p(v_L)\, p(v_C \mid v_A, v_G)\, p(v_X \mid v_C)\, p(v_M \mid v_L),$$

enforcing the exact conditional independencies of the Bayesian network in Figure 6.

**Implementation Settings.**  In our experiments, we instantiate the model with the following parameter values:

- Age ($v_A$): drawn from a mixture of three truncated Gaussian distributions,

$$v_A \sim \sum_{i=1}^{3} \pi_i \mathcal{N}(\mu_i, \phi_i^2)\big|_{[18,90]},$$

  with mixture weights $\pi = (0.5, 0.45, 0.15)$, means $\mu = (24, 38, 55)$, and standard deviations $\phi = (6, 15, 10)$.
- Gender ($v_G$): categorical variable over $\{\text{Male}, \text{Female}\}$ with probabilities
$$p(v_G) = (0.45, 0.55).$$
- Location Tier ($v_L$): categorical variable over $\{\text{Developed}, \text{Developing}\}$ with probabilities
$$p(v_L) = (0.4, 0.6).$$
- Product Category ($v_C$): categorical variable over $\{\text{Electronics}, \text{Apparel}, \text{Food \& Beverages}, \text{Furniture \& Appliances}\}$. Conditional probabilities $p(v_C \mid g, v_G)$ are defined for age group $g \in \{\text{young}, \text{middle}, \text{old}\}$ and gender $v_G$:

Table 5: Conditional distributions $p(v_C \mid g, v_G)$

| Age Group | Gender | Electronics | Apparel | Food & Beverages | Furniture & Appliances |
|-----------|--------|-------------|---------|------------------|------------------------|
| Young | Male | 0.50 | 0.25 | 0.05 | 0.20 |
| Young | Female | 0.20 | 0.50 | 0.05 | 0.25 |
| Middle | Male | 0.20 | 0.10 | 0.50 | 0.20 |
| Middle | Female | 0.10 | 0.20 | 0.55 | 0.15 |
| Old | Male | 0.10 | 0.10 | 0.65 | 0.15 |
| Old | Female | 0.10 | 0.10 | 0.60 | 0.20 |

- Price ($v_X$): conditional on category $c$, drawn from a truncated Gaussian

$$v_X \sim \mathcal{N}(\mu_c, \phi_c^2)\big|_{[0,2000]},$$

with parameters

$$(\mu_c, \phi_c) \in \{(800, 1000), (100, 500), (100, 400), (200, 500)\},$$

corresponding respectively to {Electronics, Apparel, Food & Beverages, Furniture & Appliances}.

- Payment Method ($v_M$): categorical variable over {Online Payment, Cash on Delivery} with probabilities

$$p(v_M \mid v_L = \text{Developed}) = (0.70, 0.30), \quad p(v_M \mid v_L = \text{Developing}) = (0.40, 0.60).$$

**Derived Economic Variables.** To evaluate the practical utility of synthetic data, we introduce two composite variables that simulate realistic segmentation and value estimation tasks. These targets are not included in the data generation process and are computed entirely post hoc. From a machine learning perspective, they serve to test whether models trained on synthetic data can support meaningful discriminative tasks involving high-order, nonlinear feature interactions that resemble real-world decision logic. The variables are constructed to reflect common use cases in business and marketing analytics: discount targeting and customer lifetime value estimation. Both combine domain knowledge and structural dependencies across price, age, payment behavior, and product category into proxy labels that require the generative model to faithfully reconstruct multiple conditional pathways.

**Discount Propensity ($d$).** This variable approximates price sensitivity, inspired by the concept of *price elasticity of demand*. Instead of modeling actual demand shifts, we simulate a "discount responsiveness score" based on how far a transaction price deviates from the expected norm and is modified by age and behavioral context. We first compute a z-score for the transaction price within its category:

$$z = \frac{v_X - \mu_{v_C}}{\phi_{v_C}},$$

where $\mu_{v_C}$ and $\phi_{v_C}$ are the category-wise mean and standard deviation. The composite score is then

$$S = -\tanh(z) + 0.01 \cdot \frac{(v_A - 35)^2}{100} + 0.5 \cdot \mathbf{1}\{v_M = \text{COD}\} + 0.3 \cdot \mathbf{1}\{v_L = \text{Developing}\}.$$

Each term encodes an interpretable behavioral signal: $\tanh(z)$ models diminishing sensitivity to extreme price deviations, the quadratic age term increases discount likelihood for younger and older users, peaking at age 35, COD users (+0.5) and those in developing regions (+0.3) are assumed to be more price-conscious based on consumer behavior research. The 0.01 scale factor balances the influence of the age component, keeping it comparable to binary modifiers. The final categorization is:

$$d = \begin{cases} \text{High}, & S > 1, \\ \text{Low}, & S < -1, \\ \text{Mid}, & \text{otherwise}. \end{cases}$$

**Lifetime Value Band ($\ell$).** This variable acts as a simplified proxy for *customer lifetime value* (CLV), an important metric for revenue forecasting and segmentation. CLV is typically defined as the discounted sum of future profits; we approximate this by incorporating transaction amount, demographic potential, and purchase channel/product effects. We compute a proxy score $\ell_0$ that reflects expected customer value as a function of transaction size, demographic potential, and behavioral modifiers:

$$\ell_0 = \frac{\sqrt{v_X} \cdot w_m}{\log(1 + |v_A - 35|) + 1} \cdot w_c,$$

where $w_m$ is a payment-method multiplier (1.2 for online, 0.85 for COD), encoding expected retention and conversion ease, $w_c$ is a product-category multiplier:

$$w_c = \begin{cases} 1.3, & v_C = \text{Electronics}, \\ 1.1, & v_C = \text{Apparel}, \\ 0.9, & v_C = \text{Food \& Beverages}, \\ 1.4, & v_C = \text{Furniture \& Appliances}. \end{cases}$$

The age-adjusted denominator reduces the projected value for extreme age groups, approximating long-term activity potential. We then discretize the proxy score $\ell_0$ into ordinal bands to reflect interpretable customer value segments:

$$\ell = \begin{cases} \text{High}, & \ell_0 > 20, \\ \text{Low}, & \ell_0 < 10, \\ \text{Mid}, & \text{otherwise}. \end{cases}$$

The thresholds and coefficients used are fixed, interpretable, and manually defined. They are not optimized on the data and are instead chosen to encode realistic, domain-informed assumptions. All values are fixed across all experiments and applied identically to both real and synthetic datasets to ensure a fair and reproducible evaluation of downstream discriminative utility.

## A.2 POPULATION

**Data** We use the 2023 American Community Survey (ACS) 1-Year Public Use Microdata Sample (PUMS) [1], a large-scale, nationally representative dataset released by the U.S. Census Bureau. The ACS is an ongoing survey program that provides detailed annual data on demographic, economic, housing, and social attributes of the U.S. population. Its microdata records offer rich insights that inform resource allocation, urban planning, public policy, and equity analysis at multiple geographic levels.

The 2023 ACS 1-Year PUMS includes anonymized household- and person-level records, making it well-suited for population synthesis. For this study, we extract data from households in South Carolina. We retain both household-level variables (e.g., number of persons, income, tenure, household type, vehicle ownership) and person-level attributes (e.g., age, race, education, employment, income). Since households vary in size and structure, the resulting data are inherently unstructured.

To ensure consistency and quality, we apply the following preprocessing steps:

- Household filtering: We include only family households with 2 to 5 members, drop entries with missing key fields, and normalize housing tenure and household type into interpretable categories.
- Person filtering: We retain individuals belonging to valid households and ensure non-missing values for core attributes like age and race. Records with anomalous or censored race categories or negative income are excluded.
- Variable recoding: Education levels are grouped into broader bands (e.g., high school, bachelor, master), employment status is collapsed, and race is mapped into simplified categories.
- Household-person consistency: Only households whose member count matches the reported size (NP) are retained, ensuring alignment between household and person tables.
- Final dataset: We obtain a cleaned population microdata sample consisting of 15,380 households, each associated with individual-level demographic and socioeconomic details.

The resulting dataset contains both household- and person-level records. The variables are:

**Household-level variables:**

- num_persons (2~5): Number of people in the household.
- housing_tenure (Owned, Rented): Indicates whether the household owns or rents their home.
- householder_type (Couple, Single Male, Single Female): Household composition based on adult structure.
- num_vehicles (0~6): Number of vehicles available to the household.

**Person-level variables:**

- age (0~99): Age of the individual.
- race (White, Black, Indian, Asian, Combined): Simplified racial categories derived from ACS coding.
- education (Preschool, Elementary, High School, Bachelor, Master, Doctorate): Highest educational attainment.
- employment (Under 16, Employed, Unemployed, Military): Employment status grouped into meaningful categories.
- income (0~4,209,995): Individual annual income in U.S. dollars.

Due to household size variation, the data is inherently unstructured, with each household containing a variable number of individual records. This structure poses a realistic and valuable challenge for generative models aimed at cross-level joint distribution synthesis. More detailed variable descriptions can be found at this link[2].

---

[1] https://www.census.gov/programs-surveys/acs/microdata/access.html
[2] https://www.census.gov/programs-surveys/acs/microdata/documentation.html

**Query Definitions** Accurate synthetic population data must enable meaningful analysis that informs real-world policy and planning. This includes identifying economic disparities, assessing social vulnerability, and supporting decisions on housing, mobility, and public services. To evaluate the practical value of such data, we define 16 policy-relevant queries grouped into six thematic categories. Each query includes a description and an explanation of its social significance.

**Equity**

(1) `median_income_black_households`

*Description*: Median total household income for households with at least one Black member.

*Social significance*: This indicator measures racial income inequality and highlights the economic disparities that affect Black households.

(2) `prop_multigenerational_racial`

*Description*: Proportion of multigenerational households, defined as those with both children under 18 and elderly members aged 65 or older, that include at least one non-White individual.

*Social significance*: This indicator sheds light on racial patterns in multigenerational living and provides insight into caregiving structures and household composition across different racial groups.

**Vulnerability**

(1) `prop_elderly_poverty`

*Description*: Proportion of households that include at least one member aged 65 or older and have a per-person income below 12,500.

*Social significance*: This indicator reflects the economic vulnerability of older adults and helps identify households at risk of poverty in later life.

(2) `prop_female_headed_poverty`

*Description*: Proportion of single female-headed households with a per-person income below 15,000.

*Social significance*: This indicator highlights the financial challenges faced by single women who lead households, pointing to gendered dimensions of poverty.

(3) `prop_high_dependency_ratio`

*Description*: Proportion of households in which the combined number of children and elderly members exceeds the number of working-age members.

*Social significance*: This indicator identifies households that carry a high care burden and may be more susceptible to financial and caregiving stress.

**Employment**

(1) `prop_high_edu_unemployed`

*Description*: Proportion of households where the highest level of education attained is a Master's degree or Doctorate, and no household members are employed.

*Social significance*: This indicator highlights potential underemployment or barriers to workforce participation among highly educated individuals.

(2) `prop_dual_earner_couples`

*Description*: Proportion of couple households where both adults are employed.

*Social significance*: This indicator reflects the prevalence of dual-income households and offers insight into labor force participation within traditional family structures.

(3) `avg_income_per_person_dual_earner`

*Description*: Average per-person income in couple households where both adults are employed.

*Social significance*: This indicator captures the financial outcomes associated with dual-earning and helps assess economic inequality among working families.

**Household**

(1) `age_gap_owners_vs_renters`

*Description*: Difference in the average age of household members between owner-occupied and renter-occupied households.

*Social significance*: This indicator reflects how age distribution relates to housing tenure and supports analysis of life stage patterns in housing access.

(2) `prop_multigenerational`

*Description*: Proportion of households that include both children under the age of 18 and elderly members aged 65 or older.

*Social significance*: This indicator highlights the prevalence of multigenerational living arrangements and the caregiving responsibilities they may involve.

**Demographics**

(1) `median_avg_age`

*Description*: Median value of the average age of household members across all households.

*Social significance*: This indicator provides a summary measure of age distribution at the household level and supports analysis related to population aging and demographic representation.

(2) `prop_families_with_children`

*Description*: Proportion of households that include at least one child under the age of 18.

*Social significance*: This indicator reflects the presence of families with children and helps identify demand for youth-oriented services and policies.

**Mobility**

(1) `prop_no_vehicle_low_income`

*Description*: Proportion of households with no vehicles and a per-person income below 15,000.

*Social significance*: This indicator measures transportation disadvantage among low-income households and highlights potential barriers to employment, healthcare, and other essential services.

(2) `prop_child_no_vehicle`

*Description*: Proportion of households with at least one child under the age of 18 and no vehicles.

*Social significance*: This indicator identifies families that may face challenges accessing child care, schools, and other child-related services due to limited transportation.

(3) `prop_elderly_no_vehicle`

*Description*: Proportion of households with at least one member aged 65 or older and no vehicles.

*Social significance*: This indicator highlights potential mobility limitations and the risk of social isolation among older adults without access to private transportation.

(4) `prop_high_vehicle_high_income`

*Description*: Proportion of households that own three or more vehicles and have a per-person income greater than 50,000.

*Social significance*: This indicator reflects concentrations of material wealth and can provide insight into patterns of resource consumption and their potential environmental impact.

### A.3 MOBILITY

**Data Sources.**   To construct our synthetic mobility task, we integrate two complementary datasets capturing different aspects of urban movement. The first is OpenPFLOW (Kashiyama et al., 2017), a high-resolution GPS trajectory dataset collected in the Tokyo metropolitan area. It provides detailed logs of individual movement traces, including agent ID, timestamp, and location (latitude and longitude). However, OpenPFLOW lacks semantic labels indicating trip purpose or activity. To address this, we incorporate external behavioral data from LLMob (Jiawei et al., 2024), which uses Foursquare check-ins to infer time-activity distributions across categories (e.g., "Shop & Service," "Food," "Travel & Transport") aggregated by time-of-day. Combining these two sources allows us to model not only where and when people move, but also why, a key requirement for realistic and policy-relevant mobility synthesis.

**Geographic Scope and Temporal Binning.**   We focus on a central urban region in Tokyo, bounded by longitude [139.6726, 139.8896] and latitude [35.6004, 35.7788]. This area encompasses both dense commercial hubs and residential zones, offering diverse mobility patterns, as shown in Figure 7. Time is discretized into seven semantically meaningful intervals based on established guidelines in activity-based travel modeling: 0∼6, 6∼9, 9∼12, 12∼14, 14∼17, 17∼20, 20∼24. These bins reflect common human routines (e.g., commuting peaks, lunch breaks, nighttime activity) and support alignment with LLMob activity distributions.

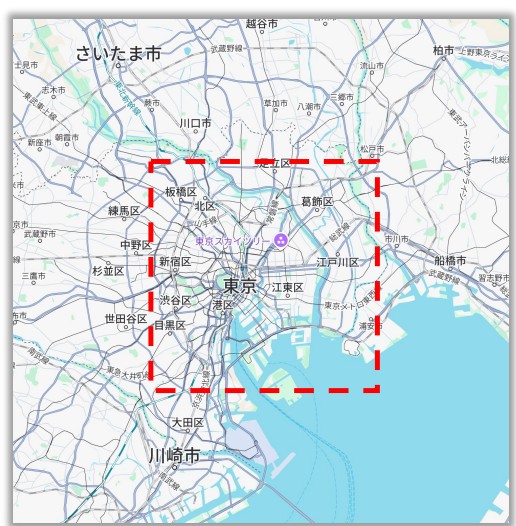

Figure 7: Study area within the Tokyo metropolitan region.

**Grid-Based Spatial Representation.**   To convert raw GPS points into a spatially structured format, we partition the study area into a uniform grid of square regions (2km resolution). Each trip's origin and destination are mapped to regions via geometric projection. This discretization supports several goals: (1) privacy, by abstracting fine-grained trajectories; (2) compatibility with OD (origin-destination) matrix modeling; and (3) alignment with spatial units used in real-world planning.

**Trip Segmentation and Transport Mode Assignment.**   To construct trip-level records from raw trajectory data, we first segment each agent's movement timeline into individual trips by identifying transition points where stationary periods are followed by motion. This segmentation is based on temporal gaps and changes in movement signals. From each trip, we extract its departure and arrival coordinates, time of occurrence, and associated transport mode. However, OpenPFLOW's original mode labels are often noisy, incomplete, or unreliable, which limits their use for structural modeling and evaluation.

To address this, we reassign transport modes using a distance-sensitive probabilistic model. Specifically, we calculate the great-circle distance between the origin and destination regions of each trip and assign a transport mode based on a calibrated distribution that reflects common behavioral patterns. Shorter trips are more likely to be labeled as walking, mid-range trips as biking, and longer trips as driving. This approach improves behavioral realism and produces a mode variable that is both interpretable and structurally informative. Placeholder trips and intra-region trips are removed to ensure all retained trips exhibit spatial displacement and meaningful mode behavior. The resulting dataset serves as a consistent and tractable basis for synthesis and evaluation.

**Activity.**   Since OpenPFLOW lacks activity labels, we do not annotate real trips with activity types. Instead, we leverage LLMob's time-conditioned activity distribution as external structural guidance. This distribution is used as a target macro-statistic during proposal generation, encouraging synthetic

trips to match realistic time-activity patterns without directly assigning activity labels to observed data.

**Outputs.** The final dataset includes:

- A cleaned trip table with fields: `time_str`, `activity`, `od`, and `transport`.
- Grid region definitions and metadata (region size, bounds, projection).
- OD matrices per time bin, capturing trip flows between region pairs.
- A region-region distance matrix used for spatial modeling and transport inference.
- A per-region top-$k$ activity summary describing local land use patterns.

**Design Rationale.** This mixed-source design reflects a practical reality: in many urban contexts, movement traces (from GPS or transit data) and semantic behaviors (from points of interest or social platforms) are available from separate sources. *Synthesizing mobility data that integrates both dimensions requires models to learn from partially aligned, heterogeneous inputs.* Our preprocessing aims to bridge this gap, transforming fragmented empirical records into a coherent, interpretable format that supports structure-aware synthesis. By modeling OD patterns, temporal rhythms, transport behavior, and activity semantics jointly, this setup enables robust evaluation of synthetic data quality from both spatial and behavioral perspectives.

## B   EXPERIMENT SETUPS

### B.1   IMPLEMENTATION DETAILS

**Macro-statistics**   For continuous variables, we adopt a hierarchical, quantile-based binning approach to construct robust and interpretable macro-statistics. In general practice, macro-statistics are defined directly according to the available target specification (e.g., survey tables or census summaries). In our experiments, however, since we require real micro-record data to compare against baselines, we compute the corresponding macro-statistics from the available ground-truth datasets. More broadly, when micro-records are available in real-world contexts, this same procedure can be applied to derive finer-grained macro-statistics, enabling more sensitive discrepancy evaluation and correction.

Specifically, for each continuous variable, the observed value range is partitioned into a fixed number of main bins (by default, 6) using quantiles of the real data distribution, such that each bin contains approximately the same number of records. This avoids bin sparsity and ensures fair representation of all regions of the distribution, including heavy tails and outliers.

To further capture local distributional features, we perform a secondary refinement step. The main bin with the largest positive discrepancy between real and synthetic data (i.e., where the real proportion exceeds the synthetic proportion by the largest margin) is selected for subdivision. Within this bin, we create a fixed number of sub-bins using uniform spacing over its range. Frequencies of these sub-bins are then normalized so that their sum equals the original frequency of the parent bin, ensuring consistency.

This two-stage binning scheme highlights both global and local mismatches between real and synthetic data, enabling more sensitive detection and correction of model errors. The quantile-based main bins ensure stability across diverse datasets, while sub-bin refinement targets the most relevant region for detailed adjustment.

All discrete variables (and discretized bins of continuous variables) are then summarized as frequency tables. For joint macro-statistics, we compute empirical contingency tables over selected groups of variables, applying the same binning strategy to any continuous components. This unified, type-agnostic representation supports scalable synthesis and provides interpretable signals for LLM-guided correction.

**Discrepancy**   The discrepancy measure $Q(\cdot, \cdot)$ between target and synthetic data is computed as the difference between their categorical frequency tables, at both marginal and joint levels. In practice, since the synthesis process can only add records rather than remove them, we focus on *positive discrepancies*, that is, bins where the target proportion exceeds the synthetic proportion. Formally, for a target statistic $\phi$ with synthetic estimate $\hat{\phi}$, the discrepancy is defined as

$$Q(\hat{\phi}, \phi) = \phi - \hat{\phi},$$

measured entry-wise across all bins or contingency cells. This formulation ensures that discrepancy signals always correspond to "underrepresented" regions in the synthetic dataset, which are precisely the areas actionable by further generation.

Crucially, because our macro-statistics are constructed from interpretable bins, every discrepancy can be attributed to a specific value range (for continuous variables), category (for discrete variables), or a combination thereof (for joints). This *attributability* property is essential for the iterative synthesis loop: it pinpoints exactly where the synthetic data underestimates the target distribution and guides the LLM to generate corrective records in those regions, rather than making undirected global adjustments.

**Hyperparameters**   We generate 5 proposals per iteration for a total of 100 iterations. In each iteration, 3 joint variable combinations are inferred for grounding. LLM inference is performed using GPT-4.1-nano with a temperature of 0.8, and Qwen2.5-7B-Instruct via the VLLM (Kwon et al., 2023) framework. Generation hyperparameters are as follows: `max_new_tokens` = 2048, `temperature` = 0.7, `top_k` = 20, and `top_p` = 0.98.

For macro-statistics and contingency calculations, if only partial target statistics (e.g., available macro-statistics) are accessible, we base the calculations on these statistics. However, if the full data

is available, we apply a two-level quantile-based binning scheme for continuous variables, with 6 main bins and 8 sub-bins per main bin, capturing both global and local structure. Discrete variables and discretized bins are handled uniformly when computing marginal and joint frequency tables.

All experiments are repeated at least three times, with the mean result reported. The uncertainty in our results arises from the stochasticity of LLM generation.

**Hardware**    All experiments are conducted on a server equipped with an Intel Xeon E5-2698 v4 CPU (40 threads), 252 GB of RAM, and four NVIDIA Tesla V100 GPUs with 32 GB of memory each.

## B.2 BASELINES

### B.2.1 E-COMMERCE TRANSACTION SYNTHESIS

We briefly summarize the baseline models used in our experiments:

**TVAE (Xu et al., 2019)** Tabular Variational Autoencoder (TVAE) models tabular data by encoding mixed-type variables into a continuous latent space via a VAE framework. It supports conditional sampling and employs Gumbel-Softmax reparameterization for discrete features. TVAE captures global structure but tends to oversmooth discrete modes, especially under imbalanced categories.

**CTGAN (Xu et al., 2019)** Conditional Tabular GAN (CTGAN) improves over TVAE by introducing a conditional GAN architecture. It uses mode-specific normalization and a balanced training sampler to enhance rare-category representation. While improving fidelity and diversity, CTGAN suffers from the inherent instability of GAN training and mode collapse.

**CopulaGAN (d'Alessandro et al., 2017)** CopulaGAN integrates copula-based modeling into the GAN framework to separate marginal estimation from dependency modeling. By embedding copula regularization into the loss function, it improves the alignment between synthetic and real data in both marginal and joint distributions. This structure-aware regularization enhances statistical fidelity across variable combinations.

**GReaT (Borisov et al., 2022)** GReaT reformulates tabular synthesis as a sequence modeling task, using an autoregressive Transformer trained on linearized table rows. It supports pretraining and zero-shot generation, demonstrating strong generalization. However, it lacks explicit distributional control, and statistical alignment with the real data is not guaranteed.

**TabSyn (Zhang et al., 2023a)** TabSyn combines a VAE encoder with a score-based diffusion model in the latent space. Mixed-type tabular data are first embedded into a structured continuous space, where a denoising diffusion model performs generative sampling. This hybrid design ensures high fidelity, fast sampling, and better category preservation. TabSyn achieves strong performance on mixed-type datasets with improved statistical accuracy and diversity.

**IPF (Kolenikov, 2014)** Iterative Proportional Fitting (IPF) is a classical macro-supervised method that operates strictly on aggregated data without access to individual micro-records. It iteratively adjusts a seed contingency table (or joint distribution) to match a set of observed marginal distributions (macro-statistics). By preserving the interaction structure of the seed while satisfying marginal constraints, IPF serves as a robust baseline for reconstructing joint distributions solely from macro-level summaries.

**Spada (Yang et al., 2025)** Spada represents a hybrid micro-macro supervised approach. Unlike traditional methods that rely solely on micro-level likelihood, we explicitly incorporate a macro-level $L_2$ loss term into the training objective. This additional regularization penalizes the Euclidean distance between the aggregated statistics of the generated samples and the ground-truth macro-statistics. By simultaneously optimizing for record-level fidelity and global distributional alignment, SPADA ensures that the synthetic data respects both individual plausibility and aggregate constraints.

**LLM-ICL (Brown et al., 2020)** LLM-ICL serves as a purely macro-supervised baseline leveraging the In-Context Learning capabilities of Large Language Models. In this setup, GPT-5.1 is provided with the entire set of target macro-statistics as prompts and instructed to directly generate micro-records. We implemented this baseline through a 40-turn interactive session within the ChatGPT interface. This design allows the model to benefit from built-in conversational memory and fully leverage all previously generated records, providing the ICL baseline with the strongest possible conditions for macro-aware generation without parameter fine-tuning.

### B.2.2 POPULATION SYNTHESIS

**CP (Candecomp/Parafac) Tensor Factorization (Carroll & Chang, 1970)** The CP baseline discretizes the joint household-person population into a high-dimensional contingency tensor $\mathcal{X} \in \mathbb{R}^{d_1 \times \cdots \times d_K}$, where each axis corresponds to a household or person attribute. A non-negative CP decomposition is applied:

$$\hat{\mathcal{X}}_{i_1,\ldots,i_K} = \sum_{r=1}^{R} \lambda_r \prod_{k=1}^{K} A^{(k)}_{i_k,r},$$

where $\lambda_r$ denotes the weight of component $r$, and $A^{(k)}$ are factor matrices. After normalization, the model yields a mixture-of-product-of-categoricals (MPC) distribution:

$$Q_{\text{CP}}(i) = \sum_{r=1}^{R} w_r \prod_{k=1}^{K} a^{(k)}_{i_k,r}.$$

Households are assigned to their most likely latent group via $g = \arg\max_r w_r \prod_k a^{(k)}_{i_k,r}$, and within each group, a person-level non-negative CP factorization model $M_g$ is fit for joint household-member sampling.

**HMM (Sun et al., 2018) and NVI (Mnih & Gregor, 2014)** We presents two hierarchical generative baselines used for structured population synthesis: HMM (Product Multinomial Hierarchical Mixture Model) and NVI (Neural Variational Inference). Both approaches are based on a shared latent graphical model but differ in the inference procedure and parameterization.

The generative process assumes a two-layer structure:

$$z_i \sim \text{Cat}(\lambda),$$
$$x_i^{(k)} \mid z_i = g \sim \text{Cat}(\Phi_g^{(k)}),$$
$$z_{ij} \sim \text{Cat}(\mu_g),$$
$$x_{ij}^{(\ell)} \mid z_{ij} = m \sim \text{Cat}(\theta_{gm}^{(\ell)}),$$

where $z_i$ is the latent household class and $z_{ij}$ is the latent member class of person $j$ in household $i$. All conditional distributions are categorical, regularized by symmetric Dirichlet priors.

**HMM: Product Multinomial Hierarchical Mixture Model** The HMM baseline employs the Expectation-Maximization (EM) algorithm for parameter estimation, alternating between posterior updates of latent variables and maximization steps.

E-step (posterior responsibilities):

$$\gamma_i^g = \Pr(z_i = g \mid \text{data}),$$
$$\rho_{ij}^m = \Pr(z_{ij} = m \mid z_i = g, \text{data}).$$

M-step (parameter updates):

$$\lambda_g \leftarrow \frac{1}{N} \sum_i \gamma_i^g,$$

$$\mu_{gm} \leftarrow \frac{\sum_{ij} \gamma_i^g \rho_{ij}^m}{\sum_{ij} \gamma_i^g},$$

$$\Phi_g^{(k)}(c) \leftarrow \frac{\sum_i \gamma_i^g \mathbb{I}(x_i^{(k)} = c)}{\sum_i \gamma_i^g},$$

$$\theta_{gm}^{(\ell)}(c) \leftarrow \frac{\sum_{ij} \gamma_i^g \rho_{ij}^m \mathbb{I}(x_{ij}^{(\ell)} = c)}{\sum_{ij} \gamma_i^g \rho_{ij}^m}.$$

This formulation is particularly suited for moderate-scale datasets with clear latent groupings and yields interpretable household and person class structures.

**NVI: Neural Variational Inference**    The NVI baseline implements amortized mean-field variational inference using neural networks to parameterize posterior distributions. The variational posterior is factorized as:

$$q(z, z_{1:M}) = \prod_i q_\Phi(z_i \mid x_i) \prod_j q_\psi(z_{ij} \mid z_i, x_{ij}),$$

Both $q_\Phi$ and $q_\psi$ are implemented using multilayer perceptrons (MLPs). Discrete latent variables $z_i$ and $z_{ij}$ are reparameterized via the Gumbel-Softmax trick:

$$\tilde{z}_i = \text{Softmax}\left(\frac{\log \gamma_i + g}{\tau}\right), \quad g \sim \text{Gumbel}(0, 1),$$

with temperature $\tau$ gradually annealed to 0.1. The model is trained to maximize the evidence lower bound (ELBO):

$$\mathcal{L} = \mathbb{E}_{q(z)}[\log p(x, z) - \log q(z \mid x)],$$

where all categorical likelihoods are modeled using softmax-parameterized logits. Optimization is performed via mini-batch stochastic gradient descent, and missing values are handled via masking or learned probabilities. This variational method provides a scalable and flexible alternative to classical EM, suitable for high-dimensional structured populations.

## B.3 METRICS

**E-commerce Transactions**    We evaluate the quality of synthetic data using a suite of metrics that jointly assess marginal fidelity, structural alignment, and downstream utility. For each metric, we detail both its purpose and our specific implementation on the e-commerce dataset.

- **Wasserstein Distance (Was) (Villani et al., 2008)**: Measures the discrepancy between real and synthetic distributions for continuous variables, capturing both location and shape. *Implementation:* We compute 1D Wasserstein distances for continuous attributes such as `price` and `age`, using their empirical distributions.

- **Total Variation Distance (TVD) (Levin & Peres, 2017)**: Quantifies the maximum probability mass difference across categories for discrete variables. *Implementation:* We compute TVD for each categorical feature (e.g., `product_category`, `user_gender`) between real and synthetic marginal distributions.

- **Jensen-Shannon Divergence (JSD) (Lin, 2002)**: Evaluates how well the generator preserves joint distributions over related variable sets. JSD is symmetric and bounded, making it suitable for sparse, high-dimensional comparisons. *Implementation:* We compute JSD over selected variable combinations that reflect latent structure in the data, including: $[v_A, v_G, v_C]$ = `user_age, user_gender, product_category`; $[v_C, v_X]$ = `product_category, payment_method`; $[v_L, v_M]$ = `location_tier, price`. Distributions are estimated via frequency tables and normalized histograms.

- **Classifier Two-Sample Test Gap (Gap) (Lopez-Paz & Oquab, 2016)**: Measures the distinguishability between real and synthetic data using a supervised model. A gap close to 0 implies synthetic data is indistinguishable from real data. *Implementation:* We train a random forest classifier to discriminate real vs. synthetic records using all raw variables as input. We report the deviation from 50% test accuracy (i.e., `Gap` $= |accuracy - 0.5|$).

- **F1 Score (Macro-Averaged)**: Assesses the downstream utility of synthetic data for real-world ML tasks. Higher F1 scores indicate that predictive signals in real data are preserved. *Implementation:* We derive label targets such as discount propensity and LTV buckets from the real data, then train classifiers (logistic regression, decision trees, and random forests) on synthetic records. Evaluation is conducted on a held-out set of real data. Reported scores are macro-averaged across all classes.

- **Rejection Rate**: Captures the proportion of synthetic records that are filtered out due to logical inconsistencies or invalid values. *Implementation:* We apply validation rules after generation (e.g., non-negative price, plausible age ranges). The rejection rate is the fraction of records discarded due to these checks.

**Populations**    Population synthesis involves household-level microdata with variable-length records. We use metrics that directly reflect structural and policy-relevant correctness.

- **Mean Relative Error (MRE)**: Evaluates the fidelity of synthetic populations on downstream analytical tasks. *Implementation:* We define 16 queries (e.g., % of single-parent households with unemployed adults") spanning key policy domains. We compare the query results on synthetic vs. real populations and report the average relative error.

- **Rejection Rate**: Reflects adherence to demographic and structural constraints in microdata generation. *Implementation:* Post-generation, we apply validation checks such as "parent must be older than child" and "head of household must be present." Rejection rate is the proportion of records violating these constraints.

Together, these metrics provide a comprehensive evaluation framework capturing statistical fidelity, structural realism, and practical utility across diverse data synthesis settings.

# C SUPPLEMENTARY RESULTS

## C.1 E-COMMERCE TRANSACTION SYNTHESIS

In this subsection, we present additional visualizations comparing the distributions of real and synthetic data.

**Synthetic Distribution Visualization**    In Figure 8 and Figure 9, we visualize the full marginal distributions for all individual variables as well as selected joint distributions over correlated variable pairs, comparing synthetic data generated by LLMSYNTHOR and GReaT. These plots complement the quantitative metrics in the main paper by illustrating how well each method captures complex interactions and category-conditioned patterns. LLMSYNTHOR shows strong visual alignment with the real data across both marginal and conditional views, especially in structured relationships such as payment method by location tier and product category by age and gender. GReaT captures some trends but tends to over-smooth (Price by Product Category) or under-represent distributional variation (Price). These comparisons highlight the qualitative fidelity of LLMSYNTHOR in preserving both local and global statistical structure.

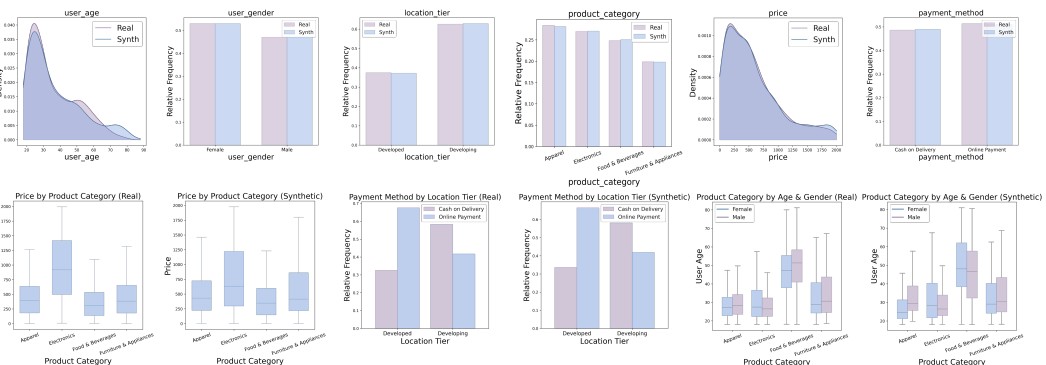

Figure 8: Visualization of the synthetic distribution generated by LLMSYNTHOR.

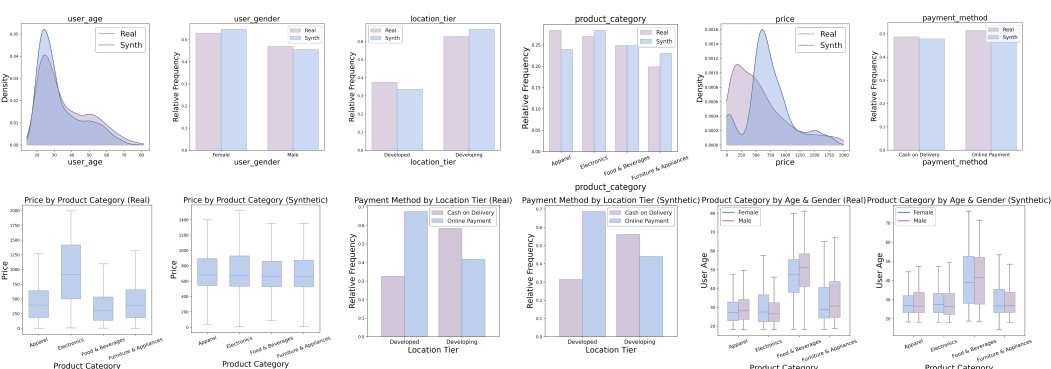

Figure 9: Visualization of the synthetic distribution generated by GReaT.

**Convergence Analysis.** Figure 10 presents empirical convergence curves tracking distributional discrepancy metrics across 100 synthesis iterations. We evaluate both marginal and joint distributions using a suite of distances, including total variation distance (TVD), Wasserstein distance, energy distance, and Hellinger distance. All metrics consistently decrease over time, indicating that LLM-SYNTHOR iteratively reduces structural divergence between real and synthetic data. Joint patterns such as product price by category and payment method by location tier also converge, demonstrating that high-order dependencies are faithfully preserved.

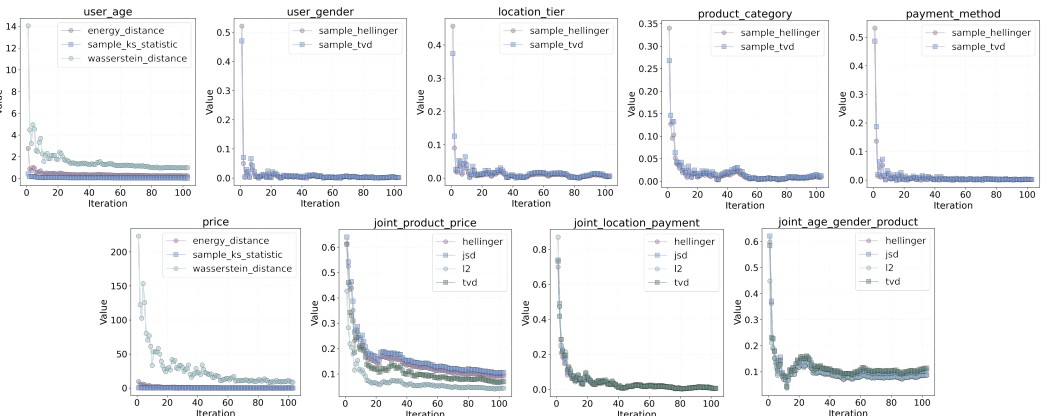

Figure 10: Convergence curve showing distributional discrepancy metrics over iterations.

The Figure 11 depicts the overall mean distributional distance between real and synthetic data as a function of synthesis iterations. To compute this curve, we aggregate several distributional metrics, including total variation distance (TVD), Wasserstein distance, energy distance, maximum mean discrepancy (MMD), Hellinger distance, Jensen-Shannon divergence (JSD), and Kullback-Leibler (KL) divergence, across both marginal and joint variables. For each iteration, the mean of all selected metrics over all variables is reported. The solid line indicates this average, while the shaded regions represent uncertainty intervals: $\pm 1$, $\pm 2$, and $\pm 3$ standard deviations, along with the min-max range across repeated runs. The sharp initial decrease and rapid plateau show that most statistical alignment is achieved within the first 50 iterations, with diminishing returns beyond 200. This highlights the efficiency and stability of the iterative synthesis process and provides practical guidance for setting the number of iterations.

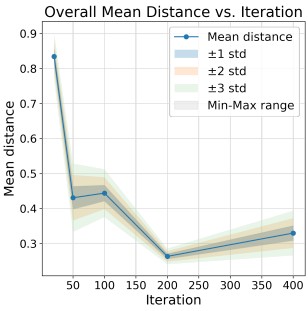

Figure 11: Mean distributional distance across synthesis iterations.

| Methods | Train Time | Generation Time | Age Was | Gender Tvd | Location Tvd | Category Tvd | Price Was | Payment Tvd | [v_A, v_G, v_C] JSD | [v_C, v_X] JSD | [v_L, v_M] JSD |
|---|---|---|---|---|---|---|---|---|---|---|---|
| GReaT | 724.7 ± 8.4 | 36.3 ± 1.2 | 2.862 | 0.016 | 0.039 | 0.045 | 169.87 | 0.009 | 0.087 | 0.382 | 0.038 |
| **Ours (10 iters)** | 0 | 24.7 ± 3.1 | 2.306 | 0.013 | 0.026 | 0.042 | 46.68 | 0.016 | 0.268 | 0.198 | 0.081 |
| **Ours (20 iters)** | 0 | 45.7 ± 5.5 | 1.302 | 0.002 | 0.002 | 0.012 | 21.15 | 0.003 | 0.076 | 0.167 | 0.025 |
| **Ours (100 iters)** | 0 | 238.6 ± 14.3 | 1.13 | 0.002 | 0.002 | 0.010 | 12.76 | 0.003 | 0.071 | 0.134 | 0.007 |

Table 6: Impact of iteration counts on efficiency and data fidelity. Unlike GReaT, our method requires zero training time. The table illustrates the flexibility of our framework: lower iterations (10 iters) yield faster generation speeds, while higher iterations (100 iters) significantly minimize distributional discrepancies (Wasserstein, TVD, and JSD).

**Efficiency Analysis.** We compare the runtime efficiency of our method (LLMSYNTHOR) against GReaT, a strong LLM-based baseline. Unlike GReaT, which requires training, LLMSYNTHOR performs zero-shot synthesis via LLM prompting. For this evaluation, we generate 2,000 records in both methods. GReaT is run using a batch size of 124, the maximum size recommended in its original paper to ensure performance, and trained over 200 epochs, which we find sufficient for convergence. LLMSYNTHOR is evaluated using GPT-4.1-nano via API:

As shown in Figure 12, Unlike GReaT, which incurs a significant computational overhead due to model fine-tuning ($724.7 \pm 8.4$ seconds), LLMSYNTHOR performs zero-shot synthesis requiring no training time. Crucially, our method demonstrates superior efficiency and effectiveness even at low iteration counts. With only 10 iterations, LLMSYNTHOR achieves a sampling time of 24.7 seconds, which is faster than GReaT's sampling time of 36.3 seconds, while simultaneously producing higher-quality data. For instance, the Wasserstein distance for the 'Price' column is significantly reduced from 169.87 (GReaT) to 46.68 (LLMSYNTHOR, 10 iters), and the 'Age' column improves from 2.86 to 2.31. While increasing iterations to 20 or 100 allows for further refinement of distributional alignment (as seen in the reduced JSD scores), the 10-iteration setting offers an optimal trade-off, outperforming the baseline in both runtime and data fidelity.

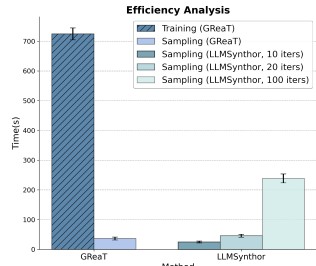

Figure 12: Efficiency comparison between GReaT and LLMSYNTHOR.

| Methods | Rej. % | Age Was | Gender Tvd | Location Tvd | Category Tvd | Price Was | Payment Tvd | [v_A, v_G, v_C] Jsd | [v_C, v_X] Jsd | [v_L, v_M] Jsd |
|---|---|---|---|---|---|---|---|---|---|---|
| TabSyn | $1.9 \pm 0.3$ | 1.20 | 0.012 | 0.007 | 0.045 | 114.12 | 0.028 | 0.083 | 0.237 | 0.027 |
| **Ours (GPT-4.1-nano)** | **$0.3 \pm 0.1$** | 1.13 | 0.002 | **0.002** | 0.022 | 12.762 | 0.003 | **0.071** | **0.134** | **0.007** |
| **Ours (Qwen2.5-7B)** | $0.5 \pm 0.1$ | 1.03 | **0.001** | 0.013 | 0.007 | **8.691** | **0.001** | 0.089 | 0.16 | 0.021 |
| **Ours (Qwen2.5-3B)** | $0.5 \pm 0.1$ | **0.93** | **0.001** | **0.002** | **0.003** | 14.367 | 0.002 | 0.082 | 0.198 | 0.139 |

Table 7: Quantitative ablation study on LLM backbones. We compare the synthesis quality of LLMSYNTHOR using GPT-4.1-nano, Qwen2.5-7B-Instruct, and Qwen2.5-3B-Instruct against the TabSyn baseline. The results highlight the framework's adaptability across models of varying parameter sizes, with bold values indicating the best performance in each category.

**Ablation: LLM Backbone Robustness.** Table 7 and Figures 13 and 14 present the performance of LLMSYNTHOR when utilizing open-source models, specifically Qwen2.5-7B-Instruct[3] and Qwen2.5-3B-Instruct[4], as alternatives to GPT-4.1-nano. The results demonstrate that our framework is highly robust to the choice of LLM backbone. Notably, the 7B model achieves parity with GPT-4.1-nano across most metrics, maintaining extremely low rejection rates (0.5%) and high fidelity in both marginal and joint distributions. Even when scaling down to the lightweight 3B model, LLMSYNTHOR continues to generate valid distributions that align well with real data, particularly in marginal statistics (e.g., achieving the lowest Wasserstein distance on 'Age'). However, we observe a slight degradation in joint distribution metrics (JSD) for the 3B model (e.g., $[v\_L, v\_M]$ rises to 0.139). We hypothesize that this is due to the limited planning and reasoning capabilities inherent to smaller parameter sizes, which may struggle with the complex multi-objective optimization required to reduce multiple discrepancies simultaneously during iterative refinement. Nevertheless, the overall performance remains competitive, confirming that LLMSYNTHOR can be effectively deployed even in resource-constrained environments.

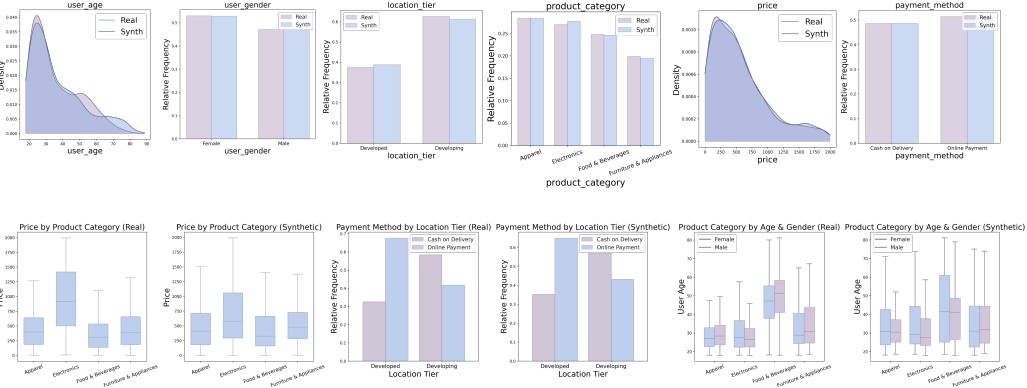

Figure 13: Performance of LLMSYNTHOR using Qwen2.5-7B-Instruct as the LLM backbone.

---

[3] https://huggingface.co/Qwen/Qwen2.5-7B-Instruct
[4] https://huggingface.co/Qwen/Qwen2.5-3B-Instruct

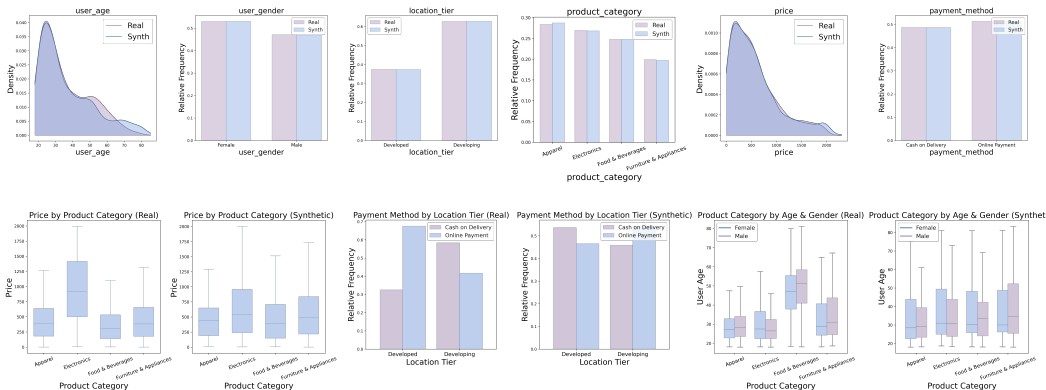

Figure 14: Performance of LLMSYNTHOR using Qwen2.5-3B-Instruct as the LLM backbone.

**Ablation: Effectiveness of Variable Dependency Inference.** Figure 15 presents the outcome of disabling joint structure grounding and guiding generation using only marginal statistics. While marginal distributions remain accurate, joint relationships (e.g., age-gender-product interactions) degrade noticeably. This highlights the importance of explicitly modeling structural dependencies for capturing realistic correlations, validating the need for our copula-based joint grounding strategy.

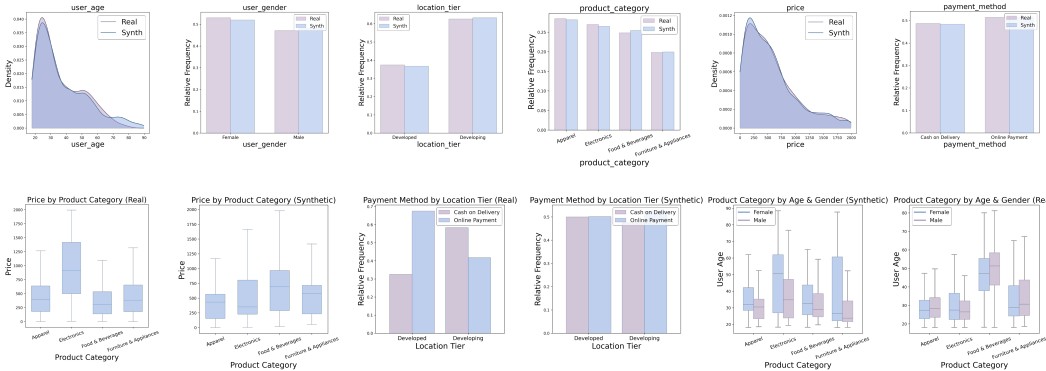

Figure 15: Ablation study: performance of LLMSYNTHOR when guided only by marginal statistics, without joint structure grounding.

## C.2 POPULATION SYNTHESIS

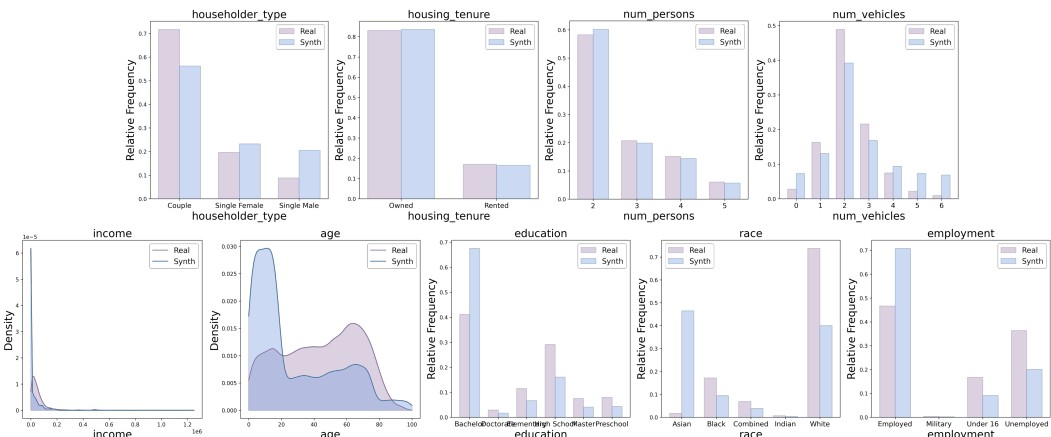

Figure 16: Visualization of the synthetic distribution generated by NVI.

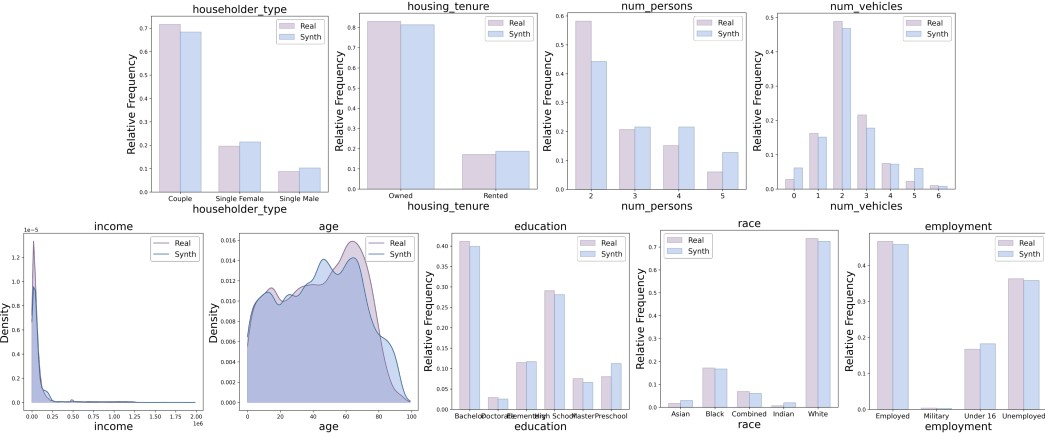

Figure 17: Visualization of the synthetic distribution generated by LLMSYNTHOR.

### C.2.1 DISTRIBUTIONAL COMPARISON IN POPULATION SYNTHESIS.

Figure 16 and 17 compare the marginal distributions of real data against synthetic data generated by NVI and LLMSYNTHOR, respectively. Both methods maintain reasonable alignment on household-level and individual-level variables. However, LLMSYNTHOR more closely matches real distributions, particularly on skewed variables such as income and age, as well as on categorical distributions like employment and race. This suggests that the iterative structure-guided mechanism in LLMSYNTHOR improves fidelity even in challenging, high-variance settings.

## C.2.2 FULL UTILITY RESULTS.

| | Real | CP | | | HMM | | | NVI | | | LLMSYNTHOR | | |
|---|---|---|---|---|---|---|---|---|---|---|---|---|---|
| | | Synth. | Δ | RE | Synth. | Δ | RE | Synth. | Δ | RE | Synth. | Δ | RE |
| median_income_black_households | 58400 | 103128 | 44728 | 0.77 | 89355 | 30955 | 0.53 | 79100 | 20700 | 0.35 | 54334.5 | 4065.5 | 0.07 |
| prop_multigenerational_racial | 0.02 | 0.18 | 0.17 | 10.81 | 0.18 | 0.16 | 10.47 | 0.18 | 0.16 | 10.63 | 0.01 | 0.01 | 0.43 |
| prop_elderly_poverty | 0.05 | 0.12 | 0.07 | 1.56 | 0.15 | 0.11 | 2.35 | 0.12 | 0.08 | 1.68 | 0.06 | 0.01 | 0.3 |
| prop_female_headed_poverty | 0.07 | 0.06 | 0.01 | 0.16 | 0.04 | 0.03 | 0.39 | 0.04 | 0.03 | 0.43 | 0.1 | 0.03 | 0.44 |
| prop_high_edu_unemployed | 0.05 | 0.15 | 0.09 | 1.67 | 0.04 | 0.01 | 0.18 | 0.04 | 0.02 | 0.34 | 0.05 | 0.01 | 0.15 |
| prop_dual_earner_couples | 0.12 | 0.01 | 0.11 | 0.88 | 0.09 | 0.03 | 0.27 | 0.16 | 0.04 | 0.3 | 0.17 | 0.05 | 0.43 |
| avg_income_per_person_dual_earner | 77984.05 | 38113.62 | 39870.43 | 0.51 | 39476.37 | 38507.68 | 0.49 | 64007.69 | 13976.36 | 0.18 | 79218.42 | 1234.37 | 0.02 |
| prop_families_with_children | 0.32 | 0.4 | 0.08 | 0.25 | 0.41 | 0.09 | 0.29 | 0.42 | 0.11 | 0.34 | 0.27 | 0.05 | 0.16 |
| prop_multigenerational | 0.03 | 0.22 | 0.19 | 6.53 | 0.17 | 0.14 | 4.92 | 0.2 | 0.17 | 5.82 | 0.03 | 0 | 0.02 |
| prop_high_dependency_ratio | 0.31 | 0.23 | 0.08 | 0.25 | 0.24 | 0.07 | 0.23 | 0.3 | 0.01 | 0.04 | 0.38 | 0.07 | 0.21 |
| median_avg_age | 48.67 | 44 | 4.67 | 0.1 | 44.67 | 4 | 0.08 | 43 | 5.67 | 0.12 | 47 | 1.67 | 0.03 |
| age_gap_owners_vs_renters | 14.92 | 0.29 | 14.64 | 0.98 | -0.61 | 15.53 | 1.04 | -0.11 | 15.03 | 1.01 | 20.65 | 5.72 | 0.38 |
| prop_child_no_vehicle | 0.01 | 0.03 | 0.02 | 2.05 | 0.01 | 0 | 0.22 | 0.01 | 0 | 0.18 | 0.01 | 0 | 0.5 |
| prop_elderly_no_vehicle | 0.01 | 0.05 | 0.03 | 3.17 | 0.02 | 0.01 | 0.46 | 0.01 | 0 | 0.16 | 0.01 | 0 | 0.38 |
| prop_high_vehicle_high_income | 0.1 | 0.08 | 0.02 | 0.17 | 0.08 | 0.02 | 0.22 | 0.09 | 0.01 | 0.11 | 0.08 | 0.02 | 0.24 |
| prop_no_vehicle_low_income | 0.01 | 0.02 | 0.01 | 0.5 | 0.01 | 0.01 | 0.53 | 0.01 | 0.01 | 0.51 | 0.02 | 0 | 0.29 |

Table 8: Full utility results showing synthetic values (Synth.), absolute errors (Δ), and relative errors.

Table 8 reports detailed results for all 16 utility queries used in the population synthesis evaluation, including the real values, synthetic estimates, absolute error (Δ), and relative error (RE). These queries span equity, vulnerability, employment, demographics, and mobility-related indicators. LLMSYNTHOR consistently achieves the lowest relative error across most metrics, confirming its ability to preserve both marginal statistics and complex structural dependencies. Notably, it significantly outperforms all baselines on sensitive or high-variance indicators such as median_income_black_households, avg_income_per_person_dual_earner, and prop_multigenerational. These results reinforce the model's utility in supporting policy-relevant analysis with privacy-preserving synthetic data.

### C.2.3 Manual Rejection Sampling for Record-realism Evaluation.

**Population**   To quantitatively evaluate the realism of generated micro-records, we construct a rule-based rejection sampling framework grounded in the semantics of population data. Among possible application domains, we choose the household population synthesis setting for realism evaluation because it contains rich micro-level semantics and well-defined constraints that make it relatively unambiguous to distinguish realistic from unrealistic records. For example, it is unrealistic for a household labeled as a couple to have no adults. This makes household population synthesis a particularly suitable benchmark for assessing fine-grained realism of synthetic records produced by LLMSYNTHOR and baseline generators.

The rejection framework is manually designed and applied uniformly across all evaluated models. It operates in two stages. Each generated household is first checked by validating every individual it contains. If any person fails person-level checks, the entire household is rejected. If all persons are valid, the household is then validated using household-level rules. Thus, a household is accepted if and only if (i) all persons are individually valid, and (ii) the household metadata and composition satisfy the predefined rules.

**Person-level rules** include:

1. **Age range**: age must lie within $[0, 99]$.

2. **Categorical validity**: fields such as `employment`, `race`, and `education` must belong to predefined valid categories.

3. **Age-education consistency**:

   (a) individuals under 5 years must have `Preschool` education;
   (b) individuals under 18 must not exceed `High School`.

4. **Income bounds**: income must be non-negative and less than 4,209,995, an empirically chosen upper limit to exclude unrealistic values.

**Household-level rules** include:

1. **Person count range**: `num_persons` must be between 2 and 5.

2. **Vehicle count range**: `num_vehicles` must lie within $[0, 6]$.

3. **Categorical validity**: `housing_tenure` and `householder_type` must belong to known enumerated types.

4. **Person count match**: the number of listed persons must exactly match `num_persons`.

5. **Adult consistency for couples**: households labeled `Couple` must contain at least two individuals aged 18 or older.

For each rule, we record the number of evaluated records, rejection counts, and rejection rates. While the rules are manually specified and do not cover every edge case, they capture a broad range of semantically meaningful inconsistencies. The rejection framework thus provides an interpretable and reproducible basis for benchmarking micro-level realism in synthetic datasets.

We apply this manual rejection sampling evaluation to synthetic data produced by several baselines (NVI, CP, HMM), our proposed LLMSYNTHOR, and the real dataset. We generate 10,000 records for each model to compute the rejection rates. We report both household- and person-level rejection rates, as well as per-rule rejection rates. Table 9 summarizes the results, omitting checks where all methods had zero rejections (e.g., `employment_valid`, `education_valid`, `income_range`, `tenure_valid`). These omissions indicate that all models trivially satisfy categorical validity and numerical bounds, while the non-zero rules reveal where models tend to fail.

Overall, the real dataset exhibits near-zero rejection, validating that the rule set correctly captures meaningful inconsistencies while not over-penalizing valid cases. In contrast, baseline methods produce high rejection rates. For instance, NVI fails on over 40% of persons due to education-age or employment-age mismatches, and nearly all households are rejected when aggregating these failures. CP and HMM improve over NVI but still suffer from 30% or more employment-age inconsistencies, and more than half of households are rejected. LLMSYNTHOR markedly outperforms all baselines: household rejection drops to 13.3% and person rejection to 6.3%, both much closer to real data. At

| Check | NVI | CP | HMM | Real | LLMSYNTHOR |
|---|---|---|---|---|---|
| ***Overall rejection rates*** ↓ | | | | | |
| Household rejection rate | 96.8 ± 0.3 | 73.9 ± 0.6 | 57.8 ± 0.5 | 0.0 ± 0.0 | **13.3 ± 0.4** |
| Person rejection rate | 45.9 ± 0.4 | 34.1 ± 0.7 | 27.8 ± 0.6 | 0.0 ± 0.0 | **6.3 ± 0.2** |
| ***Household-level checks*** ↓ | | | | | |
| householder_type_adult_consistency | 4.5 ± 0.2 | 4.4 ± 0.3 | 15.2 ± 0.5 | 0.0 ± 0.0 | **3.8 ± 0.2** |
| persons_all_valid | 96.7 ± 0.4 | 73.5 ± 0.6 | 56.1 ± 0.7 | 0.0 ± 0.0 | **10.7 ± 0.3** |
| ***Person-level checks*** ↓ | | | | | |
| age_range | 0.1 ± 0.0 | 0.5 ± 0.1 | 0.3 ± 0.1 | 0.0 ± 0.0 | **0.0 ± 0.0** |
| race_valid | 0.0 ± 0.0 | 0.8 ± 0.1 | 0.0 ± 0.0 | 0.0 ± 0.0 | **0.0 ± 0.0** |
| employment_age_consistency | 41.6 ± 0.7 | 30.9 ± 0.6 | 24.5 ± 0.5 | 0.0 ± 0.0 | **4.4 ± 0.2** |
| education_age_consistency | 42.7 ± 0.8 | 8.1 ± 0.4 | 10.7 ± 0.5 | 0.0 ± 0.0 | **2.2 ± 0.1** |

Table 9: Rejection rates (%) by rule across methods. Checks with zero rejections in all methods are omitted for brevity.

the person level, LLMSYNTHOR reduces employment-age inconsistency to $4.4\%$ and education-age inconsistency to $2.2\%$, an order of magnitude lower than baselines. At the household level, only $10.7\%$ of households are rejected due to invalid persons, compared to 56%-97% for baselines. The only area with room for further improvement is the household size distribution, where $6.5\%$ of LLMSYNTHOR households fall outside the expected range, but this remains far less consequential than the widespread inconsistencies observed in baselines.

These results demonstrate that LLMSYNTHOR produces micro-records with substantially higher realism than baseline approaches. It not only avoids obvious contradictions (e.g., children with advanced degrees, unemployed minors with high incomes) but also maintains coherence at the household level, where failures compound across individuals. By narrowing the gap to real data across multiple dimensions, the rejection analysis provides strong evidence that LLMSYNTHOR achieves realistic micro-level semantics while simultaneously enforcing macro-level statistical alignment.

**E-commerce Transactions** Each generated micro-record is evaluated independently against a set of semantic rules. A record is rejected if it violates *any* rule. The following rules are used:

1. R1: Ultra-low price for large-item categories. Reject if product_category is Electronics or Furniture & Appliances but the price is below a category-specific minimum.

2. R2: Teenager with extremely high Food & Beverages spending. Reject if user_age < 20 and product_category = Food & Beverages with price above the pre-defined high-value threshold.

3. R3: Very senior user buying ultra-expensive electronics. Reject if user_age > 75 and product_category = Electronics with price above the high-value threshold.

These rules capture the few combinations in e-commerce transactions that are unambiguously implausible. Unlike household population data, e-commerce records rarely contain structural or factual impossibilities, so most generated samples are naturally valid. Therefore, we focus on a small set of edge-case inconsistencies to test whether generative models produce rare but semantically invalid cross-variable patterns. We evaluate all models on 2,000 generated records and report total rejection counts, rejection rates, and per-rule breakdowns.

Table 10 summarizes the realism evaluation across all baselines and our method. The real dataset serves as a reference: it exhibits only $0.6\%$ rejection, indicating that the rule set correctly identifies inconsistent cases without over-penalizing valid records.

Because e-commerce transactions contain few hard semantic constraints, our realism evaluation focuses on three soft plausibility checks that capture rare but meaningful inconsistencies. Within this setting, models trained on micro-records still exhibit noticeably higher rates of these long-tail errors: CTGAN, CopulaGAN, and SPADA generate between 4-5% implausible records, often driven by unusually low big-ticket prices or unusually high expenditures by young or very senior users. TVAE and TabSyn perform more conservatively, though still above the real-data baseline. In-context

| Methods | R1 (%) ↓ | R2 (%) ↓ | R3 (%) ↓ | Rej. (%) ↓ |
|---|---|---|---|---|
| TVAE | $1.6 \pm 0.1$ | $0.1 \pm 0.1$ | $0.1 \pm 0.0$ | $1.8 \pm 0.1$ |
| CTGAN | $2.9 \pm 0.1$ | $1.9 \pm 0.1$ | $0.3 \pm 0.0$ | $5.1 \pm 0.1$ |
| CopulaGAN | $2.1 \pm 0.1$ | $2.1 \pm 0.1$ | $0.6 \pm 0.0$ | $4.8 \pm 0.1$ |
| GReaT | $3.5 \pm 0.2$ | $0.4 \pm 0.0$ | $0.0 \pm 0.0$ | $3.9 \pm 0.1$ |
| TabSyn | $0.7 \pm 0.1$ | $0.1 \pm 0.0$ | $0.1 \pm 0.0$ | $0.8 \pm 0.1$ |
| Spada | $3.6 \pm 0.1$ | $0.2 \pm 0.0$ | $0.4 \pm 0.0$ | $4.1 \pm 0.1$ |
| IPF | $0.6 \pm 0.3$ | $0.5 \pm 0.2$ | $0.7 \pm 0.0$ | $1.8 \pm 0.0$ |
| LLM-ICL | $0.0 \pm 0.0$ | $0.0 \pm 0.0$ | $0.0 \pm 0.0$ | $0.0 \pm 0.0$ |
| GT (Real Data) | $0.5 \pm 0.1$ | $0.0 \pm 0.0$ | $0.1 \pm 0.0$ | $0.6 \pm 0.1$ |
| LLMSYNTHOR | $\mathbf{0.5 \pm 0.1}$ | $\mathbf{0.0 \pm 0.0}$ | $\mathbf{0.0 \pm 0.0}$ | $\mathbf{0.5 \pm 0.1}$ |

Table 10: Rule-wise rejection rates (%) and total rejection rate across 5 runs of 2,000 generated e-commerce transactions per method.

prompting method LLM-ICL avoid such errors almost entirely but tend to sacrifice coverage or diversity. Our method, LLMSYNTHOR, achieves a rejection rate of only $0.5\%$, essentially matching the real dataset while relying solely on macro-level statistics. Although the rules used here are soft rather than strict validity constraints, the consistently low rejection rate of LLMSYNTHOR across all three checks indicates that discrepancy-guided macro-to-micro generation effectively avoids extreme-value artifacts and maintains stable micro-level plausibility.

## C.3 MOBILITY SYNTHESIS

**Additional Mobility Visualizations.** Figure 18 shows a detailed comparison of spatial-temporal flow intensity between real and synthetic data across seven time intervals throughout the day. Each map captures the aggregate origin and destination activity within the Tokyo metropolitan area during a specific time window. The synthetic data successfully preserves major spatial patterns such as morning and evening commute flows, while also capturing temporal variations in trip density. This highlights the model's ability to maintain realistic spatiotemporal dynamics.

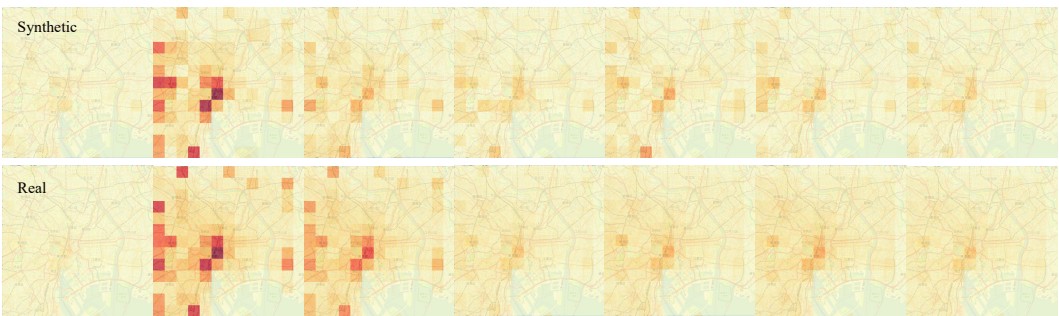

Figure 18: Spatial-temporal flow intensity maps for real and synthetic mobility data across seven time intervals: `0-6, 6-9, 9-12, 12-14, 14-17, 17-20,` and `20-24`. Each map shows trip density aggregated by region to visualize commuting and activity patterns over the day.

Figure 19 presents additional joint distribution visualizations across key mobility attributes. The left plot illustrates the correlation between transport mode and travel distance, showing that synthetic records preserve realistic distance-dependent mode preferences (e.g., longer trips by car). The middle and right plots show marginal distributions for transport modes and time intervals, further confirming strong alignment between real and synthetic mobility behavior. Together, these results demonstrate that LLMSYNTHOR can faithfully reproduce both spatial structure and behavioral signals critical for urban simulation and mobility planning.

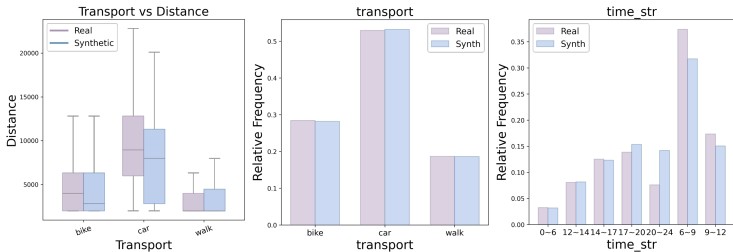

Figure 19: Distribution comparisons for mobility variables.

# D  THEORETICAL ANALYSIS

## D.1  CONVERGENCE GUARANTEE

Let $\mathcal{T}_\mathcal{V}$ be the full contingency tensor representing the joint distribution of all variables in $\mathcal{V}$. The target macro-statistics $\mathcal{S}_{\text{target}}^{\mathcal{C}}$ and synthetic macro-statistics $\mathcal{S}_{\text{synth}}^{(t)}$ each describe a corresponding contingency tensor, which can be viewed as a projection of the macro-statistics onto $\mathcal{T}_\mathcal{V}$. We denote these projections as $\mathcal{T}_{\text{target}}^{\mathcal{C}}$ and $\mathcal{T}_{\text{synth}}^{(t)}$, with $\mathcal{T}_{\text{synth}}^{(0)} = \mathbf{0}$.

**Theorem 1** (Contraction of the Mean Macro-Discrepancy). *Let* $\Delta_\star^{(t)} \equiv Q_\star\big(\mathcal{T}_{\text{synth}}^{(t)}, \mathcal{T}_{\text{target}}^{\mathcal{C}}\big)$ *with* $Q_\star(x,y) = \varphi(x-y)$ *for any norm* $\varphi$ *(positively homogeneous). Define the expected discrepancy:*

$$\bar{\Delta}^{(t)} \equiv Q_\star\big(\mathbb{E}[\mathcal{T}_{\text{synth}}^{(t)}], \mathcal{T}_{\text{target}}^{\mathcal{C}}\big),$$

$$then \quad \bar{\Delta}^{(t+1)} \leq (1-\eta_t)\,\bar{\Delta}^{(t)}, \quad hence \quad \bar{\Delta}^{(t)} \xrightarrow[t\to\infty]{} 0 \ if \ \sum_{t=0}^{\infty} \eta_t = \infty, \quad \eta_t \in (0,1].$$

*This ensures that the synthetic macro-statistics converge to the target macro-statistics as the number of iterations increases. The proof is detailed below.*

***Proof of Theorem 1.*** Write $Q_\star(x,y) = \varphi(x-y)$ with $\varphi$ a norm and let $y = \mathcal{T}_{\text{target}}^{\mathcal{C}}$. Taking the unconditional expectation in the unbiased update yields:

$$\mathbb{E}[\mathcal{T}_{\text{synth}}^{(t+1)}] = \mathbb{E}[\mathcal{T}_{\text{synth}}^{(t)}] + \eta_t \cdot \Big(y - \mathbb{E}[\mathcal{T}_{\text{synth}}^{(t)}]\Big) = (1-\eta_t)\,\mathbb{E}[\mathcal{T}_{\text{synth}}^{(t)}] + \eta_t\,y, \quad \eta_t \in (0,1].$$

Hence, by the positive homogeneity of $\varphi$,

$$\bar{\Delta}^{(t+1)} = \varphi\left(\mathbb{E}[\mathcal{T}_{\text{synth}}^{(t+1)}] - y\right) = \varphi\left((1-\eta_t)\left[\mathbb{E}[\mathcal{T}_{\text{synth}}^{(t)}] - y\right]\right) = (1-\eta_t)\,\bar{\Delta}^{(t)}.$$

Iterating this gives:

$$\bar{\Delta}^{(t)} \leq \bar{\Delta}^{(0)} \prod_{k<t}(1-\eta_k),$$

which vanishes if $\sum_k \eta_k = \infty$. $\qquad\qquad\square \qquad\qquad\qquad\qquad \square$

**Remark (How $Q_\star$ relates to our discrepancy measure $Q$).** In our implementation, $Q(\cdot)$ serves as a practical and actionable interface that directly attributes the directed discrepancy (e.g., "bike: +30%, car: -20%"), which can be easily interpreted by the LLM. This differs from the more abstract $Q_\star(\cdot)$, which represents the underlying discrepancy (e.g., weighted Total Variation Distance, TVD) used in our theoretical analysis. The function $Q(\cdot)$ is designed to provide a clear discrepancy signal, guiding the LLM in its generation of data.

The prompts derived from discrepancy signals effectively ground the LLM to make adjustments in the generated synthetic data in a manner that aligns with the expected unbiased update toward the target macro-statistics $\mathcal{S}_{\text{target}}^{\mathcal{C}}$. This process works iteratively, where each new batch of synthetic records is added to the dataset to reduce the discrepancy between the target and synthetic distributions. Specifically, the update rule can be viewed as $\mathcal{D}_{\text{synth}}^{(t+1)} = \mathcal{D}_{\text{synth}}^{(t)} \cup \mathcal{D}_{\text{batch}}^{(t)}$, where $\mathcal{D}_{\text{batch}}^{(t)}$ is a corrective batch of records generated to address the discrepancies in the data distribution.

The combination of these two factors (i) the discrepancy signal derived from $Q(\cdot)$ and (ii) the iterative data augmentation process forms a surrogate optimization process that, while not explicitly minimizing $Q_\star(\cdot)$, achieves the same end result. Thus, by only adding data (rather than directly modifying the underlying distribution), our framework can converge to the target distribution, as guaranteed by the contraction property in the theoretical analysis. This enables efficient alignment of the synthetic data with the target macro-statistics over successive iterations, as demonstrated by the empirical convergence results in Appendix C.1.

# E PROMPTS

This section provides the exact prompts used in our framework to guide the LLM during proposal generation and variable dependency inference. The first prompt, $p_{proposal}$, instructs the LLM to generate sampleable generation plans that align with target macro-statistics. The second prompt, $p_{copula}$, is used to elicit joint dependency among variable combinations by treating the LLM as a nonparametric copula. Both prompts are designed to be type-agnostic and modular, enabling effective alignment across heterogeneous datasets.

## E.1 LLM PROPOSAL SAMPLING

---

**$p_{proposal}$: LLM Proposal Sampling**

```
## Output Format:
```json
{{
  "n_proposals": n,
  "proposal1": {{
    "reason": "..." ,
    "proposal": "...",
    "num": n1
  }}
  "proposal2": {{
    "reason": "...",
    "proposal": "...",
    "num": n2
  }},
  "..."
}}
```

## Information:

**Joint Guidance:** `{joint_guide}`
**Marginal Guidance:** `{marginal_guide}`
**Variables:** `{data_desc}`

---

## Rules:
Create no more than {n_proposals} proposals totaling {n_samples} samples:
1. Each proposal must follow Joint and Marginal Guidance, do not improvise beyond the
provided Guidance.
2. The reason for each proposal should explain the realistic meaning of this proposal
and how it follows the provided Guidance by referencing frequencies one by one.
3. 'min', 'max', 'category' must use the actual values mentioned in **Variables**:
Categorical variables must be a single valid candidate string (case-sensitive), and
numerical variables must be a list of two integer or float numbers (e.g., [3.0, 5.1]).
4. Do not include multiple ranges or categories per variable within a single proposal.
5. Structure of generated data must match the **Output Format**, and all variables must
be included.
6. If a variable value has a high frequency, that value should be selected in multiple
proposals.
7. Most generated samples should, as much as possible, prioritize satisfying components
that are common across the Guidances, and the \num" for each proposal should be
determined based on the frequencies specified in the Guidance.
8. Only list generation proposals; do not generate data or add extra text.
9. Do not return an empty JSON. Do not use escaped characters such as \\r\\n, \\t, or
\". Do not include any comments, markdown formatting, or explanatory text.

---
Now return a pure, valid, and non-empty JSON in English that can be directly parsed by
json.loads() in Python
```

---

In our implementation, each proposal specifies a well-defined distribution for every variable. For discrete variables, the proposal directly assigns a single valid category value. For continuous variables, the proposal gives an explicit value range (for example, [3.0, 5.1]), from which values can be sampled uniformly or with another simple scheme (e.g., using numpy in Python). The num field in each proposal specifies how many records to generate from that proposal, enabling the LLM to allocate record counts across proposals based on the frequencies and guidance provided. This mechanism lets the LLM actively plan for diversity and statistical alignment among the generated records by

balancing the number and distribution of proposals. Importantly, to encourage transparent and high-quality reasoning, we employ chain-of-thought (Wei et al., 2022) prompting: the LLM is instructed to explicitly explain the rationale behind each proposal, referencing frequency statistics and the provided guidance step by step. This makes the proposal process more interpretable and reliably aligned with the specified constraints.

Importantly, this proposal format is just one practical instantiation of LLM Proposal Sampling. The framework is highly extensible. A proposal's "distribution" could be defined not just as value ranges or categories, but as executable code, tool calls, or pointers to external generators (such as diffusion models with ControlNet for images, or other LLM-based agents for specialized content). This flexibility allows LLM Proposal Sampling to serve as a universal, high-level distributional controller, guiding external generators or hybrid pipelines toward statistically faithful and scenario-aligned synthetic data, regardless of data type or target domain.

### E.2 LLM AS A COPULA FOR VARIABLE DEPENDENCY INFERENCE

---

**$p_{copula}$: LLM as a Copula for Variable Dependency Inference**

```
## Information
**Variables:** ```{data_desc}```

## Output Format:
{{
  "1": ["var1", "var2", ...],
  "2": ["var1", "var2", ...],
  ...
}}

### Example Output JSON:
{{
  "1": ["Temperature", "Humidity"],
  "2": ["Humidity", "WeatherCondition"],
}}

---
Your task is to extract **at most {n_joints}** correlated variable groups based on the
**Variables** summaries and present them in JSON format.
1. Ensure each group contains two or more variables.
2. Format the correlated variable groups according to the **Output Format**.
3. Output must be valid JSON.
```

---

# F RUNNING EXAMPLES

## F.1 EXAMPLE OF A SINGLE ITERATION

This subsection provides a concrete example to show how each iteration improves the alignment between synthetic and target distributions.

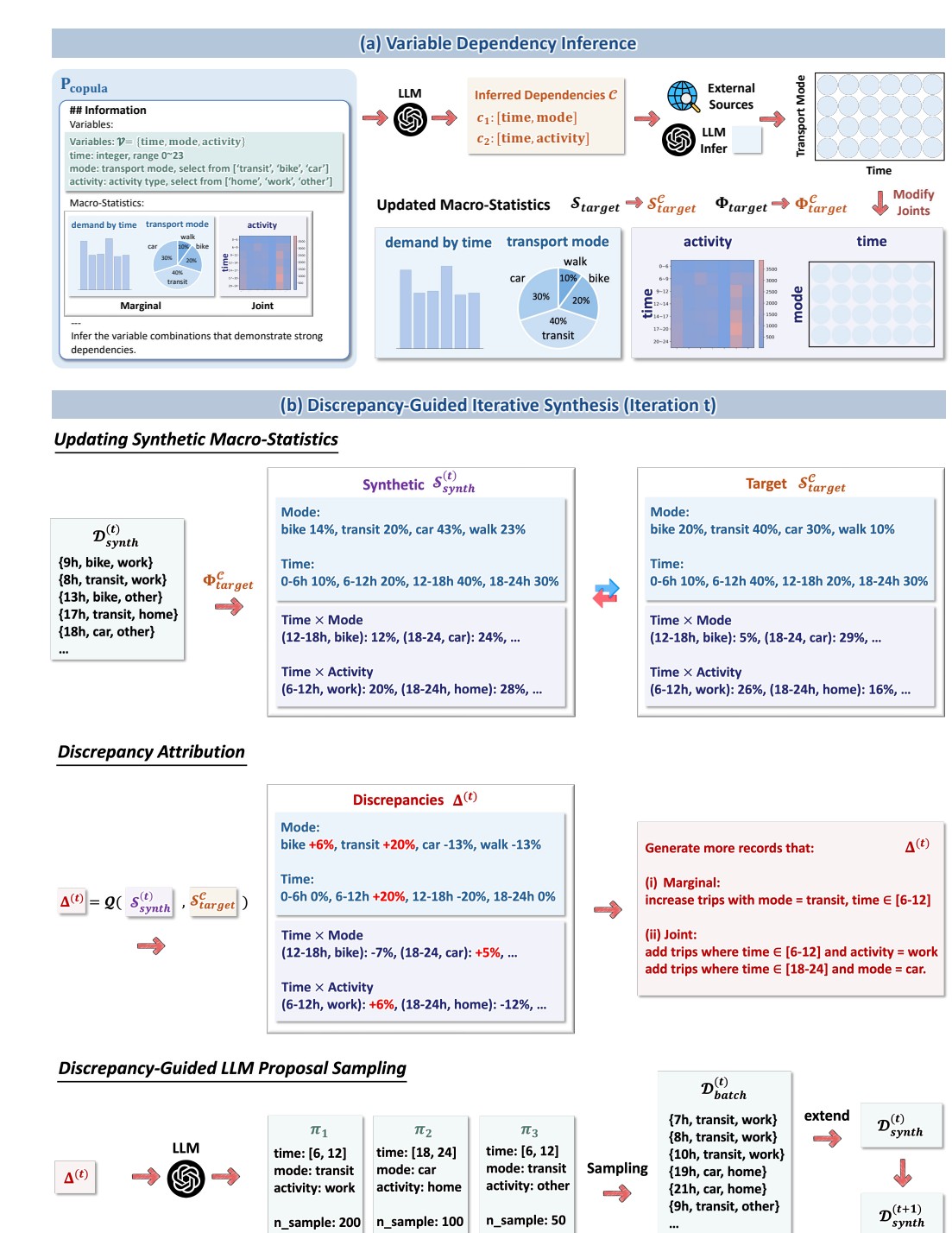

Figure 20: A simplified running example of discrepancy-guided iterative synthesis.

## F.2 DISCREPANCY-GUIDED GENERATION

This subsection explains how the model identifies the most significant discrepancies in each iteration and samples accordingly, thereby guiding the generation of micro-records to progressively reduce the gap between the synthetic and target distributions.

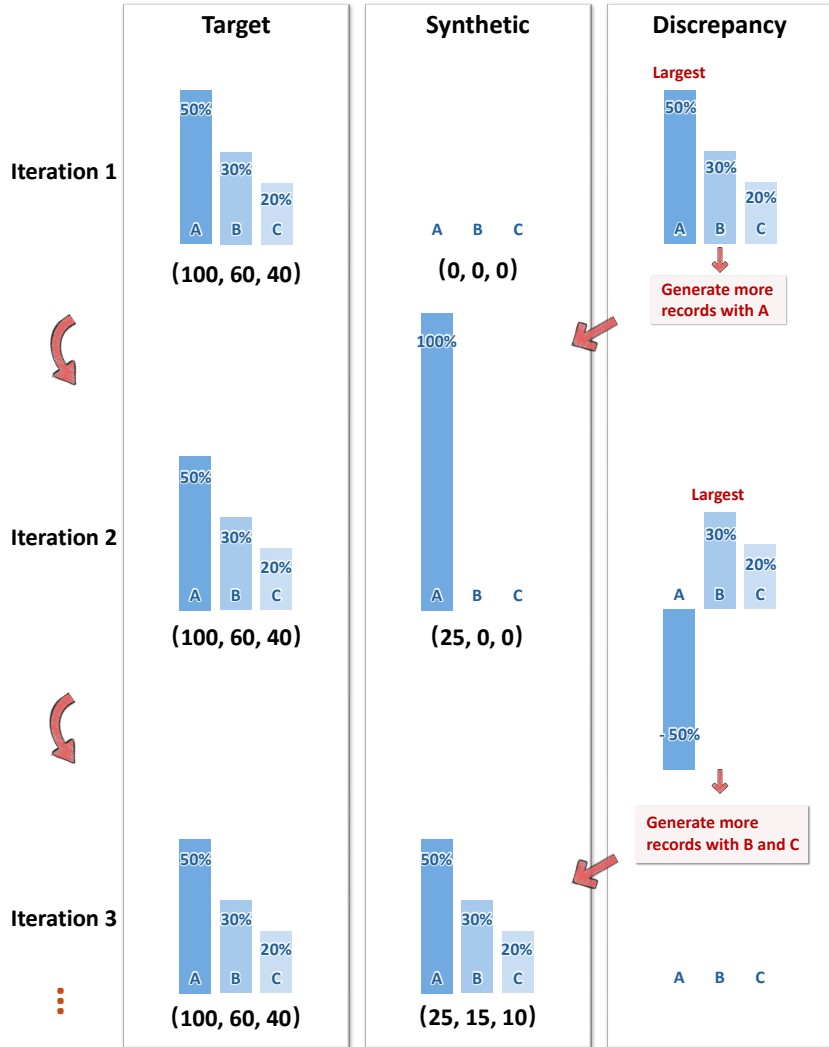

Figure 21: **Discrepancy-guided Iterative Synthesis.** The figure illustrates how discrepancies between the target and synthetic distributions guide the generation of micro-records. In each iteration, the model samples from the top discrepancies (highlighted in red) to determine where more records should be generated. In iteration 1, the synthetic dataset is initialized as $(0, 0, 0)$, and the model generates more records with category A to match the target distribution. In subsequent iterations, discrepancies for categories B and C are addressed by generating additional records, improving statistical alignment with the target.

