# OpenReview forum: "LLMSynthor: Macro-Aligned Micro-Records Synthesis with Large Language Models"
_ICLR.cc/2026/Conference — Submitted to ICLR 2026_

### Official Review · Reviewer_Xt7k · 2025-10-29

**Soundness:** 1
**Presentation:** 1
**Contribution:** 2
**Rating:** 2
**Confidence:** 4

**Summary:**

This work aims to generate realistic individual-level records that follow pre-specified population-level statistics. Individual-level records are much harder to collect than population-level statistics, but are useful to expert practitioners in domains like public health or urban planning. The ability to simulate realistic data at the micro-level while following macro-level statistics would help bridge this gap.

The authors propose to tackle this problem by using an LLM to guide the generation of individual-level records while steering the generation based on the discrepancy with high-level statistics. To improve generation efficiency, they implement “generation plans” where the LLM doesn't generate records directly but instead creates joint distributions that can be sampled from efficiently.

The problem is interesting and well motivated, but the paper suffers from several flaws in its method, experimental validation and presentation.

**Strengths:**

- The problem of micro-level generation following macro-level statistics is interesting
- Efforts to improve and evaluate the practical downstream utility in the method (producing generation plans for faster records generation) and experimental results (prediction of derived variables using synthesised data) are appreciated.

**Weaknesses:**

Major:

- W1: the method relies heavily on the LLM for variable dependency inference and micro-level generation plans synthesis. This leads to several concerns in terms of realism, bias and auditability, detailed below.
    - a. the realism of the generated records hinges on the LLM’s ability, but LLMs are notoriously prone to hallucinations. Furthermore, matching the macro-level statistics in aggregate does not guarantee data realism at the low-level. The realism of the generated low-level records should be thoroughly demonstrated experimentally, which is not the case currently (cf W4).
    - b. LLM generation might introduce bias at both at the variable dependency inference stage (ie the model might group spuriously correlated variables) and micro-level generation stage levels (ie the actual joint distribution between selected variables). This is partly noted by the authors in the discussion (l.459-460), but should be acknowledged and discussed beyond the misalignment with target statistics.
    - c. Auditability —ie providing practitioners information to understand how the micro-records were generated— is essential to mitigate the realism and bias pitfalls previously mentioned. This is not possible here since the micro-records are generated directly by the LLM (via the generation plans), which operates as a black-box.
- W2: In Section 3.2, the authors state that “If an inferred dependency lacks corresponding data, practitioners are encouraged to collect additional statistics, if unavailable, the LLM can approximate the joint distribution using existing macro-statistics” (l.215). If I understand correctly this means that if the LLM proposes a variable subset that is not exactly available in the macro-statistics S_target, then the method doesn’t work unless the practitioners collect additional statistics. This is a major limitation for the practical applicability of the method. Meanwhile, the claim that LLM can approximate the joint distribution itself (l.215-216) is unsubstantiated.
- W3: The discrepancy-guided iterative synthesis scheme (Section 3.3) means that generated low-level records are not i.i.d, since records from time t+1 depend on previously generated records at times ≤t (via the discrepancy signals \Delta^t). I would expect that i.i.d data is a common assumption in low-level records —as noted by the authors l.464.
- W4: A thorough evaluation of the realism of generated records is missing from the experimental results. Currently this is only briefly mentioned in Appendix C.3 and limited to illustrative examples. As mentioned in W1, the realism of the generated records is by no means guaranteed by the method and should therefore be extensively assessed experimentally. One way to evaluate this could be to use a dataset where ground-truth micro-level records are available, run the method based only on the macro-level statistics, and measure the similarity between ground-truth and generated low-level records distributions at various granularity levels (not just at the macro-level).
- W5: In Section 4.2, the comparison to low-level generation baselines is unfair since only LLMSynthor has access to the macro-level statistics. A fairer comparison with these baselines would be in terms of the realism of the generated low-level records (cf W4). Fairer baselines for macro-level statistics alignment include direct prompting of the LLM (a form of ablation experiment on the proposed method, eg without discrepancy-guided iterative sampling), and simulator-based methods (eg [1]).
- W6: Figure 1 and 2 are illegible. I had a lot of trouble following them even after getting a good grasp of the method from the text. I would recommend updating these figures to make them less cluttered and improving clarity.
- W7: the related work section is missing simulator-based methods, such as [1], which are highly relevant to this problem.

Minor:
- The two definitions in Section 3 are informal. They could be made sharper by separating the core definition to discussion points around it.
- Typos: problem in sentence l.200; missing period l.409

[1]: Holt et al, “G-sim: Generative simulations with large language models and gradient-free calibration.” (ICML’2025)

**Questions:**

- l.216: “The aggregation operator is then updated..” - what does this update look like in practice?
- What if variables in one of the c_k corresponds to a subset of a joint statistics available in the macro-records (eg macro records contain joint statistics over variables (x,y,z), but one c_k=(y,z))? Can we marginalise over the non-included variables?

---

> ### Author Response · Authors · 2025-11-20
> **Overview**
>
> We are grateful to the reviewer for their time and for providing such a comprehensive and detailed review.
>
> We sincerely believe that your rigorous questioning and insightful suggestions have significantly elevated the quality and clarity of our manuscript. To fully address the depth of your concerns, we have executed a substantial revision of the paper:
>
> - **Expanded Baselines**: We have incorporated additional baseline comparisons, now reported in **Table 1 and Table 2.**
> - **Rigorous Realism Evaluation**: We have conducted a quantitative evaluation of record realism based on domain constraints. As suggested, we have moved these key results to the **main text (Table 4)** and provided a detailed methodology in **Appendix C.2.3.**
> - **Clarity & Definitions**: We have meticulously refined the mathematical **definitions(1 & 2)** for precision and redesigned **Figure 1** and **Figure 2** to maximize readability and logical flow.
> - **Updated Related Work**: We have revised the Related Work section to better contextualize our contributions within the latest literature.
> - **LLM-Specific Analysis**: Addressing your concerns regarding the generative engine, we have added comprehensive experiments and discussions on **diversity, bias mitigation, and auditability**, which are now featured in **Figure 5 of the main text**.
>
> We have carefully addressed your comments below. We hope these extensive new experiments and revisions satisfactorily resolve your concerns, and we look forward to further discussion.

---

> ### Author Response · Authors · 2025-11-20
> **(W1a,b & W4) Response to W1a,b & W4: Quantitative Assessment of Micro-Level Realism and LLM-Induced Bias**
>
> # W1a & W4: Micro-Record Realism
>
>
> We thank the reviewer for raising these important concerns. We fully agree that (i) LLMs may hallucinate, and (ii) macro-level alignment alone does not guarantee micro-level realism. These are exactly the issues addressed in our expanded realism evaluation.
>
> Please refer to **General Response 2 (GR2: Realism Evaluation)**, where we now include:
> - A quantitative realism study on the household (population) dataset, a domain with rich semantics and clear plausibility rules.
> - A complementary realism evaluation for the e-commerce setting, using rule-based rejection checks (R1–R3).
> - Results showing LLMSynthor achieves very low rejection rates, close to full-supervision baselines, confirming that hallucinations and implausible samples are rare in practice.
>
> These results demonstrate that the realism of LLMSynthor’s generated micro-records is quantitatively validated, and the framework remains robust despite relying only on macro-level supervision.
>
> ---
>
> # W1b: LLM-Induced Bias
>
> We appreciate the reviewer’s thoughtful concern regarding potential biases introduced during dependency inference and micro-level generation. Below we clarify the mechanism and provide new quantitative bias results to directly address this point.
>
> In LLMSynthor, the LLM is responsible for grouping variables to guide joint constraint enforcement, not to dictate how the joint distribution is generated. Crucially, **grouping only controls which variables are aligned together** during each proposal step. It does not introduce new dependencies.
> - If the grouping is too narrow, some joint dependencies may initially be treated independently. This slows convergence but does not create spurious correlations.
> - If the grouping is too broad, multiple macro constraints are enforced together. This may accelerate convergence, but the process is still fully grounded in external macro supervision.
>
> Importantly, LLMSynthor uses iterative discrepancy correction, repeatedly comparing generated data to the target macro statistics. This mechanism guarantees that any deviation measured by macro-statistics is progressively corrected, ensuring that the system cannot drift arbitrarily.
>
> **New Evaluation: Conditional Bias (Figure 5, updated manuscript)**
> To directly quantify whether any biased or spurious dependencies are introduced, we now include conditional JS divergence evaluations on key behavior-demographic pairs. Specifically, we measure:
> - P(product_category | age)
> - P(product_category | gender)
> - P(price | product_category)
> - P(payment_method | location_tier)
>
> This conditional JS divergence explicitly detects:
> - Spurious correlations not supported by real data,
> - Missing or diluted dependencies,
> - The population skews that marginal metrics cannot reveal.
>
> As shown in **Figure 5** in the updated manuscript, the plot shows how conditional JS divergence decays as the number of synthetic records increases. Across all tested variable pairs, we observe that:
> - All curves converge toward near-zero divergence, indicating high alignment between synthetic and real conditional distributions.
> - The initial bias varies, but rapidly decreases with more samples due to iterative correction.
> - Even the more complex dependency P(price | product_category) stabilizes well below 0.05 divergence after 1,000 samples.
>
> These results validate that LLMSynthor:
> - Does not introduce artificial dependencies via grouping,
> - It is robust to initial misalignment,
> - And faithfully recovers realistic conditional relationships via macro-level supervision and iterative refinement.

---

> ### Author Response · Authors · 2025-11-20
> **(W1c & W2) Response to Auditability & Model Transparency and Robustness to Incomplete Data**
>
> # W1c: Method Auditability
>
>
> We thank the reviewer for emphasizing the importance of auditability. We fully agree that understanding how micro-records are generated is essential for safe deployment.
>
> 1. **Comparative Auditability**: While we acknowledge that the internal weights of an LLM are complex, **LLMSynthor significantly improves auditability compared to mainstream deep generative baselines** (e.g., GANs, VAEs, or Diffusion models).
>
> - Baselines: In models like CTGAN or TabSyn, the generation process involves sampling from a high-dimensional latent space via opaque neural transformations. The "reasoning" for why a specific record was generated is mathematically untraceable for a human practitioner.
> - LLMSynthor: In contrast, our framework introduces an explicit, human-readable intermediate layer: the Proposal. Every generated record can be traced back to a specific, interpretable instruction (e.g., "Generate a household with High Income in Urban Area"). This allows practitioners to inspect what the model was trying to generate before inspecting the final output, providing a level of semantic transparency unavailable in end-to-end neural baselines.
>
> 2. **Constrained Generation vs. Free Generation**: It is also important to clarify that the LLM in our framework does not operate as an unconstrained black box. It is strictly **bound by the iterative discrepancy feedback loop**. The generation is explicitly steered by external, measurable statistical signals. This means the "cause" of a generated record is not just the LLM's internal state, but the visible need to reduce a specific statistical error (e.g., "We need more high-income samples to match the census"). This causal link makes the system behavior more predictable and debuggable.
>
> 3. **Path Toward Mechanistic Interpretability** (Future Work): We agree that fully decoding the step-by-step internal processing of LLMs remains an open challenge for the entire field. However, using LLMs as backbones allows us to benefit from rapid advancements in mechanistic interpretability. Recent works [4] have begun to develop white-box methods to trace how LLMs construct outputs. Because LLMSynthor is compatible with open-source models (as demonstrated with Qwen-7B/3B in our new experiments), these emerging interpretability tools can be directly applied to our framework in future work to further demystify the micro-generation process.
>
> ---
>
> # W2: Robustness to Incomplete Macro-Statistics
>
> We thank the reviewer for this insightful scenario, which has high practical relevance. In real-world settings, complete macro-statistics may indeed be unavailable. We appreciate the opportunity to clarify the framework's behavior and demonstrate its robustness through new experiments.
> 1. **Clarification**: The Framework Does Not "Break" First, we clarify that LLMSynthor remains fully functional even if joint macro-statistics for specific inferred groups are missing.
> - Mechanism: If a specific dependency group lacks corresponding macro-supervision, the model simply omits the discrepancy computation for that specific group.
> - Impact: This does not halt the generation. Instead, the model falls back to the LLM's inherent semantic priors to approximate that distribution, while still being constrained by other available statistics.
>
> To validate this "approximation" capability, we conducted a **"Hold-out Validity Test"**. We deliberately withheld ground-truth macro-statistics for specific variables (Age, Gender, Location, one by one) and compared LLMSynthor against LLM-ICL, which relies purely on priors. As shown in the table below, LLMSynthor significantly outperforms the baseline even when specific supervision is removed:
>
> | Methods           |    v_A Was   |    v_G Tvd    |    v_L Tvd    |    v_C Tvd    |     v_X Was    |    v_M Tvd    |
> |-------------------|:------------:|:-------------:|:-------------:|:-------------:|:--------------:|:-------------:|
> | LLM-ICL           | 26.26 ± 3.12 | 0.040 ± 0.006 | 0.413 ± 0.038 | 0.347 ± 0.042 | 371.87 ± 21.54 | 0.175 ± 0.027 |
> | Ours (macro-only) |  4.06 ± 0.58 | 0.028 ± 0.004 | 0.012 ± 0.003 | 0.066 ± 0.006 |  92.94 ± 7.91  | 0.045 ± 0.009 |
>
> We sincerely thank the reviewer for raising this point, as it provided us with the opportunity to demonstrate a distinct advantage of our framework. While the total absence of statistical guidance would indeed fundamentally constrain the accuracy of any synthesis method. The results confirm that unlike baselines that fail even without micro-records, LLMSynthor can effectively combine available macro-statistics with LLM priors to generate high-quality outputs even when specific statistics are missing. Thus, we view this flexibility not as a limitation, but as a unique strength that allows for robust synthesis in data-sparse environments.
>
> ---
>
> ## References
>
> [4] Gao, Leo, et al. "Weight-sparse transformers have interpretable circuits". arXiv preprint, 2511.13653

---

> ### Author Response · Authors · 2025-11-20
> **(W3,5,6,7 & Minors & Q1,2)  Sampling Assumptions, Baseline Fairness, and Presentation Improvements**
>
> # W3: Clarification on the I.I.D. Assumption
>
> We thank the reviewer for this insightful observation. It is correct that our iterative generation scheme introduces coupling between records across steps via discrepancy feedback. However, we do not assume that records must be strictly i.i.d., nor is this a requirement for high-quality synthesis in real-world structured domains. In fact, many real-world datasets are inherently non-i.i.d., and modeling them as such would ignore crucial structural or distributional dependencies.
>
> For example:
> - In household data, individuals within the same household are clearly non-independent (e.g., age, employment, family role are jointly structured).
> - Even in census or demographic datasets, group-level correlations and survey stratification lead to structured, non-i.i.d. sampling.
>
> Our method explicitly embraces these complexities: the discrepancy-guided synthesis process iteratively steers generation toward a global statistical target, producing a dataset that aligns with external aggregate constraints. This iterative correction does not imply biased or path-dependent outputs, it merely adjusts the overall distribution to better match macro-level expectations. The final generated records are not temporally ordered, and are meant to be interpreted as a synthetic population sample consistent with the target distribution, not as i.i.d. draws from a fixed latent generator.
>
> We therefore respectfully suggest that applying the i.i.d. standard here may not be appropriate. In fact, **strict i.i.d. sampling would likely fail to respect the complex joint dependencies encoded in macro-level constraints**, which is precisely the challenge our method addresses.
>
> ---
> # W5 & W7: Baseline Fairness, Ablation Studies, and Comparison with Simulators
>
> We thank the reviewer for raising these important concerns. As clarified in **General Response 1**, all low-level baselines (TVAE, CTGAN, GReaT, etc.) operate with privileged access to full micro-records, while LLMSynthor relies solely on macro-level constraints. This makes the comparison stricter, not more favorable, for our method. To address this, we also evaluate low-level realism directly (**see GR2**), and include additional macro-only and macro-supervised baselines, such as IPF and macro-supervised Spada, and also an LLM-ICL in context learning as suggested to provide comparisons under aligned supervision levels.
>
> Regarding **W7**, we appreciate the suggestion to include simulator-based methods. We have updated the Related Work section to incorporate and discuss these approaches, and highlight how LLMSynthor differs in its ability to synthesize structured micro-records under flexible, unconstrained macro targets, without requiring pre-specified agent behaviors or programmatic simulators.
>
> ---
> # W6 & Minors: Improvements to Figures, Mathematical Definitions, and Presentation
>
> We thank the reviewer for their careful attention to the presentation quality of our manuscript. We have addressed these points in the revised version:
> - **Figures 1 and 2**: We have carefully redesigned both figures to maximize readability, improving font sizes, layout clarity, and visual flow.
> - **Definitions 1 and 2**: We have refined these definitions to be more mathematically sound and precise, ensuring a sharper formalization of the problem setting.
> - **Typos**: We have conducted thorough proofreading and corrected all identified typos throughout the text.
>
> ---
> # Q1 & Q2: Aggregation Updates
>
> We thank the reviewer for raising this. Our dependency inference mechanism identifies meaningful subsets of correlated variables, and the corresponding discrepancy supervision is then constructed by aggregating the synthetic records over these subsets. In practice, once a dependency group (c_k) is inferred (e.g., (c_k = (y, z))), the aggregation operator is dynamically updated to compute the macro-statistics over exactly those variables from the synthetic records. This ensures that the discrepancy feedback is aligned with the dependency structure discovered by the LLM.
>
> If the available macro-statistics cover a **superset of the variables (e.g., (x,y,z)),** the discrepancy computation **automatically marginalizes** over the non-overlapping variables (in this case, x) to compute the target joint over (y,z). **This ensures that all available macro signals are leveraged when relevant**, and the supervision aligns exactly with the inferred dependency structure.
>
> This design enables flexible alignment: when macro-statistics provide richer information than strictly required, our framework still benefits from them without violating the modularity or correctness of discrepancy computation.

---

### Official Review · Reviewer_isfU · 2025-10-29

**Soundness:** 2
**Presentation:** 3
**Contribution:** 2
**Rating:** 4
**Confidence:** 2

**Summary:**

The paper presents LLMSYNTHOR, a framework for generating micro-level synthetic records that align with given macro-level statistics. It repurposes a pre-trained LLM as a macro-aware simulator that iteratively refines data generation through discrepancy feedback. The approach combines (i) LLM-based inference of variable dependencies and (ii) proposal-level sampling to improve efficiency and alignment. The paper evaluates the framework on three domains—mobility, e-commerce, and population synthesis—covering both structured and unstructured data. The authors claim the framework improves both realism and efficiency compared to standard LLM sampling, and demonstrate qualitative controllability through proposal-level generation instead of micro-record generation.

**Strengths:**

Synthesizing datasets using LLM is a good topic.

**Weaknesses:**

- The paper lacks a description of important details, such as performance metric computation and the proposal for the micro-records sampling step. In Table 2, it is unclear what the Jsd column means. For other specified metrics, it is unclear how they are computed and why those metrics make sense in this application.

- Figure 11 appears to suggest that LLM-based generation still scales poorly with dataset size, potentially far slower than purely generative baselines like GReaT, which trains once and can then sample nearly 10x times faster.

- The paper claims that LLMSYNTHOR ensures realistic micro-records, yet there is no quantitative or human validation of realism.

- All compared baselines (TVAE, CTGAN, CopulaGAN, GReaT, TabSyn, CP, HMM, NVI) are trained on full micro-record datasets, whereas LLMSYNTHOR only accesses macro statistics. Evaluation is non-comparable: the baselines operate with privileged access to individual-level data.

- Synthesizing micro-records that align with target macro-statistics is conceptually similar to synthesizing samples conditioned on target population characteristics. The paper may benefit from comparing with related work in this direction [1, 2, 3], which also explores LLM-based methods for generating population-level or behaviorally diverse synthetic data.

References:
- [1] Bui, Ngoc, et al. "Mixture-of-personas language models for population simulation." arXiv preprint arXiv:2504.05019 (2025).
- [2] Choi, Hyeong Kyu, and Yixuan Li. "Picle: Eliciting diverse behaviors from large language models with persona in-context learning." arXiv preprint arXiv:2405.02501 (2024).
- [3] Yu, Yue, et al. "Large language model as attributed training data generator: A tale of diversity and bias." Advances in neural information processing systems 36 (2023): 55734-55784.

**Questions:**

- Are there any macro-only or aggregate-aware baselines that could serve as fairer comparisons?
- How many LLM invocations are made per dataset? What is the number of micro-records per proposal?
- How does LLMSYNTHOR validate the authenticity of generated records? Are there rule-based or domain constraints to reject implausible samples (e.g., “infant working full-time”)?
- Was any human evaluation or domain-expert assessment performed to verify the realism of the synthetic records or the proposals?
Once an LLM outputs a proposal (e.g., “50 records: origin=Brooklyn, time ∈ [6,9]”), how are the individual attribute values sampled? Uniformly? Using empirical distributions?
- How interpretable and realistic are these proposals in practice? Do domain experts consider them meaningful?

---

> ### Author Response · Authors · 2025-11-20
> **Overview**
>
> We sincerely thank the reviewer for their constructive feedback, particularly for pointing out areas where our methodological details lacked clarity and for raising critical questions regarding efficiency and realism.
>
> To directly address these concerns, we have made the following significant updates to the manuscript:
>
> - **Clarification of Details**: We have thoroughly proofread the paper and added a dedicated Section B.3 in the Appendix to rigorously formalize the specific metric details and definitions that were previously ambiguous.
> - **Comprehensive Efficiency Analysis**: Addressing your concern on computational cost, we have conducted a new, extensive efficiency study. The results demonstrate that **LLMSynthor achieves superior fidelity at faster sampling speeds compared to the baseline (e.g., beating GReaT in both speed and quality at 10 iterations)**. These findings are now highlighted in **Figure 5** of the updated main text, with a full analysis provided in **Appendix C.1.**
> - **Quantitative Realism Evaluation**: To address your query on how record realism is measured, we have implemented a rigorous quantitative evaluation based on strict domain-specific semantic constraints. We have moved these key results to **Table 4** of the main text to ensure visibility, with detailed experimental setups described in the **new Appendix C.2.3.**
>
> Please find our detailed responses below. We hope these extensive revisions and additional experiments fully address your concerns, and we look forward to further discussion.

---

> ### Author Response · Authors · 2025-11-20
> **(W1,2,3,4 & Q1, 2, 3) Response to W1-W4 & Q1-Q3: Methodological Clarifications and Experimental Validity**
>
> # W1: Clarification of Evaluation Metrics
> Thank you for pointing out the need for clearer descriptions of our evaluation metrics. We apologize for the lack of detail in the initial submission.
>
> We have now added a dedicated subsection in the **Appendix (Section B.3, updated manuscript)** that defines all metrics, details their computation, and explains why each metric is appropriate for evaluating macro-level alignment, micro-level realism, and downstream utility. This new subsection ensures that the evaluation methodology is fully transparent.
>
> ---
> # W2 & Q2: Scalability Analysis and Efficiency Trade-offs
>
> We thank the reviewer for raising this important concern regarding scalability. We have conducted extensive new experiments to demonstrate that LLMSynthor achieves a superior efficiency-accuracy trade-off compared to baselines. These new results have been incorporated into **Figure 5** of the revised main text, with a comprehensive breakdown including full tables, visualization, and detailed analysis provided in **Appendix C.1**.
>
> | Methods                |  Train time |  generation time |  v_A  |  v_G  | v_L | v_C |   v_X   |  v_M  | [v_A, v_G, v_C] | [v_C, v_X] | [v_L, v_M] |
> |------------------------|:-----------:|:----------------:|:-----:|:-----:|:-------:|:-------:|:-------:|:-----:|:---------------:|:----------:|:----------:|
> |                        |             |                  |  Was  |  Tvd  |   Tvd   |   Tvd   |   Was   |  Tvd  |       Jsd       |     Jsd    |     Jsd    |
> | GReaT                  | 724.7±8.4 |    36.3±1.2    | 2.862 | 0.016 |  0.039  |  0.045  | 169.866 | 0.009 |      0.087      |    0.382   |    0.038   |
> | LLmSynthor (10 iters)  |      0      |    24.7±3.1    | 2.306 | 0.013 |  0.026  |  0.042  |  46.68  | 0.016 |      0.268      |    0.198   |    0.081   |
> | LLmSynthor (20 iters)  |      0      |    45.7±5.5    | 1.302 | 0.002 |  0.002  |  0.012  |  21.15  | 0.003 |      0.076      |    0.167   |    0.025   |
> | LLmSynthor (100 iters) |      0      |   238.6±14.3   |  1.13 | 0.002 |  0.002  |  0.010  |  12.762 | 0.003 |      0.071      |    0.134   |    0.007   |
>
> To directly address the reviewer’s concern about scalability, we performed a new runtime-accuracy study by varying LLMSynthor’s refinement iterations (10 → 20 → 100):
>
> - 10 Iterations (Speed & Quality Win): With only 10 iterations, LLMSynthor is already faster than GReaT end-to-end (sampling time: 24.7s vs. 36.3s) while simultaneously outperforming GReaT on nearly all fidelity metrics.
> - 20 Iterations (Dominant Performance): Increasing to 20 iterations allows LLMSynthor to consistently surpass GReaT across every metric, while still maintaining a competitive low runtime.
> - 100 Iterations (SOTA Fidelity): 100-iteration setting (**as used in the main paper**), LLMSynthor achieves state-of-the-art accuracy, demonstrating smooth scaling with refinement depth.
>
> Crucially, this comparison highlights a structural advantage: GReaT’s inference speed is only possible after a costly 724-second training phase. In stark contrast, LLMSynthor incurs zero training cost, adapts immediately to new datasets without retraining, and exposes a flexible accuracy-efficiency trade-off unavailable to GReaT.
>
> Overall, LLMSynthor offers a **superior pareto frontier**: it can be faster than GReaT (10 iters), more accurate than GReaT (20 iters), or maximize fidelity (100 iters). This flexibility demonstrates strong scalability rather than a limitation. As LLM inference speeds continue to accelerate, the practical value of LLMSynthor will only increase.
>
> ---
> # W3 & Q3: Quantitative Evaluation of Micro-Record Realism
>
> We thank the reviewer for raising this point. A detailed response including quantitative realism evaluation, rule-based validity checks, and cross-domain analyses is provided in our **General Response 2 on Realism Evaluation**, where we show that LLMSYNTHOR achieves realism comparable to ground-truth micro-records across both population and e-commerce settings.
>
> ---
> # W4 & Q1: Baseline Comparisons:
> We thank the reviewer for the observation. We clarify in our **General Response 1** on Baseline Comparability that all tabular generative baselines indeed require full micro-record supervision, whereas LLMSynthor operates under the strictly harder macro-only setting. This difference does not give LLMSynthor an advantage.
>
> **Regarding Q1**, we note that macro-only or aggregate-aware generative methods remain largely absent in the literature. **This gap is precisely what LLMSynthor is designed to fill. To our knowledge, LLMSynthor is the first framework that explicitly targets micro-record synthesis under macro-only supervision.**
>
> We additionally include new macro-supervised and macro-only baselines (IPF, macro-augmented Spada, and an LLM-ICL model with full macro-statistic prompts). These added results reaffirm LLMSynthor’s superiority even under strictly controlled and directly comparable evaluation.

---

> ### Author Response · Authors · 2025-11-20
> **(W5 & Q4,5) Response to Q4, Q5 & W5: Realism Validation, Interpretability, and Comparison with Related Work**
>
> # W5: Distinctions from Persona-based Simulators
>
> We thank the reviewer for highlighting these relevant and inspiring works. We agree that they represent an important parallel direction in LLM-based generation. We have incorporated a detailed discussion of these papers ([1], [2], [3]) into the Related Work section of the revised manuscript.
>
> We agree that persona-based and population-conditioned LLM generation share the high-level idea of guiding LLM outputs using coarse population descriptors. However, these approaches differ fundamentally from our objective: LLMSynthor aims to reconstruct an entire micro-level distribution that exactly aligns with arbitrary and potentially high-dimensional macro-statistics, while works such as Mixture-of-Personas, Picle, and Attributed Data Generators focus on conditioning LLMs on personas or attributes to elicit diverse textual or behavioral samples, without guaranteeing consistency with aggregate statistics or supporting iterative discrepancy correction. These methods cannot serve as baselines in our setting, as they **do not**:
> 1. Enforce macro-statistic constraints,
> 2. Generate multi-attribute structured records,
> 3. Construct population-level joint distributions.
>
> We have added a discussion of these conceptual connections and distinctions to the related-work section, and we clarify in the General Response that LLMSynthor addresses a different, and strictly harder problem of macro-constrained micro-record synthesis, which is not supported by these persona-based generation frameworks.
>
> However, we see significant potential for synergy. LLMSynthor could serve as a "statistical alignment layer" for these simulators in future work. For instance, while Mixture-of-Personas generates diverse behaviors, our discrepancy-guided mechanism could be applied on top to ensure that the aggregate distribution of those behaviors perfectly matches real-world survey data or specific demographic targets. This would combine the behavioral richness of persona-based simulation with the statistical rigor of our framework.
>
> ---
>
> # Q4 & Q5: Expert Assessment, Realism Validation, and Proposal Interpretability
>
>
> We thank the reviewer for these insightful questions regarding the verification and interpretability of our generated records. We have addressed these interconnected points below.
>
> 1. **Expert-Designed Auditing as Scalable Evaluation (Response to Q4)**: While we agree that human evaluation is valuable, large-scale manual assessment is often cost-prohibitive and prone to subjectivity. To address this, we employed a rigorous and scalable alternative: **Expert-Designed Rule-Based Auditing.** As detailed in **General Response 2** and **Table 4 (Main Text)**, we collaborated with domain knowledge to codify strict semantic constraints (e.g., "a child cannot be a primary householder,"). These rules serve as a deterministic proxy for expert assessment, allowing us to quantitatively audit the realism of thousands of micro-records. The near-zero rejection rates achieved by LLMSynthor demonstrate that it respects these expert-defined logic boundaries far better than traditional deep generative models.
> 2. **Interpretability of Proposals (Response to Q5)**: Unlike diffusion or GAN-based tabular generators, whose latent operations are opaque "black boxes," LLMSynthor is explicitly designed for interpretability.
> - Transparent Reasoning: Each proposal corresponds to a human-understandable joint constraint (e.g., origin = Brooklyn, time ∈ [6, 9] or household_type = Single Parent, income > 50k). This exposes the model's intermediate reasoning, allowing practitioners to directly inspect, audit, or even modify generation logic.
> 3. **Extensible Sampling Procedure**: Regarding the specific sampling mechanism queried in Q4: once the LLM outputs a transparent proposal, LLMSynthor samples the remaining unconstrained attributes from the proposal-conditioned distribution. In our current implementation, this defaults to **uniform sampling** within valid domains to maximize coverage. However, the framework is **fully extensible**: practitioners can easily plug in empirical distributions, domain-specific priors, or learned samplers if higher fidelity is required. Importantly, the proposal-level generation ensures that the critical constrained attributes dominate the record's semantics, providing stability and realism even under simple sampling schemes.
>
> ---
>
> ## References
>
> [1] Yang, Shuo, et al. "Doubling Your Data in Minutes: Ultra-fast Tabular Data Generation via LLM-Induced Dependency Graphs." Proceedings of the 2025 Conference on Empirical Methods in Natural Language Processing. 2025.
>
> [2] Kolenikov, Stanislav. "Calibrating survey data using iterative proportional fitting (raking)." The Stata Journal 14.1 (2014): 22-59.
>
> [3] Brown, Tom, et al. "Language models are few-shot learners." Advances in neural information processing systems 33 (2020): 1877-1901.

---

### Official Review · Reviewer_5i4z · 2025-11-02

**Soundness:** 2
**Presentation:** 4
**Contribution:** 2
**Rating:** 4
**Confidence:** 4

**Summary:**

The paper proposes a simulator that uses a pre-trained LLM for synthesizing tabular data that match given macro-statistics. They treat the LLM as a non-parametric copula to infer variable dependencies and decide which joints to control and use discrepancy-guided iterative synthesis with LLM Proposal Sampling to reduce the macro-level gaps between current synthetic aggregates and targets. Experiments on mobility, e-commerce (tabular), and population synthesis show good macro alignment and downstream utility; the paper also reports strong performance versus tabular baselines.

**Strengths:**

- The paper is very well written with illustrative figures.
- Synthetic data generation with LLMs is an important and timely problem.
- Method design feels practical. The loop is simple and likely easy to implement.
- Some limitations of LLMs are clearly discussed.

**Weaknesses:**

- Evaluation alignment vs. leakage. In section 4.2, it is not fully clear which macro-stats are used as inputs to LLMSynthor. Are they the same as evaluation metrics? If they coincide, the method could look stronger than baselines because it is explicitly optimizing those statistics, whereas baselines also model micro-level realism beyond the chosen stats.
- Baselines breadth & recency. The population-synthesis baselines (CP/HMM/NVI) appear to be outdated relative to modern diffusion/tabular-LLM generators. Similarly, some tabular synthesis, TabSyn and GReaT are included, but they seems old as well. How's your method compared to recent LLM-based tabular generator,s such as - Yang et al Doubling Your Data in Minutes: Ultra-fast Tabular Data Generation via LLM-Induced Dependency Graphs. or Long et al. LLM-TabLogic: Preserving Inter-Column Logical Relationships in Synthetic Tabular Data via Prompt-Guided Latent Diffusion.

**Questions:**

- (Line ~140) Variable types. You define micro-records over variables (discrete or continuous). Why is the taxonomy restricted to these two? Can your method be extended to synthesizing other dataset formats with texts, for example, since you're using LLMs?
-  Some citations are needed for baselines in population synthesis experiments.
- Why synthetic e-commerce? Could you replicate section 4.2 on a public retail/marketplace dataset (e.g., an Amazon open dataset) to demonstrate performance with real macro targets and natural noise? What prevents doing this today?
- Can you please report the quantitative evaluation of the realism of micro-records for the experiment in section 4.2?

---

> ### Author Response · Authors · 2025-11-20
> **Overview**
>
> We sincerely thank the reviewer for their insightful questions and for their careful observation regarding the household synthesis baselines.
>
> To strictly address your concerns and strengthen the manuscript, we have made the following major updates:
> - **Expanded Baselines**: We have incorporated the methods you recommended as baselines. The updated performance comparisons are now reported in **Table 1 and Table 2 of the revised manuscript**.
> - **Quantitative Realism Evaluation**: As requested, we have conducted a rigorous quantitative analysis of record realism. Key results have been featured prominently in the **main text (Table 4)**, with a comprehensive breakdown of the experimental setup provided in the new **Appendix C.2.3**.
> - **Real-World Comparisons**: We have further expanded our experiments to include detailed comparisons against real-world datasets to validate practical utility.
>
> Please find our detailed point-by-point responses below. We hope these extensive revisions fully address your concerns, and we look forward to further discussion.

---

> ### Author Response · Authors · 2025-11-20
> **(W1 & W2 & Q1 & Q2) Response to W1 & W2: Evaluation Fairness and Baseline Breadth and Q1 & Q2 (Methodological Scope)**
>
> We thank the reviewer for their careful attention to the rigor of our evaluation. We have addressed these interconnected concerns by expanding our baseline suite and clarifying the structural constraints of the chosen tasks.
>
>
> # W1: Evaluation Alignment vs. Leakage
>
>
> | Methods               |   Rej. %  |  Age  | Gender |  Loc. |  Cat. |  Price  | Payment | [v_A, v_G, v_C] | [v_C, v_X] | [v_L, v_M] |
> |-----------------------|:---------:|:-----:|:------:|:-----:|:-----:|:-------:|:-------:|:---------------:|:----------:|:----------:|
> |                       |           |  Was  |   Tvd  |  Tvd  |  Tvd  |   Was   |   Tvd   |       Jsd       |     Jsd    |     Jsd    |
> | TabSyn (**micro**)        | 1.9 ± 0.3 | 1.196 |  0.012 | 0.007 | 0.045 |  114.12 |  0.028  |      0.083      |    0.237   |    0.027   |
> | Spada (**macro + micro**) | 4.1 ± 0.9 |  1.42 |  0.013 | 0.003 | 0.016 |  54.603 |  0.025  |      0.085      |    0.225   |    0.103   |
> | IPF (**macro-only**)      | 3.9 ± 0.6 |  5.27 |  0.018 | 0.026 | 0.012 | 139.957 |  0.035  |      0.117      |    0.504   |    0.113   |
> | LLM-ICL (**macro-only**)  | 0.1 ± 0.0 | 20.61 |  0.026 | 0.024 | 0.441 | 279.743 |  0.163  |      0.437      |    0.591   |    0.476   |
> | Ours (**macro-only**)     | 0.3 ± 0.1 |  1.13 |  0.002 | 0.002 | 0.022 |  12.762 |  0.003  |      0.071      |    0.134   |    0.007   |
>
>
> Regarding W1: A detailed clarification is provided in **General Response 1** (also the table above), where we rigorously define the relationship between training information and evaluation metrics. We demonstrate why LLMSynthor does not benefit from privileged information and have introduced new **macro-supervised and macro-only** baselines (e.g., **IPF, Spada, LLM-ICL**) to ensure a strictly fair comparison.
>
>
> # W2: Baseline Breadth & Recency
>
> Regarding W2: We appreciate the suggestions and have updated the manuscript to clarify the baseline landscape for each task:
> - Population Synthesis (Hierarchical Data): This task involves generating unstructured micro-records (e.g., households containing varying numbers of persons with internal dependencies). To the best of our knowledge, no existing diffusion-based or tabular-LLM generator is architecturally capable of handling such hierarchical data. Consequently, the CP, HMM, and NVI models included in our evaluation represent the actual state-of-the-art for this specific domain. If the reviewer is aware of modern deep generative models capable of variable-length, multi-entity synthesis, we would be eager to include them.
> - E-commerce (Tabular Data): For the standard tabular setting, we have incorporated the reviewer's feedback on recent LLM-based generators. LM-TabLogic is not publicly available, Spada provides released code, and we have now added it as a baseline (**see General Response 1**).
>
> Overall, we expanded the baseline suite where possible and clarified why population synthesis inherently lacks comparable modern baselines.
>
> ---
>
> # Q1: Variable types
>
>
> Thank you for the question. We follow the standard tabular-data taxonomy used in prior synthesis work (e.g., TabSyn, TVAE, CTGAN, GReaT), where variables are categorized into discrete and continuous types because these are the **primary statistical forms** required for defining macro-level aggregates (histograms, means, quantiles, joint tables, etc.).
> This choice is not a limitation of LLMSynthor, but rather reflects the scope of the datasets used in our experiments and is consistent with common practice.
>
> Regarding text fields, we agree that LLMs are naturally capable of generating unstructured content. While LLMSynthor was not designed as a general-purpose text generator, the framework can in principle serve as a macro-level controller for text synthesis,i.e., using discrepancy feedback to steer text distributions toward aggregate targets (topic proportions, sentiment ratios, style distributions, etc.). This is an interesting direction, but it is out of scope for the current paper, whose primary goal is generating structured micro-records aligned with macro-statistics. We have added a clarification in the revision.
>
> # Q2: Citations for baselines in population synthesis experiments.
>
> Thank you for pointing this out. We have added the appropriate citations for all population-synthesis baselines (CP, HMM, NVI) in Section B.2.2 in the revised version.

---

> ### Author Response · Authors · 2025-11-20
> **(Q3 & Q4) Response to Q3 & Q4: Real-World Applicability and Quantitative Realism Evaluation**
>
> # Q3: Justification for Synthetic Data and Real-World Applicability
>
> Thank you for the thoughtful question. The main reason we use a synthetic e-commerce dataset is evaluation correctness.
>
> The core challenge in micro-record synthesis is **joint-distribution fidelity**, i.e., capturing which variables are actually correlated and to what extent. On real-world transactional datasets, the ground-truth joint distributions across multi-way variable groups are **unknown and often impossible to estimate** reliably without very strong assumptions. This makes rigorous and reproducible evaluation difficult: Marginal alignment is straightforward for most baselines, but rigorous evaluation requires ground-truth joint distributions, which are only available in a controlled synthetic setting, because real datasets do not expose tractable multi-way correlations.
>
> Using a synthetic dataset lets us **precisely control which variable dependencies really exist**, so we can:
> 1. evaluate whether models recover the correct correlations,
> 2. compute joint JSD and Wasserstein distances faithfully,
> 3. compare models under a fully-known ground truth.
>
> That said, nothing prevents LLMSynthor from running on real retail data. To address the reviewer’s suggestion, we have now **added a new experiment on a publicly available retail dataset** (The dataset is publicly available on Kaggle at the following link: [Kaggle Link](https://www.kaggle.com/code/jijagallery/analyzing-e-commerce-transaction-data/input)), using real macro-statistics derived directly from the data. Results show that LLMSynthor continues to achieve strong macro alignment and realistic micro-record generation.
>
> | Variable   | Units_Sold↓ | Discount_Applied↓ |  Revenue↓  |  Clicks↓  | Impressions↓ | Conversion_Rate↓ | Category↓ |  Region↓  | Units_Sold & Revenue (Joint)↓ |
> |------------|:-----------:|:-----------------:|:----------:|:---------:|:------------:|:----------------:|:---------:|:---------:|:---------------------:|
> | Metrics    |     Was     |        Was        |     Was    |    Was    |      Was     |        Was       |    TVD    |    TVD    |          JSD          |
> | TabSyn     |   48.7835   |       0.0743      |   255.189  |   1.346   |    19.801    |      0.0613      |   0.105   |   0.097   |         0.534         |
> | LLMSynthor |  **11.656** |     **0.0239**    | **23.342** | **0.341** |  **10.8167** |    **0.0199**    | **0.004** | **0.003** |       **0.301**       |
>
> ---
>
> Regarding natural noise in real macro aggregates, we:
> - introduce a noise-perturbation experiment (5–10% Gaussian noise on both numeric and categorical aggregates),
> - show stable convergence, and
> - demonstrate that LLMSynthor does not overfit noisy or imperfect macro statistics.
>
>
> | Methods    |     Age    |    Gender    |   Location   |   Category   |    Price    |    Payment   | [v_A, v_G, v_C] |  [v_C, v_X]  |  [v_L, v_M]  |
> |------------|:----------:|:------------:|:------------:|:------------:|:-----------:|:------------:|:---------------:|:------------:|:------------:|
> |            |      W     |      Tvd     |      Tvd     |      Tvd     |      W      |      Tvd     |       Jsd       |      Jsd     |      Jsd     |
> | LLmSynthor |    1.13    |     0.002    |     0.002    |     0.010    |    12.762   |     0.003    |      0.071      |     0.134    |     0.007    |
> | +5% Noise  | 1.20±0.02 | 0.003±0.001 | 0.002±0.000 | 0.012±0.001 | 13.63±0.25 | 0.004±0.000 |   0.084±0.003  | 0.143±0.005 | 0.009±0.001 |
> | +10% Noise | 1.30±0.04 | 0.005±0.001 | 0.004±0.001 | 0.015±0.002 | 13.65±0.32 | 0.006±0.001 |   0.077±0.003  | 0.158±0.006 | 0.011±0.001 |
>
> Together, these experiments validate that synthetic e-commerce was chosen purely for correct joint-distribution evaluation, and LLMSynthor also performs effectively on real-world noisy macro-statistics.
>
>
> ---
>
> # Q4: Quantitative evaluation of the realism of micro-records
>
> Thank you for the suggestion. We now provide quantitative realism assessments for Section 4.2; please refer to General Response 2, where we report rejection-rate–based evaluations for both population and e-commerce data, together with detailed rule sets and analysis.

---

### Official Review · Reviewer_Edw1 · 2025-11-05

**Soundness:** 3
**Presentation:** 3
**Contribution:** 2
**Rating:** 4
**Confidence:** 2

**Summary:**

The paper introduces a new framework for synthesizing realistic micro-records (individual-level data) to align with the given macro-statistics (aggregate-level constraints). The method iteratively generates batches of synthetic micro-records, where each iteration measures the discrepancy between current synthetic macro-statistics and target aggregates. They empirically validate their method in three different applications (urban mobility, e-commerce transactions, population synthesis).

**Strengths:**

- This paper addresses a practical challenge in data synthesis: generating individual-level records when only aggregate statistics are available.
- The experimental setup on three different applications demonstrate broad applicability and outperforms the considered baselines.

**Weaknesses:**

- How do you prevent "over-correction", where repeated discrepancy-guided sampling causes loss of diversity?
- Method limitations. In real world applications, macro-statistics might be noisy or incomplete, can the method incorporate uncertainty over the target aggregates?
- Limited empirical evaluation. How sensitive is the framework to LLM quality?

**Questions:**

See above.

---

> ### Author Response · Authors · 2025-11-20
> **Overview**
>
> We thank the reviewer for their insightful questions and constructive feedback.
>
> To directly address your concerns, we have incorporated extensive new experiments in the updated manuscript:
> - **Diversity & Noise Robustness**: We have added **Figure 5** and corresponding analysis in the main text to rigorously evaluate the diversity of the generated data and the performance of LLMSynthor under varying noise levels.
> - **Backbone Robustness**: To further demonstrate the generalization capability of our framework, we have included new ablation studies using different LLM backbones (e.g., Qwen-7B/3B) in **Appendix C.1**.
>
> Please find our detailed point-by-point responses below. We hope these additional experiments and clarifications fully address your concerns, and we look forward to further discussion.

---

> ### Author Response · Authors · 2025-11-20
> **(W1 & W2) Response to W1: Diversity Preservation and Iterative Stability & W2: Robustness to Noisy or Incomplete Macro-Statistics**
>
> ## W1: Diversity Preservation and Iterative Stability
>
> We agree that naively amplifying discrepancy signals could in principle cause mode collapse, but **LLMSynthor is explicitly designed to avoid over-correction**:
>
> The discrepancy signal is **directed, bounded, and self-correcting**. As illustrated in Fig. 19 and from line 267-268 (updated manuscript), the discrepancy is computed as signed difference. This matters when:
> - When A is under-generated (Iteration 1), δ(A) > 0 prompts ''generate more A''.
> - Once A becomes sufficient (Iteration 2), δ(A) flips sign and immediately suppresses over-generation.
> - Newly emerged gaps in B or C become the largest positive δ and are corrected next.
>
> This **cyclical** balancing mechanism prevents runaway amplification and makes the updates monotonic with respect to the ℓ₁ distance between marginals and the target distribution.
>
> **In other words, the system cannot push a category indefinitely, because once the synthetic distribution surpasses the target, the signal reverses.**
>
> To justify this, we have added a diversity evaluation over the full generation process. As shown in **Figure 5 in the updated main text**, macro-level JS divergence decreases monotonically, while the diversity level increases initially and then quickly stabilizes around the ground-truth diversity. Importantly, the diversity gap remains near zero after stabilization, indicating that the discrepancy-guided updates correct only the aggregate proportions without narrowing the underlying distributional support. We observe no mode collapse or degradation, even in late iterations when macro alignment is already achieved.
>
> These results confirm that LLMSynthor’s updates are **self-balancing rather than over-corrective**, and the iterative process **does not harm sample diversity.**
>
> ---
>
> ## W2: Robustness to Noisy or Incomplete Macro-Statistics
>
> We thank the reviewer for highlighting the importance of handling uncertainty in real-world macro-statistics. We would like to emphasize that LLMSynthor naturally supports multiple forms of uncertainty modeling, because the framework does not require deterministic target values, it only requires constraints that can be either probabilistic, interval-valued, or distributional. Uncertainty can be incorporated at different levels. A simple way to incorporate uncertainty is Proposal-level uncertainty (LLM-aware noise modeling). During proposal generation, the LLM already conditions on macro-statistics to decide which joints to control.
>  If macro aggregates are noisy, we can explicitly encode the noise level (e.g., ''the mode share is 30% ± 5%'') in the prompt, allowing the LLM to:
> - generate broader or less rigid proposals,
> - avoid overcommitting to spurious spikes in noisy aggregates, and
> - allocate probability mass proportionally to uncertainty.
> In practice, this makes proposals more conservative and more stable, rather than forcing them to chase potentially noisy point estimates.
>
> To demonstrate this empirically, we include a noise-perturbation study where we inject 5–10% Gaussian noise into the target macro-statistics: for numerical aggregates, we add zero-mean Gaussian perturbations proportional to their scale, and for categorical distributions, we perturb each category proportion by Gaussian noise followed by renormalization to ensure a valid probability simplex. We then evaluate LLMSynthor against the original noise-free ground-truth aggregates to test whether the method is misled by noisy targets. LLMSynthor remains stable:
>
> | Methods    |     Age    |    Gender    |   Location   |   Category   |    Price    |    Payment   | [v_A, v_G, v_C] |  [v_C, v_X]  |  [v_L, v_M]  |
> |------------|:----------:|:------------:|:------------:|:------------:|:-----------:|:------------:|:---------------:|:------------:|:------------:|
> |            |      W     |      Tvd     |      Tvd     |      Tvd     |      W      |      Tvd     |       Jsd       |      Jsd     |      Jsd     |
> | LLmSynthor |    1.13    |     0.002    |     0.002    |     0.010    |    12.762   |     0.003    |      0.071      |     0.134    |     0.007    |
> | +5% Noise  | 1.20 ±0.02 | 0.003 ±0.001 | 0.002 ±0.000 | 0.012 ±0.001 | 13.63 ±0.25 | 0.004 ±0.000 |   0.084 ±0.003  | 0.143 ±0.005 | 0.009 ±0.001 |
> | +10% Noise | 1.30 ±0.04 | 0.005 ±0.001 | 0.004 ±0.001 | 0.015 ±0.002 | 13.65 ±0.32 | 0.006 ±0.001 |   0.077 ±0.003  | 0.158 ±0.006 | 0.011 ±0.001 |
>
> This confirms that LLMSynthor does not overfit noisy aggregates and naturally handles uncertain or incomplete macro-statistics.

---

> ### Author Response · Authors · 2025-11-20
> **Response to W3: Sensitivity to LLM Quality and Backbone Robustness**
>
> We thank the reviewer for this insightful suggestion, which has significantly strengthened our empirical evaluation. We agree that understanding the framework's dependency on LLM capability is crucial for reproducibility and broad adoption.
> Our framework is designed to be robust to model scale. The framework relies on the LLM primarily for **variable dependency inference (grouping variables)** and **proposal-level conditional generation**. These are tasks with relatively low entropy compared to open-ended creative generation; they require logical adherence to constraints rather than vast world knowledge. Therefore, we hypothesized that even smaller, open-source models would suffice.
>
> **New Empirical Evidence**: To validate this, we have expanded our evaluation to include open-source models of varying scales (**Appendix C.1 in the updated manuscript**).
>
> 1. Existing 7B Experiment (**Figure 13**): As shown in **Appendix C.1**, LLMSynthor with Qwen2.5-7B-Instruct achieves nearly identical macro-alignment performance to the primary GPT-4 backbone. This confirms that a mid-sized open-source model is sufficient for high-quality synthesis.
>
> 2. New **3B-Scale Experiment (Figure 14)**: To further push the limits of the framework, we additionally evaluated LLMSynthor with the lightweight Qwen2.5-3B-Instruct. Remarkably, the synthetic distributions remain well-aligned with the targets, and the convergence curves closely resemble those of the larger models.
>
> | Methods                 |    Rej. \%    |    Age   |   Gender  |  Location |  Category |   Price   |  Payment  | [v\_A, v\_G, v\_C] | [v\_C, v\_X] | [v\_L, v\_M] |
> |-------------------------|:-------------:|:--------:|:---------:|:---------:|:---------:|:---------:|:---------:|:------------------:|:------------:|:------------:|
> |                         |               |    Was   |    Tvd    |    Tvd    |    Tvd    |    Was    |    Tvd    |         Jsd        |      Jsd     |      Jsd     |
> | TabSyn                  |   1.9 ± 0.3   |   1.20   |   0.012   |   0.007   |   0.045   |   114.12  |   0.028   |        0.083       |     0.237    |     0.027    |
> | **Ours (GPT-4.1-nano)** | **0.3 ± 0.1** |   1.13   |   0.002   | **0.002** |   0.022   |   12.762  |   0.003   |      **0.071**     |   **0.134**  |   **0.007**  |
> | **Ours (Qwen2.5-7B)**   |   0.5 ± 0.1   |   1.03   | **0.001** |   0.013   |   0.007   | **8.691** | **0.001** |        0.089       |     0.16     |     0.021    |
> | **Ours (Qwen2.5-3B)**   |   0.5 ± 0.1   | **0.93** | **0.001** | **0.002** | **0.003** |   14.367  |   0.002   |        0.082       |     0.198    |     0.139    |
>
>
> **Detailed Quantitative Analysis (Updated Table 7): We have updated Table 7 in Appendix C.1 to provide a granular comparison.**
>
> - Marginal Alignment: All models (GPT-4.1-nano, 7B, 3B) achieve excellent marginal alignment (low Wasserstein distance), demonstrating that even the 3B model effectively follows univariate statistical guidance.
> - Joint Alignment (Trade-offs): We observe a slight degradation in joint distribution metrics (JSD) for the 3B model compared to the 7B and GPT-4.1 versions. We hypothesize that while small models excel at following explicit instructions, they have slightly weaker "planning" capabilities required to resolve complex, multi-objective conflicts during iterative refinement.
> - Conclusion: Despite this minor trade-off, the 3B model's performance remains competitive and statistically valid (rejection rates remain low).
>
> These results confirm that LLMSynthor is not critically dependent on massive proprietary LLMs. The framework is highly versatile, maintaining strong performance on consumer-grade hardware (3B/7B models), which significantly lowers the barrier to entry for practical deployment.

---

### Author Response · Authors · 2025-11-20
**General Response 3: Clarifying the Distinct Scope and Unique Capabilities of LLMSynthor**

We appreciate the reviewers’ detailed feedback. We notice that some concerns regarding baseline comparisons may stem from viewing LLMSynthor solely through the lens of traditional tabular synthesis. We would like to clarify that **LLMSynthor addresses a fundamentally broader and more challenging problem** scope than standard deep tabular generators.

---

# 1. LLMSynthor requires only macro-statistics, whereas all tabular baselines require full micro-records.

Deep tabular generators (CTGAN, TVAE, GReaT, TabSyn, etc.) learn the entire joint distribution from thousands of micro-records. Their training objective is to reproduce the full data distribution, which implicitly includes every macro-statistic used in our evaluation. In contrast, LLMSynthor operates solely from aggregate-level information. It learns no micro-level distribution from data. That LLMSynthor can match or surpass these baselines despite accessing strictly less information is a strong validation of the framework.


# 2. LLMSynthor naturally supports unstructured and hierarchical data, which tabular baseline models cannot handle.
Tabular generators are limited to fixed-dimensional tables and cannot generate hierarchical micro-structures such as households with variable-size member lists. LLMSynthor, leveraging proposal-level generation, handles structured, semi-structured, and unstructured records uniformly. Our household synthesis experiment demonstrates accurate generation of complex multi-entity records, something none of the baselines can model without significant efforts on adaptation.


# 3. LLMSynthor supports event-level simulation with arbitrary context, without retraining.
LLMSynthor can:
- **condition on arbitrary new contexts** at generation time (e.g., policy shocks, hypothetical scenarios),
- generate sequences of events rather than one-shot samples,
- adapt to new domains without any retraining or fine-tuning.
This capability is completely absent in diffusion-, GAN-, or tabular-LLM models, which must be retrained from scratch whenever the generative conditions or target distributions change.


# 4. A single unified framework handles all of the above.
 LLMSynthor provides a single mechanism, proposal-level conditional generation guided by macro discrepancies, that works across
 - pure tabular data
 - hierarchical population synthesis
 - context-conditioned event simulation
 - incomplete or noisy macro-statistics
 - settings where micro-records are entirely unavailable

---

These unique capabilities are not only theoretical but are **concretely reflected in our experimental design**. The three evaluated tasks, **tabular** synthesis, **unstructured household** generation, and **context-driven event simulation**, are deliberately chosen to highlight distinct strengths of LLMSynthor:
- Tabular synthesis illustrates LLMSynthor’s ability to rival fully-supervised baselines without micro-record access;
- Household generation demonstrates its capacity to handle hierarchical, non-tabular structures;
- Event simulation showcases flexible adaptation to arbitrary conditions without retraining.

This combination of flexibility, domain generality, and reliance on only aggregate information is not offered by any existing method. Therefore, while we benchmark LLMSynthor against tabular baselines for completeness, it is important to emphasize that these models solve a narrower problem requiring far more information.

---

**We sincerely hope the reviewers will consider these distinct strengths when evaluating the contributions of our framework.**

---

### Author Response · Authors · 2025-11-20
**General Response 2: Quantitative evaluation of micro-level Realism**

We thank the reviewer for highlighting the critical importance of micro-level realism. While our original submission included a comprehensive quantitative realism evaluation within the population-synthesis experiment. We have now moved the household-level realism evaluation to the **main text (Table 4)** to ensure high visibility.
Specifically, **Section 4.3 and Table 4** report rejection rates based on strict semantic constraints, with detailed definitions provided in **Appendix C.2.3**. We intentionally selected the population synthesis task for this evaluation because it offers rich micro-level semantics and unambiguous rules, allowing for rigorous quantification of realism without relying on subjective human annotation. These hard constraints serve as a ground-truth audit for semantic plausibility. Results show that LLMSynthor achieves significantly reduced rejection rates compared to baselines, demonstrating that our method successfully balances accurate macro-statistics with realistic micro-records.

| Check                                 |       NVI       |        CP       |       HMM       |    LLMSynthor   |      Real      |
|---------------------------------------|:---------------:|:---------------:|:---------------:|:---------------:|:--------------:|
| Overall rejection rates ↓ |                 |                 |                 |                 |                |
|   Household rejection rate            | 96.8 ± 0.3 | 73.9 ± 0.6 | 57.8 ± 0.5 | 13.3 ± 0.4 | 0.0 ± 0.0 |
|   Person rejection rate               | 45.9 ± 0.4 | 34.1 ± 0.7 | 27.8 ± 0.6 |  6.3 ± 0.2 | 0.0 ± 0.0 |
| Household-level checks ↓  |                 |                 |                 |                 |                |
|   householder_type_adult_consistency  |  4.5 ± 0.2 |  4.4 ± 0.3 | 15.2 ± 0.5 |  3.8 ± 0.2 | 0.0 ± 0.0 |
|   persons_all_valid                   | 96.7 ± 0.4 | 73.5 ± 0.6 | 56.1 ± 0.7 | 10.7 ± 0.3 | 0.0 ± 0.0 |
| Person-level checks ↓     |                 |                 |                 |                 |                |
|   age_range                           |  0.1 ± 0.0 |  0.5 ± 0.1 |  0.3 ± 0.1 |  0.0 ± 0.0 | 0.0 ± 0.0 |
|   race_valid                          |  0.0 ± 0.0 |  0.8 ± 0.1 |  0.0 ± 0.0 |  0.0 ± 0.0 | 0.0 ± 0.0 |
|   employment_age_consistency          | 41.6 ± 0.7 | 30.9 ± 0.6 | 24.5 ± 0.5 |  4.4 ± 0.2 | 0.0 ± 0.0 |
|   education_age_consistency           | 42.7 ± 0.8 |  8.1 ± 0.4 | 10.7 ± 0.5 |  2.2 ± 0.1 | 0.0 ± 0.0 |

---

To further address the reviewer’s feedback, we have conducted an additional realism analysis for the e-commerce experiment. While transaction records naturally contain fewer structural rigidities than household data (i.e., most transactions are inherently plausible), we designed a targeted set of domain-informed rules to detect subtle plausibility violations. These include category–price inconsistencies, age–category mismatches, and unrealistic purchases (e.g., high-end electronics bought by seniors). We include these results for completeness, though we maintain that the household setting offers a more rigorous testbed for realism due to its richer semantic structure.

| Methods        | R1 (\%) ↓ | R2 (\%) ↓ | R3 (\%) ↓ | Rej. (\%) ↓ |
|----------------|:---------------------:|:---------------------:|:---------------------:|:-----------------------:|
| GT (Real Data) |     0.5 ± 0.    |     0.0 ± 0.    |     0.1 ± 0.    |      0.6 ± 0.     |
| TVAE           |     1.6 ± 0.    |     0.1 ± 0.    |     0.1 ± 0.    |      1.8 ± 0.     |
| CTGAN          |     2.9 ± 0.    |     1.9 ± 0.    |     0.3 ± 0.    |      5.1 ± 0.     |
| CopulaGAN      |     2.1 ± 0.    |     2.1 ± 0.    |     0.6 ± 0.    |      4.8 ± 0.     |
| GReaT          |     3.5 ± 0.    |     0.4 ± 0.    |     0.0 ± 0.    |      3.9 ± 0.     |
| TabSyn         |     0.7 ± 0.    |     0.1 ± 0.    |     0.1 ± 0.    |      0.8 ± 0.     |
| Spada          |     3.6 ± 0.    |     0.2 ± 0.    |     0.4 ± 0.    |      4.1 ± 0.     |
| IPF            |     0.6 ± 0.    |     0.5 ± 0.    |     0.7 ± 0.    |      1.8 ± 0.     |
| LLM-ICL        |     0.0 ± 0.    |     0.0 ± 0.    |     0.0 ± 0.    |      0.0 ± 0.     |
| LLMSynthor     |     0.5 ± 0.    |     0.0 ± 0.    |     0.0 ± 0.    |      0.5 ± 0.     |

As expected, rejection rates remain low across the board, yet the results highlight our method's advantage. The LLM-ICL baseline yields the lowest rejection rate (0.0%), confirming the inherent semantic strength of LLMs. LLMSynthor achieves the next-best performance (0.5% ± 0.1%), closely matching ground-truth levels and substantially outperforming deep generative baselines such as CTGAN (5.1%), CopulaGAN (4.8%), and GReaT (3.9%). These findings confirm that LLMSynthor effectively avoids spurious attribute combinations, leveraging LLM priors to maintain semantic plausibility even without access to micro-records during training.

---

### Author Response · Authors · 2025-11-20
**General Response 1: Expanded Baseline Methods and Clarification on Fair Comparison**

We appreciate the reviewer’s constructive suggestion, which allows us to further strengthen our contribution.

---

# 1. We added three new experiments across different supervision levels
To further address this concern, we introduce three new baselines specifically designed for the macro-supervision setting.


| Methods               |   Rej. %  |  Age  | Gender |  Loc. |  Cat. |  Price  | Payment | [v_A, v_G, v_C] | [v_C, v_X] | [v_L, v_M] |
|-----------------------|:---------:|:-----:|:------:|:-----:|:-----:|:-------:|:-------:|:---------------:|:----------:|:----------:|
|                       |           |  Was  |   Tvd  |  Tvd  |  Tvd  |   Was   |   Tvd   |       Jsd       |     Jsd    |     Jsd    |
| TabSyn (**micro**)        | 1.9 ± 0.3 | 1.196 |  0.012 | 0.007 | 0.045 |  114.12 |  0.028  |      0.083      |    0.237   |    0.027   |
| Spada (**macro + micro**) | 4.1 ± 0.9 |  1.42 |  0.013 | 0.003 | 0.016 |  54.603 |  0.025  |      0.085      |    0.225   |    0.103   |
| IPF (**macro-only**)      | 3.9 ± 0.6 |  5.27 |  0.018 | 0.026 | 0.012 | 139.957 |  0.035  |      0.117      |    0.504   |    0.113   |
| LLM-ICL (**macro-only**)  | 0.1 ± 0.0 | 20.61 |  0.026 | 0.024 | 0.441 | 279.743 |  0.163  |      0.437      |    0.591   |    0.476   |
| Ours (**macro-only**)     | 0.3 ± 0.1 |  1.13 |  0.002 | 0.002 | 0.022 |  12.762 |  0.003  |      0.071      |    0.134   |    0.007   |



First, we modify a recent LLM-based hybrid tabular generator SPADA [1] to include explicit macro-level supervision during training, even with access to both micro- and macro-statistics, its macro-alignment remains weaker than LLMSynthor.

Second, we incorporate Iterative Proportional Fitting (IPF) [2] as a classical macro-only simulator, despite having direct access to all marginal and joint constraints, it produces micro-records with significantly poorer realism.

Third, we evaluate an LLM-ICL [3] baseline in which GPT-5.1 is given the entire set of macro-statistics as in-context prompts and asked to directly generate micro-records. We implemented this baseline through a 40-turn interactive session in the ChatGPT interface, where the model benefits from OpenAI’s built-in conversational memory and can fully leverage all previously generated records. This setup provides the ICL baseline with the strongest possible conditions for macro-aware generation. Nevertheless, its performance remains substantially worse than LLMSynthor, underscoring that one-shot or conversational ICL is insufficient without iterative discrepancy-guided correction, which is essential for stable macro-level alignment.

Together, these experiments demonstrate that LLMSynthor is not advantaged by ''leakage'' of evaluation metrics. On the contrary, it outperforms both micro-supervised, macro-supervised, and macro-only baselines despite relying on strictly less information.

---


# 2. Core clarification
LLMSynthor only sees the target macro-statistics. In the tabular synthesis task (i.e., E-commerce transactions synthesis), the marginal statistics and dependency structure are inferred through the Variable Dependence Inference Module. While tabular baselines (TVAE, CTGAN, CopulaGAN, TabSyn, GReaT, etc.) are trained on full micro-records, meaning they observe:
- all one-way marginals
- all multi-way joint distributions
- all conditional distributions
- and thus all macro-statistics used in our evaluation
This means: Baselines have strictly more supervision and a strictly easier task. LLMSynthor solves the harder setting: generate micro-records using only aggregates.

---

# 3. Why baseline training already optimizes macro-statistics
All tabular generative models, including VAEs, GANs, diffusion models, and LLM-based tabular generators,optimize an objective equivalent to minimizing a divergence between the true data distribution $p(x)$ and the model distribution $q_\theta(x)$:
$$\min_\theta D\bigl(p(x) \| q_\theta(x)\bigr)$$

For example, Diffusion-based tabular generators minimize denoising score matching, which is equivalent to:
$$\min_\theta \left\| \nabla_x \log p(x) - \nabla_x \log q_\theta(x) \right\|_2^2.$$

Because every macro-statistic is an expectation under the data distribution,
$$S_{\mathrm{macro}} = \mathbb{E}_{x\sim p(x)}[f(x)],$$

Any method that fits $q_\theta(x) \approx p(x)$ necessarily matches **all** macro-statistics. Thus, baselines implicitly optimize all evaluation metrics simply by training on the full micro-records.

---

# 4. LLMSynthor uses less supervision and is therefore solving a harder problem
LLMSynthor never observes micro-records.
It does not see:
- any micro-level correlations not included in the macro-input
- any micro-level support constraints
- any long-tail structure or higher-order dependencies
Thus, our method performs well despite missing information that all baselines rely on. This is exactly the setting many real-world agencies face, they only have aggregated statistics, not individual data.

---

### Author Response · Authors · 2025-11-20
**Overview: Summary of Revisions and Methodological Contributions**

We sincerely thank the reviewers for their constructive feedback and insightful suggestions. These comments have significantly strengthened the rigor and presentation of our work. In this revision, we have conducted extensive new experiments and structural updates to address the concerns raised:
1. **Expanded Baseline Suite & Fair Comparison**: To better contextualize our performance, we have expanded our evaluation to distinguish between micro-record supervised (e.g., TabSyn, GReaT) and macro-only baselines. This explicitly highlights our framework's advantage: LLMSynthor achieves competitive or superior performance despite accessing strictly less information (aggregate-only) compared to baselines that require sensitive micro-record training.
2. **Rigorous Realism Evaluation (Moved to Main Text)**: We have conducted a comprehensive quantitative analysis of record-level realism across both population and e-commerce domains.
- Results: As shown in the new Table 4 (Main Text) and Appendix C.2.3, LLMSynthor achieves near-zero rejection rates, significantly outperforming deep generative models (which frequently produce semantically invalid combinations) and demonstrating superior semantic plausibility.
- Visibility: We have moved these key findings to the main text to underscore the semantic validity of our generated micro-records.
3. **Comprehensive Stability, Efficiency, and Interpretability Analysis**: We have added a new stability analysis in Figure 5 (Main Text) and a detailed breakdown in Appendix C.1 (Figures 11-14, Tables 6-7).
- **Diversity & Bias**: We demonstrate that our discrepancy-guided mechanism effectively steers generation toward ground-truth diversity while progressively mitigating bias.
- **Efficiency**: New runtime studies confirm that with only 10 iterations, LLMSynthor surpasses the GReaT baseline in both speed and quality.
- **Interpretability**: We provide further evidence that our proposal-level generation offers transparency unavailable in black-box neural baselines.

---
Clarified Reiteration of **Unique Contributions**:

While we include comparisons with standard tabular generators for completeness, we respectfully emphasize that **LLMSynthor is designed to tackle a fundamentally broader and more difficult problem**. Unlike baselines that rely on memorizing micro-records, our framework reconstructs data distributions **using only privacy-preserving aggregate statistics**. It can naturally **handle hierarchical and unstructured data**, such as variable-sized households, which lie beyond the capabilities of fixed-schema tabular models. Moreover, LLMSynthor **supports zero-shot adaptation** to novel contexts without the need for retraining. By offering **a unified approach** to these diverse challenges, our method addresses a critical gap that existing baselines are not equipped to fill.

**We hope the reviewers will take these distinct capabilities into account when evaluating the scope and contribution of our work.**
We deeply appreciate the time and effort the reviewers have invested in evaluating our work. We look forward to the discussion phase and are more than willing to provide further details or experiments to fully address any remaining questions.

---

### Meta-Review · Area_Chair_hNU6 · 2026-01-07

**Summary:**

Reviewers broadly agreed that the problem of synthesizing micro-level data from macro-only statistics is important and timely. However, the primary concerns centered on the soundness and completeness of the proposed methodology. In particular, reviewers questioned whether macro-level alignment and limited rule-based audits are sufficient to establish realistic micro-level data generation, especially for complex or unobserved dependencies. Additional concerns focused on the heavy reliance on LLM priors for dependency inference, the auditability and verifiability of the generation process, and the lack of strong guarantees regarding robustness when macro statistics are incomplete or noisy. While the rebuttal adds valuable empirical analyses and clarifications, these concerns reflect fundamental methodological uncertainties rather than missing experiments, and therefore remain only partially addressed.

**Reviewer Concerns:**

While the rebuttal adds meaningful experiments, expanded baselines, and clarifications, particularly regarding robustness to noisy macro-statistics, the core methodological concerns remain only partially addressed. Macro-level alignment and rule-based realism checks improve empirical confidence but do not fully establish that the synthesized micro-records capture realistic joint structure beyond audited constraints. The approach still relies heavily on LLM-driven dependency inference and proposal generation, with limited guarantees against bias, spurious correlations, or failure modes introduced by model priors. Overall, the response strengthens the empirical evidence but does not fully resolve the foundational uncertainty about micro-level fidelity and interpretability.

**Reviewer Scores:**

Reviewer Edw1 would likely hold their score at 4, as the added diversity, noise robustness, and backbone ablation experiments address several concerns, but questions about broader applicability and methodological assumptions remain. Reviewer 5i4z may raise their score from 4 to 6, as the authors substantially expanded the baseline suite, clarified evaluation fairness, added quantitative micro-level realism analyses in the main text, and incorporated real-world dataset validation, directly addressing this reviewer’s core criticisms. Reviewer isfU would likely hold their score at 4, since metric clarifications and efficiency analyses improve transparency, but concerns about scalability, comparability, and reliance on LLM priors are not fully resolved. Reviewer Xt7k would likely hold their score at 2, as fundamental issues regarding auditability, realism guarantees beyond rule-based checks, dependence on LLM-inferred dependencies, and methodological assumptions persist despite revisions.

---

### Decision · Program_Chairs · 2026-01-26

Reject